# MOMENTUM PARTICLE MAXIMUM LIKELIHOOD

## ABSTRACT

Maximum likelihood estimation (MLE) of latent variable models is often recast as an optimization problem over the extended space of parameters and probability distributions. For example, the Expectation Maximization (EM) algorithm can be interpreted as coordinate descent applied to a suitable free energy functional over this space. Recently, this perspective has been combined with insights from optimal transport and Wasserstein gradient flows to develop particle-based algorithms applicable to wider classes of models than standard EM.

Drawing inspiration from prior works which interpret 'momentum-enriched' optimisation algorithms as discretizations of ordinary differential equations, we propose an analogous dynamical systems-inspired approach to minimizing the free energy functional over the extended space of parameters and probability distributions. The result is a dynamic system that blends elements of Nesterov's Accelerated Gradient method, the underdamped Langevin diffusion, and particle methods.

Under suitable assumptions, we establish quantitative convergence of the proposed system to the unique minimiser of the functional in continuous time. We then propose a numerical discretization of this system which enables its application to parameter estimation in latent variable models. Through numerical experiments, we demonstrate that the resulting algorithm converges faster than existing methods and compares favourably with other (approximate) MLE algorithms.

## 1 INTRODUCTION

In this work, we are interested in parameter estimation for (probabilistic) latent variable models of the form $p_\theta(y, x)$, where $\theta \in \mathbb{R}^{d_\theta}$ denotes the model parameters, $x \in \mathbb{R}^{d_x}$ is an unobserved (or latent) variable and $y \in \mathbb{R}^{d_y}$ is an observed variable. Throughout this work, we treat $y$ as fixed. The type II maximum likelihood approach (Good, 1983) aims to estimate $\theta$ by maximization of the *marginal* likelihood $p_\theta(y) := \int p_\theta(y, x) \, \mathrm{d}x$. However, in various models of practical interest, this integral is not available in closed form, so direct optimisation of this objective is infeasible.

One approach to circumventing this intractability is to construct a tractable objective which is defined over an extended space, and whose optimal values coincide with that of the original maximum likelihood estimation (MLE) problem. For latent variable models, this can be carried out as follows (Dempster et al., 1977): writing $\mathcal{P}\left(\mathbb{R}^{d_x}\right)$ for the space of probability distributions over the latent space, define a 'free energy' functional $\mathcal{E} : \mathbb{R}^{d_\theta} \times \mathcal{P}\left(\mathbb{R}^{d_x}\right) \to \mathbb{R}$ by

$$\mathcal{E}(\theta, q) := \int q(x) \log \left( \frac{q(x)}{p_\theta(y, x)} \right) \, \mathrm{d}x.$$

One can then compute directly that $\mathcal{E}(\theta, q) = -\log p_\theta(y) + \mathsf{KL}(q, p_\theta(\cdot \mid y))$, where $\mathsf{KL}$ is the Kullback-Leibler divergence. Since it holds that $\min \left\{ \mathcal{E}(\theta, q) : q \in \mathcal{P}\left(\mathbb{R}^{d_x}\right) \right\} = -\log p_\theta(y)$, minimizing the free energy functional over the extended space of parameters and probability distributions (over the latent space), one will minimize the function $\theta \mapsto -\log p_\theta(y)$, hence solving the original maximum likelihood estimation problem.

This representation suggests exploring practical procedures for minimising the free energy functional $\mathcal{E}$. Neal and Hinton (1998) describe that the classical EM algorithm is precisely coordinate descent applied to $\mathcal{E}$. In particular, the EM approach is readily applicable to models for which the conditional distributions $p_\theta(x \mid y)$ are available in closed form, precisely because this enables the minimization of $q$ with $\theta$ held fixed to be carried out tractably. Recent work of Kuntz et al. (2023) has sought to move

beyond this class of models by modelling the distribution $q$ nonparametrically. Drawing inspiration from the mature literature on gradient flows (e.g., see Ambrosio et al. (2005)), this work constructs several gradient flows of the functional $\mathcal{E}$ over the joint space of extended space of parameters and probability distributions over the latent space. The resulting flow, called particle Gradient Descent (PGD), simultaneously evolves a parameter estimate, written $\theta_t$, and an estimate of the posterior represented by a finite particle system, written $q_t$. These gradient flows are then discretized in time, yielding an algorithm qualitatively similar to gradient descent of $\mathcal{E}$.

In convex optimization, it is well understood that while the gradient descent method has many favourable theoretical guarantees, it is also suboptimal. In particular, under standard computational models (Nemirovskij and Yudin, 1983), there are practical algorithms which yield substantially better worst-case convergence guarantees and improved practical performance (e.g., see Nesterov (1983)). Among optimization algorithms which achieve these 'accelerated' convergence rates, a common feature is the presence of 'momentum' effects in the dynamics of the algorithm. Roughly speaking, gradient descent of a function $f$ is analogous to solving the ordinary differential equation (ODE) $\dot{\theta}_t = -\nabla f(\theta_t)$, then a 'momentum-type' method might instead resemble solving a second-order ODE like $\ddot{\theta}_t + \gamma \dot{\theta}_t + \nabla f(\theta_t) = 0$ for some 'damping' parameter $\gamma > 0$ (Su et al., 2014; Wibisono et al., 2016). In the context of machine learning, Sutskever et al. (2013) demonstrated empirically that incorporating momentum effects into stochastic gradient descent (SGD) could have substantial benefits, such as eliminating the performance gap between standard SGD and competing 'Hessian-free' deterministic methods based on higher-order differential information (Martens, 2010).

Inspired by the success of momentum methods, the present work seeks to explore whether PGD can similarly be accelerated by utilizing 'momentum' strategies. Our contributions are as follows: (1) we construct a continuous-time flow that incorporates momentum effects into PGD; (2) under suitable conditions, we establish convergence in continuous time; (3) we derive a discretization of the flow that can achieve better performance than PGD and compare competitively against other methods.

The structure of this manuscript is as follows: in Section 2, we review existing work on gradient flows on Euclidean space $\mathbb{R}^{d_\theta}$, probability space $\mathcal{P}(\mathbb{R}^{d_x})$, and extended space $\mathbb{R}^{d_\theta} \times \mathcal{P}(\mathbb{R}^{d_x})$. We also review momentum methods on Euclidean space $\mathbb{R}^{d_\theta}$, and probability space $\mathcal{P}(\mathbb{R}^{d_x})$. Then, in Section 3, we describe how momentum can be incorporated into the PGD; the result is a momentum-enrich dynamical system called Momentum PGD (MPGD). After, in Section 4, we establish convergence of the continuous-time MPGD. In the Section 5, we describe how to discretize the continuous MPGD algorithm to produce our proposed algorithm. Finally, in Section 6, we study on the toy examples of how the choice of momentum parameters affects the optimization process, as well as the effects of various choices in the discretization process. As a large-scale experiment, we compare our proposed method for training Variational Autoencoders (Kingma and Welling, 2014) against current methods of training latent variable models.

## 2 BACKGROUND

The main goal of the paper is to propose algorithmic improvements to the PGD algorithm of Kuntz et al. (2023), which is derived as a gradient flow on an extended space of parameters and probability distributions, by incorporating some notion of momentum into the dynamics. Towards this goal, we first provide an overview of standard gradient flows on these spaces while detailing precisely how momentum ideas have been successfully incorporated in related scenarios.

### 2.1 GRADIENT FLOWS ON $\mathbb{R}^d$, AND THEIR ACCELERATION

Given a function $f : \mathbb{R}^d \to \mathbb{R}$ of sufficient regularity, the (Euclidean) gradient flow of $f$ is given by the ODE: $\dot{\theta}_t = -\nabla_\theta f(\theta_t)$ where $\nabla_\theta f$ is the standard (Euclidean) gradient of $f$, i.e., the vector such that $\theta \approx \tilde{\theta} \implies f(\tilde{\theta}) = f(\theta) + \langle \nabla_\theta f(\theta), \tilde{\theta} - \theta \rangle + o\left(\|\tilde{\theta} - \theta\|_2\right)$. Discretizing the gradient flow in time with a standard forward Euler integrator yields the celebrated gradient descent method. Thus, it is natural to ask if "accelerated" gradient descent, such as Nesterov (1983)'s Accelerated Gradient (NAG) and Polyak (1964)'s Heavy ball, can be viewed from the lens of discretizations of a dynamical system. This has proved a fruitful research question (Su et al., 2014; Shi et al., 2021). A recurring pattern observed is that many successful accelerated algorithms can be viewed as discretizations

of a certain 'momentum-enriched' system, i.e., corresponding to the evolution of a kinetic particle that carries both a position and a momentum. For example, Nesterov's Accelerated Gradient (NAG) method (Nesterov, 1983) can be seen as a discretization of Wilson et al. (2016, Equation 7)'s system of ODEs

$$\dot{\theta}_t = \eta m_t, \quad \dot{m}_t = -\gamma \eta m_t - \nabla_\theta f(\theta_t), \tag{1}$$

where $\gamma, \eta$ are positive hyperparameters (see Appendix I.2). For convex, quadratic $f$, this system reproduces the dynamics of a damped harmonic oscillator (McCall, 2010), and this equips the hyperparameters of the ODE with intuitive interpretations: $\eta^{-1}$ represents the mass of the particle, and $\gamma$ determines the strength of the damping. Additionally, in the massless limit (i.e. sending $\eta \to \infty$ and rescaling time appropriately), one can formally recover that the solution follows the gradient flow $\dot{\theta}_t = -\nabla_\theta f(\theta_t)$.

Towards generalising this construction, it is useful to introduce the 'Hamiltonian' function $F_\eta(\theta, m) := f(\theta) + \frac{\eta}{2}\|m\|_2^2$, as well as the 'damping matrix' $\mathsf{D}_\gamma := \begin{pmatrix} 0_d & -I_d \\ I_d & \gamma I_d \end{pmatrix}$, which, upon abbreviating $\vartheta = (\theta, m)$, allows us to express this system of ODEs as

$$\dot{\vartheta}_t = -\mathsf{D}_\gamma \nabla_\vartheta F_\eta(\vartheta_t).$$

This identifies the system as an instance of a 'damped' or 'conformal' Hamiltonian flow; e.g., see McLachlan and Perlmutter (2001); Maddison et al. (2018). Stated concisely, we observe that NAG admits an interpretation as a '$\gamma$-damped Hamiltonian flow of $F_\eta$'. Related observations were made in Wibisono et al. (2016); Wilson et al. (2016).

## 2.2 Gradient Flows on $\mathcal{P}(\mathbb{R}^d)$, and their Acceleration

At least in abstract, one can consider following a similar program for solving optimisation tasks over the space of probability distributions on Euclidean space, $\mathcal{P}(\mathbb{R}^d)$. One challenge in making this idea practical is how to endow this space with a suitable geometric structure compatible with relevant algorithmic primitives. That is, when considering functionals $\mathcal{F} : \mathcal{P}(\mathbb{R}^d) \to \mathbb{R}$ and distributions $q, \tilde{q}$ which are 'close' to one another in a relevant sense, we want to be able to make estimates of the form $\mathcal{F}(\tilde{q}) \approx \mathcal{F}(q) + \mathrm{D}\mathcal{F}[q](\tilde{q} - q)$, for some suitable notion of derivative $\mathrm{D}\mathcal{F}$. In our work, we consider the geometry induced by the Wasserstein-2 metric (Jordan et al., 1998). Other choices for distances will generate different geometries and algorithms (e.g., the Stein geometry (Duncan et al., 2023; Liu, 2017; Sharrock and Nemeth, 2023)).

The important objects are then as follows: given a functional $\mathcal{F} : \mathcal{P}(\mathbb{R}^d) \to \mathbb{R}$ and distributions $q, \tilde{q} \in \mathcal{P}(\mathbb{R}^d)$, define the functional gradient of $\delta_q \mathcal{F} : \mathcal{P}(\mathbb{R}^d) \to (\mathbb{R}^d \to \mathbb{R})$ so that $q \approx \tilde{q} \implies \mathcal{F}(\tilde{q}) \approx \mathcal{F}(q) + \int \delta_q \mathcal{F}[q](x)(\tilde{q}(x) - q(x)) \,\mathrm{d}x$. Now, let $v$ denote a vector field, let $\epsilon > 0$, and write $q_\epsilon$ for the law of $x + \epsilon v(x)$ when $x \sim q$. Under suitable assumptions, one can then compute that $\mathcal{F}(q_\epsilon) = \mathcal{F}(q) + \epsilon \int q(\mathrm{d}x) \langle \nabla_x \delta_q \mathcal{F}[q](x), v(x) \rangle + o(\epsilon)$ as $\epsilon \to 0^+$, where $\langle \cdot, \cdot \rangle$ is the Euclidean inner product (Ambrosio et al., 2005, Lemma 10.4.1). Therefore, the Wasserstein gradient of $\mathcal{F}$ is defined as $\nabla_W \mathcal{F}(q) = -\nabla_x \cdot (q \nabla_x \delta_q \mathcal{F}[q])$. Interpreting gradients from the perspective of steepest descent, this expression reflects the notion that in order to reduce $\mathcal{F}(q)$ by updating $q$ by a perturbation which is small in the sense of the 2-Wasserstein geometry, it would be suitable to take each $x$ in the support of $q$ and transport it towards $x - \epsilon \nabla_x \delta_q \mathcal{F}[q](x)$ (for more details, see Appendix C.1).

Having defined these notions, the Wasserstein gradient flow (WGF) of the functional $\mathcal{F}$ then corresponds to the partial differential equation (PDE) $\dot{q}_t = -\nabla_W \mathcal{F}(q_t)$. A significant example of such a flow is given by taking $\mathcal{F} : q \mapsto \mathsf{KL}(q, p)$ for some fixed $p \in \mathcal{P}(\mathbb{R}^d)$. In this case, one can compute $\delta_q \mathcal{F}[q] = \log \frac{q}{p}$ and so, the WGF is given by:

$$\dot{q}_t = -\nabla_W \mathcal{F}(q_t) = \nabla_x \cdot \left( q_t \nabla_x \log \left( \frac{q_t}{p} \right) \right) = -\nabla_x \cdot (q_t \nabla_x \log p) + \Delta_x q_t.$$

An insight of Jordan et al. (1998) was to observe that this PDE is precisely the Fokker-Planck PDE corresponding to the overdamped Langevin SDE with invariant measure $p$, i.e., $\mathrm{d}X_t = \nabla_x \log p(x) \,\mathrm{d}t + \sqrt{2}\,\mathrm{d}W_t$ where $W_t$ denotes the standard Wiener process. As such, we can interpret

the simulation of the overdamped Langevin SDE as gradient descent of $\mathcal{F}$. Following the earlier discussion of momentum-enriched dynamical systems for accelerated optimisation over $\mathbb{R}^d$, it is natural to ask whether, with a suitably modified objective functional, and a suitable interpretation of damped Hamiltonian flows, can one obtain accelerated methods for solving optimisation problems over the space of probability distributions?

This question was answered in the affirmative by Ma et al. (2021), who proceed in two steps. Firstly, instead of working with the functional $\mathcal{F} : \mathcal{P}\left(\mathbb{R}^d\right) \to \mathbb{R}$ which sends $q$ to $\mathsf{KL}\left(q, p\right)$, they fix a hyperparameter $\eta > 0$ and define a new functional on the space of momentum-enriched probability distributions, i.e. $\mathcal{G}_\eta : \mathcal{P}\left(\mathbb{R}^d \times \mathbb{R}^d\right) \to \mathbb{R} : q \mapsto \mathsf{KL}\left(q, p \otimes r_\eta\right)$ where $r_\eta = \mathcal{N}\left(u \mid 0, \eta^{-1} I_d\right)$ and $p \otimes q$ is the product measure; $u$ will denote momentum variables which are conjugate to $x$ in all that follows. Secondly, they extend the notion of a gradient flow over $\mathcal{P}\left(\mathbb{R}^d\right)$ to the notion of a $\gamma$-damped Hamiltonian flow of $\mathcal{G}_\eta$ over $\mathcal{P}\left(\mathbb{R}^d \times \mathbb{R}^d\right)$ as follows: abbreviating $\Upsilon := (x, u)$, define an evolution of probability distributions $q_t \in \mathcal{P}\left(\mathbb{R}^d \times \mathbb{R}^d\right)$ by $\dot{q}_t = \nabla_\Upsilon \cdot \left(q_t \mathsf{D}_\gamma \nabla_\Upsilon \delta_q \mathcal{G}_\eta\left[q_t\right]\right)$. Thirdly, much as Jordan et al. (1998) observe that the WGF of $\mathcal{F}$ corresponds to the evolution of the overdamped Langevin SDE, Ma et al. (2021) observe that this damped Hamiltonian flow admits an analogous correspondence with the underdamped Langevin SDE. This amounts to the claim that the underdamped Langevin SDE admits an interpretation as a '$\gamma$-damped Hamiltonian flow of $\mathcal{G}_\eta$' in the space of probability distributions. Finally, Ma et al. (2021) shows that under suitable conditions, this process converges to its stationary distribution more rapidly than the corresponding overdamped Langevin SDE. As such, the transition from gradient flow to damped Hamiltonian flow is again able to deliver accelerated rates of convergence.

## 2.3 Gradient Flows on $\mathbb{R}^{d_\theta} \times \mathcal{P}\left(\mathbb{R}^{d_x}\right)$

Combining the previous two settings, we now consider the optimization of objectives defined over a product of an Euclidean space and probability distributions over another Euclidean space (potentially of distinct dimension). Given a functional $\mathcal{E} : \mathbb{R}^{d_\theta} \times \mathcal{P}\left(\mathbb{R}^{d_x}\right) \to \mathbb{R}$, the 'extended' gradient flow of $\mathcal{E}$ with respect to the product Euclidean-Wasserstein geometry is given by

$$\dot{\theta}_t = -\nabla_\theta \mathcal{E}\left(\theta_t, q_t\right), \quad \dot{q}_t = -\nabla_W \mathcal{E}\left(\theta_t, q_t\right),$$

where $\nabla_W$ denotes the Wasserstein gradient with respect to the $q$ variable.

In Kuntz et al. (2023), this extended gradient flow is used as a tool for solving maximum likelihood estimation problems for latent variable models. In particular, they seek to minimize the objective functional $\mathcal{E}\left(\theta, q\right)$. Thus, the extended gradient flow takes the form

$$\dot{\theta}_t = \int q_t\left(\mathrm{d}x\right) \nabla_\theta \ell\left(\theta_t, x\right), \quad \dot{q}_t = \nabla_x \cdot \left(q_t\left(x\right) \nabla_x \left(\log q_t\left(x\right) - \ell\left(\theta_t, x\right)\right)\right),$$

where $\ell(\theta, x) := \log \rho_\theta(x)$ and $\rho_\theta(x) := p_\theta(y, x)$. Following the approach of Jordan et al. (1998), we can interpret the gradient flow of $\mathcal{E}$ as describing the evolution of the coupled ODE-SDE system

$$\dot{\theta}_t = \int q_t\left(\mathrm{d}x\right) \nabla_\theta \ell\left(\theta_t, x\right), \quad \mathrm{d}X_t = \nabla_\theta \ell\left(\theta_t, X_t\right) \mathrm{d}t + \sqrt{2}\, \mathrm{d}W_t,$$

where $q_t := \mathrm{Law}\left(X_t\right)$. Since the drift of $\theta_t$ depends on the law of $X_t$, this is said to be a 'distribution-dependent' or 'McKean-Vlasov' process (Kac, 1956; McKean Jr, 1966). PGD is obtained via discretization in time (see Appendix I.1).

## 3 Momentum Particle-based Maximum Likelihood Flow

Drawing inspiration from prior works as detailed in Section 2, we endeavour to accelerate gradient flows over the extended space of parameters and probability distributions by i) enriching both components of the space with momentum variables, ii) defining a suitable extension of the functional of interest which is defined over this momentum-enriched extended space, iii) constructing a suitable damped Hamiltonian flow on this space, and iv) interpreting the resulting flows as a coupled system of ODEs and SDEs, which can then be discretised. In particular, we will carry out this program to describe a (doubly-)momentum-enriched modification of the PGD algorithm. First, we enrich both spaces with a momentum variable, which is performed by equipping the parameter space

with momentum $m \in \mathbb{R}^{d_\theta}$, and the latent space with momentum $u \in \mathbb{R}^{d_x}$. Accordingly, we move from working on the space of probability distributions over the latent space $\mathbb{R}^{d_x}$ to the space of probability distributions over the momentum-enriched latent space, i.e. $\mathcal{P}\left(\mathbb{R}^{d_x} \times \mathbb{R}^{d_x}\right)$. Subsequently, we fix positive friction hyperparameters $(\eta_\theta, \eta_x)$ (corresponding to the parameter and latent spaces respectively), define the (unnormalised) momentum-enriched joint target law $\rho_{\theta,\eta_x}(x,u) := p_\theta(y,x) r_{\eta_x}(u)$ (where again we denote $r_{\eta_x}(u) = \mathcal{N}\left(u \mid 0, \eta_x^{-1} I_{d_x}\right)$), and finally, define the doubly momentum-enriched free energy functional $\mathcal{F}_{\eta_\theta, \eta_x} : \mathbb{R}^{d_\theta} \times \mathbb{R}^{d_\theta} \times \mathcal{P}\left(\mathbb{R}^{d_x} \times \mathbb{R}^{d_x}\right)$ as

$$\mathcal{F}_{\eta_\theta, \eta_x}(\theta, m, q) := \int q(x,u) \log \left( \frac{q(x,u)}{\rho_{\theta,\eta_x}(x,u)} \right) \mathrm{d}x\,\mathrm{d}u + \frac{\eta_\theta}{2}\|m\|_2^2.$$

We now define the extended $(\gamma_\theta, \gamma_x)$-damped Hamiltonian flow of $\mathcal{F}_{\eta_\theta, \eta_x}$ as (abbreviating $\vartheta = (\theta, m)$ and $\Upsilon = (x, u)$)

$$\dot{\vartheta}_t = -\mathsf{D}_{\gamma_\theta} \nabla_\vartheta \mathcal{F}_{\eta_\theta, \eta_x}(\vartheta_t, q_t)$$
$$\dot{q}_t = \nabla_\Upsilon \cdot (q_t \mathsf{D}_{\gamma_x} \nabla_\Upsilon \delta_q F_{\eta_\theta, \eta_x}[\vartheta_t, q_t]).$$

Following a similar path to earlier derivations, this evolution corresponds to the coupled, momentum-enriched ODE-SDE system

$$\mathrm{d}\theta_t = \eta_\theta m_t\,\mathrm{d}t \tag{2a}$$
$$\mathrm{d}m_t = -\gamma_\theta \eta_\theta m_t\,\mathrm{d}t - \nabla_\theta \mathcal{E}(\theta_t, q_{t,X})\mathrm{d}t \tag{2b}$$
$$\mathrm{d}X_t = \eta_x U_t\,\mathrm{d}t \tag{2c}$$
$$\mathrm{d}U_t = -\gamma_x \eta_x U_t\,\mathrm{d}t + \nabla_\theta \ell(\theta_t, X_t)\,\mathrm{d}t + \sqrt{2\gamma_x}\,\mathrm{d}W_t, \tag{2d}$$

where $q_{t,X} = \mathrm{Law}(X_t)$. This is again a McKean-Vlasov process. In Proposition 3.1, we establish the existence and uniqueness of strong solutions to (2). The proof can be found in Appendix G.

**Proposition 3.1** (Existence and Uniqueness of strong solutions to (2)). *Under Assumption 2, there exists a unique strong solution to (2) for any initial condition $(\vartheta_0, q_0) \in \mathbb{R}^{2d_\theta} \times \mathcal{P}(\mathbb{R}^{2d_x})$.*

The choice of the hyperparameters $(\eta_\theta, \gamma_\theta, \eta_x, \gamma_x)$ is crucial to the practical performance of MPGD. We observe that (as is also the case for NAG, the underdamped Langevin SDE, and other similar systems) there are three different qualitative regimes for the dynamics: i) the underdamped regime, in which the parameter values oscillate, ii) the overdamped regime, in which one recovers PGD-type behaviour, and iii) the critically-damped regime, in which oscillations are largely suppressed, but the momentum effects are still able to accelerate the convergence behaviour relative to PGD. However, obtaining parameters to achieve critical damping is problem-dependent and, at the time of writing, rigorous approaches are limited to simple targets (e.g., see Dockhorn et al. (2022)).

## 4  CONVERGENCE OF THE FLOW

In this section, we study the quantitative convergence properties of the flow described in Section 3. Following the work of Wilson et al. (2016); Wibisono et al. (2016); Ma et al. (2021), we establish convergence through a suitable Lyapunov argument (Lyapunov, 1992). In particular, we will exhibit a nonnegative function $\mathcal{L}$ which majorises $\mathcal{F}$ and is contracted at an exponential rate along the evolution of the flow. From this, it will be straightforward to deduce estimates on the convergence of $\mathcal{F}$ to its minimal value.

**Assumption 1** (Extended Log Sobolev Inequality). *Suppose that $\theta^*$ maximizes the marginal likelihood, i.e., $\theta^* = \arg\sup_{\theta \in \mathbb{R}^{d_\theta}} p_\theta(y)$, and that there exists a constant $C_\mathcal{E} > 0$, such that*

$$\mathcal{E}(\theta, q) + \log p_{\theta^*}(y) \leq \frac{1}{2C_\mathcal{E}} \left\|\nabla \mathcal{E}(\theta, q)\right\|_q^2, \quad \forall \theta \in \mathbb{R}^{d_\theta},\ q \in \mathcal{P}(\mathbb{R}^{d_x}),$$

*where*

$$\left\|\nabla \mathcal{E}(\theta, q)\right\|_q^2 := \left\|\nabla_\theta \mathcal{E}(\theta, q)\right\|^2 + \left\|\nabla_x \log q - \nabla_x \log \rho_\theta\right\|_q^2. \tag{3}$$

**Assumption 2** (K-Lipschitz Gradient). *We assume that the potential $\ell(\theta, x)$ has a K-Lipschitz gradient. More precisely, there exists some constant $K > 0$ such that for all*

$$\left\|\nabla_{(\theta,x)} \ell(\theta, x) - \nabla_{(\theta,x)} \ell(\theta', x')\right\| \leq K\left(\|x - x'\| + \|\theta - \theta'\|\right), \quad \forall \theta \in \mathbb{R}^{d_\theta}, \forall x \in \mathbb{R}^{d_x}.$$

As a first step towards this, one can compute the time derivative of $\mathcal{F}$ along the flow as

$$\dot{\mathcal{F}} = -\gamma_\theta \left\| \nabla_m \mathcal{F} \right\|^2 - \gamma_x \left\| \nabla_u \delta_q \mathcal{F} \right\|^2 \leq 0;$$

see Proposition E.1 for the calculations. This establishes that $\mathcal{F}$ is non-increasing along the flow, and while providing some useful a priori assurances on the stability of the flow, is not sufficient for establishing convergence to the minimiser of $\mathcal{F}$. This is due in part to the degeneracy of this derivative term, noting that it contains no gradients with respect to $(\theta, x)$; this is a common phenomenon for momentum-enhanced dynamical systems. As such, we follow the constructions of Wilson et al. (2016); Ma et al. (2021) and add a sort of 'twisted' gradient term, to re-introduce these missing gradients. We proceed as follows: let $(\tau_\theta, \tau_{\theta m}, \tau_m, \tau_x, \tau_{xu}, \tau_x)$ be non-negative constants such that the matrices

$$T_{(\theta,m)} = K^{-1} \begin{pmatrix} \tau_\theta I_{d_\theta} & \frac{\tau_{\theta m}}{2} I_{d_\theta} \\ \frac{\tau_{\theta m}}{2} I_{d_\theta} & \tau_m I_{d_\theta} \end{pmatrix}, \quad T_{(x,u)} = K^{-1} \begin{pmatrix} \tau_x I_{d_x} & \frac{\tau_{xu}}{2} I_{d_x} \\ \frac{\tau_{xu}}{2} I_{d_x} & \tau_u I_{d_x} \end{pmatrix}$$

are positive-semidefinite, with $K$ the Lipschitz gradient constant of Assumption 2. Define now

$$\left\| \nabla_{(\theta,m)} \mathcal{F} \right\|_{T_{(\theta,m)}}^2 := \left\langle \nabla_{(\theta,m)} \mathcal{F}, T_{(\theta,m)} \nabla_{(\theta,m)} \mathcal{F} \right\rangle,$$

$$\left\| \nabla_{(x,u)} \delta_q \mathcal{F} \right\|_{T_{(x,u)}}^2 := \left\langle \nabla_{(x,u)} \delta_q \mathcal{F}, T_{(x,u)} \nabla_{(x,u)} \delta_q \mathcal{F} \right\rangle,$$

which are readily seen to be non-negative. Finally, define the putative Lyapunov function

$$\mathcal{L}(\theta, m, q) := \mathcal{F}(\theta, m, q) + \left\| \nabla_{(\theta,m)} \mathcal{F} \right\|_{T_{(\theta,m)}}^2 + \left\| \nabla_{(x,u)} \delta_q \mathcal{F} \right\|_{T_{(x,u)}}^2.$$

In Proposition 4.1, we provide sufficient conditions under which $\mathcal{L}(\theta, m, q)$ contracts exponentially along the MPGD flow. The conditions amount to suitable adaptations of classical conditions like the Polyak-Lojasiewicz and Logarithmic Sobolev inequalities to the present setting, along with a smoothness condition on the gradients of the log-likelihood.

**Proposition 4.1** (Exponential Convergence of the Continuous Flow). *Under the $K$-Lipschitz gradient Assumption 2 and the log-Sobolev Assumption 1 with constant $C_\mathcal{E}$, given parameters $(\gamma_x, \gamma_\theta, \eta_x, \eta_\theta)$, if the constants $(\tau_\theta, \tau_{\theta m}, \tau_m, \tau_x, \tau_{xu}, \tau_u)$ satisfy the inequalities in* (34) *for some rate $\varphi$, then it holds that*

$$\dot{\mathcal{L}}_t \leq -\varphi C_\mathcal{E} \mathcal{L}_t.$$

*Moreover, we have exponential convergence of $\mathcal{F}_t$, i.e., $\mathcal{F}_t + \log p_{\theta_*} \leq \mathcal{L}_0 \exp\left(-\varphi C_\mathcal{E} t\right)$.*

For brevity, the assumptions, proof and inequalities are deferred to Appendix F. In Appendix F.3, we demonstrate concrete choices of hyperparameters for which the inequalities of (34) are satisfied.

## 5 DISCRETIZATION

In order to realize an actionable algorithm, we need to discretize the dynamics of Equation (2) both in space and in time. Following the approach of Kuntz et al. (2023), we approximate $q_{t,X} \approx \frac{1}{M} \sum_{i \in [M]} \delta_{X_t^{i,M}} =: q_{t,X}^M$ using a set of particles. This yields the system

$$d\theta_t^M = \eta_\theta m_t^M \, dt \tag{4a}$$

$$dm_t^M = -\gamma_\theta \eta_\theta m_t^M \, dt - \nabla_\theta \mathcal{E}\left(\theta_t^M, q_{t,X}^M\right) \, dt \tag{4b}$$

$$\text{for } i \in [M], \quad dX_t^{i,M} = \eta_x U_t^{i,M} \, dt \tag{4c}$$

$$\text{for } i \in [M], \quad dU_t^{i,M} = -\gamma_x \eta_x U_t^{i,M} \, dt + \nabla_\theta \ell\left(\theta_t^M, X_t^{i,M}\right) \, dt + \sqrt{2\gamma_x} \, dW_t^i. \tag{4d}$$

where $\{W_t^i\}_{i=1}^M$ comprises $M$ independent standard Wiener processes. Proposition 5.1 provides asymptotic justification about using a particle approximation for $q_{t,X}$ to approximate the flow in (2). The proof can be found in the Appendix H.

**Proposition 5.1.** *Under Assumption 2, we have*

$$\lim_{M \to \infty} \mathbb{E} \sup_{t \in [0,T]} \{\|\vartheta_t - \vartheta_t^M\|^2 + W_2^2(q_t^{\otimes M}, q_t^M)\} = 0,$$

*where $\vartheta_t := (\theta_t, m_t)$, $q_t^{\otimes M} := \Pi_{i=1}^M q_t$ with $q_t = \text{Law}(X_t, U_t)$ are defined by the SDE in* (2)*; and $\vartheta_t^M := (\theta_t^M, m_t^M)$, and $q_t^M = \text{Law}(\{(X_t^{i,M}, U_t^{i,M})\}_{i=1}^M)$ in* (4)*.*

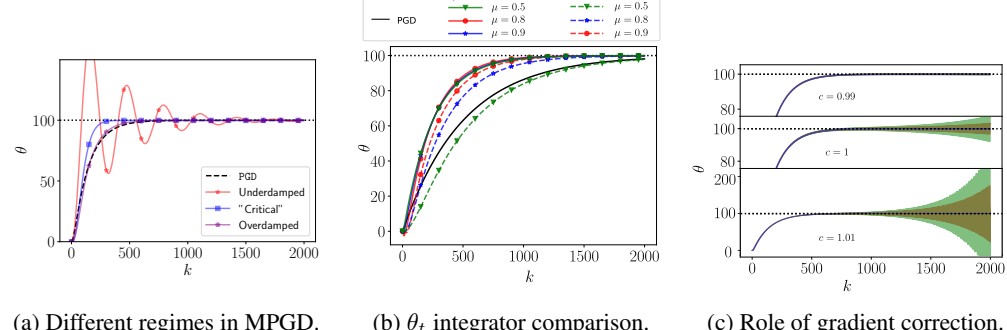

(a) Different regimes in MPGD.  (b) $\theta_t$ integrator comparison.  (c) Role of gradient correction.

Figure 1: **Toy Hierarchical Model**. (a) Different regimes that arise from varying the momentum parameters; (b) Comparison between our Exponential (Exp) integrator and a Nesterov-like integrator for different momentum parameters; (c) we compare the performance of the MPGD with and without gradient correction.

As noted by Ma et al. (2021), naive application of an Euler-Maruyama scheme to momentum-enriched dynamics may be insufficient for attaining an accelerated rate of convergence. We appeal to the literature on the discretization of underdamped Langevin dynamics to design an appropriate integrator. One integrator that can achieve accelerated rates is (Euler) Exponential Integrator (Cheng et al., 2018; Sanz-Serna and Zygalakis, 2021; Hochbruck and Ostermann, 2010). The main strategy is to 'freeze' the nonlinear components of the dynamics in a suitable way, and then solve the resulting linear SDE. We also incorporate a partial update inspired by Sutskever et al. (2013)'s interpretation of NAG. More precisely, given $\left(\theta_0, m_0, \{(X_0^i, U_0^i)\}_{i=1}^M\right)$, define an approximating SDE on the time interval $t \in [0, h]$ as

$$\mathrm{d}\tilde{\theta}_t = \eta_\theta \tilde{m}_t \, \mathrm{d}t \tag{5a}$$

$$\mathrm{d}\tilde{m}_t = -\gamma_\theta \eta_\theta \tilde{m}_t \, \mathrm{d}t - \nabla_\theta \mathcal{E}\left(\bar{\theta}_0, \tilde{q}_{0,X}^M\right) \, \mathrm{d}t \tag{5b}$$

$$\text{for } i \in [M], \quad \mathrm{d}\tilde{X}_t^i = \eta_x \tilde{U}_t^i \, \mathrm{d}t \tag{5c}$$

$$\text{for } i \in [M], \quad \mathrm{d}\tilde{U}_t^i = -\gamma_x \eta_x \tilde{U}_t^i \, \mathrm{d}t + \nabla_x \ell\left(\tilde{\theta}_h, X_0^i\right) \, \mathrm{d}t + \sqrt{2\gamma_x} \, \mathrm{d}W_t^i, \tag{5d}$$

where $\tilde{q}_{t,X}^M := \frac{1}{M} \sum_{i=1}^M \delta_{\tilde{X}_t^i}$, $\bar{\theta}_0 := \theta_0 + \frac{\iota_\theta(h)}{\gamma_\theta} \tilde{m}_0$, and $\iota_\theta(t) := 1 - \exp(-\eta_\theta \gamma_\theta t)$ which (for reasons which will become clear when solving the SDE) can be thought of as a partial update to $\tilde{\theta}_h$. This is a *linear* SDE, which can then be solved analytically, yielding the following iteration (see Appendix I.4 for details):

$$\tilde{\theta}_k = \tilde{\theta}_{k-1} + \frac{\iota_\theta(h)}{\gamma_\theta} \tilde{m}_{k-1} - \frac{1}{\gamma_\theta}\left(h - \frac{\iota_\theta(h)}{\gamma_\theta \eta_\theta}\right) \nabla_\theta \mathcal{E}\left(\bar{\theta}_{k-1}, \tilde{q}_{k-1,X}^M\right)$$

$$\tilde{m}_k = (1 - \iota_\theta(h))\tilde{m}_{k-1} - \frac{\iota_\theta(h)}{\gamma_\theta \eta_\theta} \nabla_\theta \mathcal{E}\left(\bar{\theta}_{k-1}, \tilde{q}_{k-1,X}^M\right)$$

$$\text{for } i \in [M], \quad \tilde{X}_k^i = \tilde{X}_{k-1}^i + \frac{\iota_x(h)}{\gamma_x} \tilde{U}_{k-1}^i + \frac{1}{\gamma_x}\left(h - \frac{\iota_x(h)}{\gamma_x \eta_x}\right) \nabla_x \ell\left(\tilde{\theta}_k, \tilde{X}_{k-1}^i\right) + L_\Sigma^{XX} \xi_k^i$$

$$\text{for } i \in [M], \quad \tilde{U}_k^i = (1 - \iota_x(h))\tilde{U}_{k-1}^i + \frac{\iota_x(h)}{\gamma_x \eta_x} \nabla_x \ell\left(\tilde{\theta}_k, \tilde{X}_{k-1}^i\right) + L_\Sigma^{XU} \xi_k^i + L_\Sigma^{UU} \xi_k'^{,i},$$

where $\iota_x(t) := 1 - \exp(-\eta_x \gamma_x t)$, $\left\{\xi_k^i, \xi_k'^{,i}\right\}_{i \in [M]} \overset{\text{i.i.d.}}{\sim} \mathcal{N}(0_{d_x}, I_{d_x})$, $\bar{\theta}_{k-1} := \tilde{\theta}_{k-1} + \frac{\iota_\theta(h)}{\gamma_\theta} \tilde{m}_{k-1}$, and $\left(L_\Sigma^{XX}, L_\Sigma^{XU}, L_\Sigma^{UU}\right)$ are suitable scalars that depend on $(\eta_x, \gamma_x)$, and which can be found in the Appendix Equation (64).

The astute reader will notice that our approximating ODE-SDE (i.e., Equation (5)) deviates from Cheng et al. (2018)'s style of discretization. Specifically, the difference lies in where we compute the gradient in Equations (5b) and (5d) which we refer to as *gradient correction*. This subtle choice reflects the decision to utilize partially and fully updated approximations to $\tilde{\theta}_h$, i.e., the partially

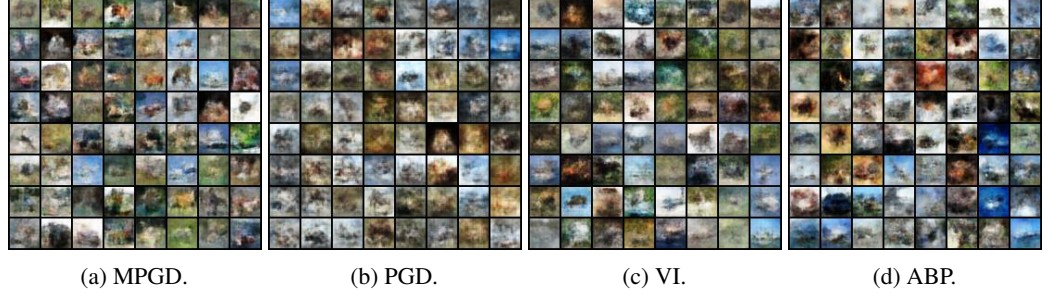

| (a) MPGD. | (b) PGD. | (c) VI. | (d) ABP. |

Figure 2: **CIFAR-10**. Samples generated from various algorithms.

updated $\bar{\theta}_0$ in Equation (5b) and fully updated $\tilde{\theta}_h$ in Equation (5d). This is reminiscent of NAG's use of a partial parameter update to compute the gradient, unlike Polyak's Heavy Ball, which does not (Sutskever et al., 2013, Section 2.1). In our case, we empirically found that this choice is critical for a more stable discretization and, as a result, enables the algorithm to take larger step sizes to allow the algorithm to travel "faster" (see Toy HMM in Section 6).

## 6 EXPERIMENT

### 6.1 TOY HIERARCHICAL MODEL

As a toy example, we consider the hierarchical model proposed in Kuntz et al. (2023). The model is given by $p_\theta(y, x) := \prod_{i=1}^{N} \mathcal{N}(y_i|x_i, 1)\mathcal{N}(x_i|\theta, 1)$, where $N = 100$ is the number of data points. The dataset $y$ is sampled from a model with $\theta = 100$. In this experiment, we wish to understand the behaviour of MPGD compared against PGD. We are particularly interested in (1) how the momentum parameters $(\eta_\theta, \gamma_\theta, \eta_x, \gamma_x)$ affect the optimization process; (2) comparing our proposed discretization to another that follows in the style of NAG (detailed in Appendix I.3); and (3) the role of the gradient correction term described in Section 5. The results are shown in Figure 1. Further experiment details can be found in Appendix J.4.1.

In Figure 1a, we show that different regimes can arise from different choices of hyper-parameters (as discussed in Section 3). In Figure 1b, we compare the performance of MPGD using (our) Exponential integrator with a NAG-like discretization for $(\theta_t, m_t)$-component. We vary the "momentum coefficient", i.e., we have $\mu \in \{0.5, 0.8, 0.9\}$ with $\mu_\theta = \mu_x = \mu$. It can be seen that MPGD with our exponential integrator for $\theta_t$ performs better than NAG-like integrator (see Appendix J.3 for more discussion). In Figure 1c, we show the effect of the gradient correction term for three different step sizes in $(\theta, m)$-components while keeping the step sizes in $(X, U)$ fixed. The different lines in the figure are generated from a step size of $c \cdot 10^{-5/2}$ where $c \in \{0.99, 1, 1.01\}$. It can be seen that our proposed method is more effective than the other discretization. In red, we show MPGD with the absence of gradient correction in Equation (5b) (i.e., when we use $\theta_0$ instead of $\bar{\theta}_0$ in Equation (5b)), and, in green, we show the MPGD when the gradient correction is absent in both Equation (5b) and Equation (5d) (i.e. when we use $\theta_0$ instead of $\bar{\theta}_0$ and $\tilde{\theta}_h$ in Equation (5b) and Equation (5d)). It can be seen that usage of the gradient correction term results in a more stable algorithm.

### 6.2 IMAGE GENERATION

For this task, we consider two datasets: MNIST (LeCun et al., 1998) and CIFAR-10 (Krizhevsky and Hinton, 2009). For the model, we use a variational autoencoder (Kingma and Welling, 2014) with a VampPrior (Tomczak and Welling, 2018). As baselines, we compare our proposed method MPGD against PGD, Alternating Backpropagation (ABP) (Han et al., 2017), Short Run (SR) (Nijkamp et al., 2020), and amortized variational inference (VI) (Kingma and Welling, 2014). ABP is the most similar to PGD. They differ in that ABP takes multiple steps of the (unadjusted and overdamped) Langevin algorithm (ULA) instead of PGD's single step. They (MPGD, PGD, ABP) are also *persistent*, meaning that the ULA chain starts at the previous particle location, whereas SR restarts the chain

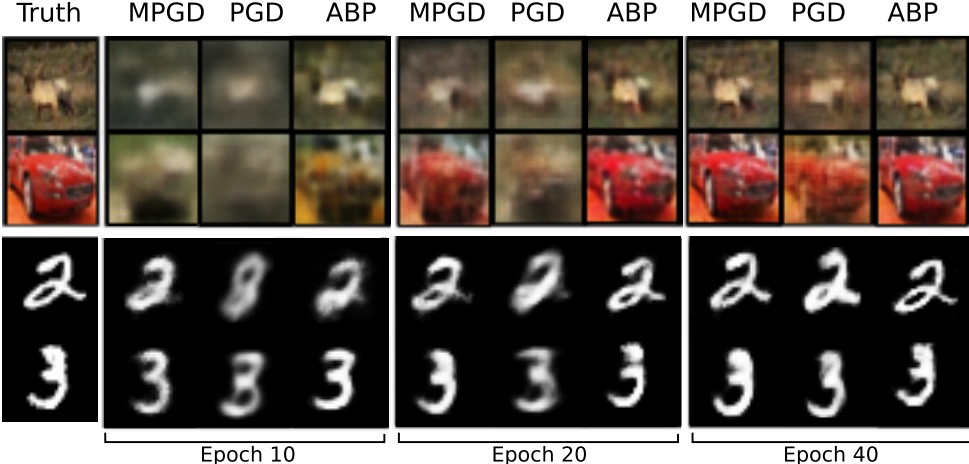

| Truth | MPGD | PGD | ABP | MPGD | PGD | ABP | MPGD | PGD | ABP |

Figure 3: **Posterior Cloud vs Epochs**. We show the evolution of the reconstruction of a particle for *persistent* methods. The particle is taken at epoch $\{10, 20, 40\}$ on MNIST and CIFAR-10.

| Method | MPGD | PGD | ABP | SR | VI |
|---|---|---|---|---|---|
| MNIST | **48.7** $\pm$ 1.5 | 102.4 $\pm$ 1.1 | **53.9** $\pm$ 13 | 134.34 $\pm$ 1.6 | 72.6 $\pm$ 1.5 |
| CIFAR | **93.2** $\pm$ 1.9 | 104.2 $\pm$ 5.4 | 96.9 $\pm$ 3.1 | 140.6 $\pm$ 16.3 | **93.4** $\pm$ 0.9 |

Table 1: **FID scores** on MNIST and CIFAR-10 after the final epoch. In **bold**, we indicate the lowest two scores on average. We write $(\mu \pm \sigma)$ to indicate the mean $\mu$ and standard deviation $\sigma$ of the FID score calculated over three independent trials.

at a random location sampled from the prior but like ABP runs the chain for several steps. Further experiment details can be found in Appendix J.4.2.

We are interested in the generative performance of the resulting models. For a qualitative measure, we show the samples produced by the model in Figure 2 for CIFAR (for MNIST samples, see Figure 5 in the Appendix). As a quantitative measure, we report the Fréchet inception distance (FID) (Heusel et al., 2017) as shown in Table 1. It can be seen that our proposed method does well compared against other baselines. The closest competitor is ABP. As noted by Kuntz et al. (2023), ABP can reap the benefits of taking multiple ULA steps to locate the posterior mode quickly and reduce the transient phase. This hypothesis is further confirmed in Figure 3, where we visualize the evolution of a single particle across various epochs. It can be seen that ABP's and MPGD's distinct methods of reducing the transient phase (by taking multiple steps or utilizing momentum) are effective. As a competitor SR does perform badly, which we attribute to its non-persistent property: by restarting the particles at the prior and running short chains, it is unable to overcome the bias of short chains and locate the posterior mode. Variational inference performs well for CIFAR but surprisingly falls slightly short in MNIST. We hypothesise that this is due to the choice of parametrization of the variational distribution.

## 7 CONCLUSION AND FUTURE WORK

We presented momentum PGD, a method of incorporating momentum into particle gradient descent. We establish the convergence in continuous time, and through experiments, we showed that, for suitably chosen momentum parameters, the resulting algorithm achieves better performance than PGD. Future work includes: a systematic method of tuning momentum parameters $(\eta_\theta, \gamma_\theta, \eta_x, \gamma_x)$ (or some justification for using the momentum coefficient heuristic), theoretical characterization of the difference between our discretization scheme compared with other potential schemes akin to Sanz-Serna and Zygalakis (2021).

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

# Appendices

## A  NOTATION

The following table summarizes some key notation used throughout.

| | |
|---|---|
| $\mathcal{E}$ | Free Energy. |
| $\mathcal{F}$ | Momentum-enriched Free Energy |
| $z_t$ | The tuple $(\theta_t, m_t, q_t) \in \mathbb{R}^{d_\theta} \times \mathbb{R}^{d_\theta} \times \mathcal{P}(\mathbb{R}^{d_x} \times \mathbb{R}^{d_x})$ |
| $\nabla_a f$ | Euclidean Gradient of $f$ w.r.t. $a$ |
| $\nabla_{(a,b)} f$ | $[\nabla_a f, \nabla_b f]^\top$ |
| $\ell$ | $\ell(\theta, x) := \log p_\theta(y, x)$ |
| $\rho_\theta$ | $\rho_\theta(x) := p_\theta(y, x)$ |
| $\nabla_a \cdot$ | Divergence operator w.r.t. $a$ |
| $\nabla_a^*$ | Adjoint operator of $\nabla_a$ (see Appendix F.4.2). |
| $[n]$ | $[n] := \{1, ..., n\}$ |
| $\Delta$ | Laplacian operator, $\Delta = \nabla \cdot \nabla$ |
| $\mathcal{P}(\mathbb{R}^d)$ | The space of probability measures that are absolutely continuous w.r.t. Lebesgue measure (have densities) and possess finite second moments. |
| $\langle \cdot, \cdot \rangle$ (and $\|\cdot\|$) | Euclidean inner product or Frobenius inner product (and it's inner norm) |
| $\langle \cdot, \cdot \rangle_\rho$ (and $\|\cdot\|_\rho$) | $L^2(\rho)$ inner product (and its norm) |
| $o(\epsilon)$ | Bachman-Landau little-o notation |
| $\partial F(\mu)$ | Fréchet subdifferential (see Definition C.1) |

## B  RELATED WORK

The present work sits at the juncture of i) deterministic gradient flows for optimising objectives over "parameter" spaces, typically expressible through the discretization of ODEs, and ii) stochastic gradient flows for optimising objectives over the space of probability measures, typically expressible through discretisation of mean-field SDEs. The former class of problems is too vast to be properly surveyed here, effectively including a large proportion of modern continuous optimisation problems. The latter class has seen substantial growth over the past few years in particular, with various problems related to sampling (Liu, 2017; Bernton, 2018; Garbuno-Inigo et al., 2020; Duncan et al., 2023), variational inference (Yao and Yang, 2022; Lambert et al., 2022; Diao et al., 2023), and the training of shallow neural networks (Mei et al., 2018; Chizat and Bach, 2018; Nitanda and Suzuki, 2017; Hu et al., 2021; Nitanda et al., 2022; Chizat, 2022; Chen et al., 2022; Suzuki et al., 2023) being studied in this framework. While there exist earlier works which combine optimisation with Markovian sampling (e.g. stochastic approximation approaches to the EM algorithm (Delyon et al., 1999; De Bortoli et al., 2021), training of energy-based models (Hinton, 2002), and hyperparameter tuning in MCMC (Andrieu and Thoms, 2008; Roberts and Rosenthal, 2009)), the connection to gradient flows remains somewhat under-developed at present. We hope that the present work can encourage further exploration of these connections.

## C  GRADIENT FLOW ON $\mathbb{R}^{d_\theta} \times \mathcal{P}(\mathcal{X})$

In this section, we will describe gradient flows on the extended space $\mathbb{R}^{d_\theta} \times \mathcal{P}(\mathcal{X})$ where $\mathcal{X} = \mathbb{R}^{d_x}$. We begin with an exposition of gradient flows in Wasserstein space, i.e. the space of distributions $\mathcal{P}(\mathcal{X})$ endowed with the Wasserstein-2 metric. We aim to give an intuitive introduction as opposed to a rigorous one; those readers with an interest in the latter are directed to Ambrosio et al. (2005). Using the ideas in Ambrosio et al. (2005), we show how the notion of gradients can be generalized to the product space $\mathbb{R}^{d_\theta} \times \mathcal{P}(\mathcal{X})$.

## C.1   GRADIENT FLOW ON $\mathcal{P}(\mathcal{X})$

This exposition aims to introduce the intuition and necessary objects. For a rigorous treatment, we point the reader to (Ambrosio et al., 2005). We begin by describing the gradient flow on $\mathcal{P}(\mathcal{X})$ endowed with the Wasserstein-2 metric. The Wasserstein distance is defined as

$$W_2^2(p,q) = \inf_{\pi \in \Pi(p,q)} \int_{\mathbb{R}^d \times \mathbb{R}^d} \|x - y\|^2 \, \pi(\mathrm{d}x \times \mathrm{d}y),$$

where $\Pi(p,q)$ is the set of all couplings between $p$ and $q$.

In Ambrosio et al. (2005, Chapter 11), the authors discuss various approaches for adapting gradient flows on well studied spaces (such as Euclidean and Riemannian spaces) to the Wasserstein space. One of these approaches proceeds by first defining suitable notions of tangent space and subdifferential, following which the simple definition of gradient flow modelled on Riemannian manifolds can then reproduced. In this case, the Fréchet subdifferential (Ambrosio et al., 2005, Definition 10.1.1) is defined as follow:

**Definition C.1.** *[Fréchet differential on Wasserstein Space] Let $F : \mathcal{P}(\mathcal{X}) \to \mathbb{R}$ be a sufficiently regular function. We say that $\xi \in L^2(q)$ belongs to the Fréchet subdifferential $\partial F(q)$ if for all $q' \in \mathcal{P}(\mathcal{X})$, we have*

$$F(q') - F(q) \geq \left\langle \xi, t_q^{q'} - i \right\rangle_q + o(W_2(q, q')),$$

*where $t_p^q$ is the optimal map between $p$ and $q$ (Ambrosio et al., 2005, see (7.1.4)) and $i$ is the identity map. Furthermore, if $\xi \in \partial F(q)$ also satisfies*

$$F(t_\# q) - F(q) \geq \langle \xi, t - i \rangle_q + o(\|t - i\|_q),$$

*for all $t \in L^2(q)$, then we say that $\xi$ is a **strong** subdifferential.*

See Ambrosio et al. (2005, Definition 10.1.1) for more details. The strong subdifferential can be thought of as the (Wasserstein) "gradient" of $F$. Equipped with this notion of gradient, we can define the gradient (descent) flow of $F : \mathcal{P}(\mathcal{X}) \to \mathbb{R}$ as follows:

**Definition C.2** (Gradient Flow). *We say that a curve $p_t : [0,1] \to \mathcal{P}(\mathcal{X})$ is a gradient flow of $F : \mathcal{P}(\mathcal{X}) \to \mathbb{R}$ if for all $t > 0$, it satisfies the continuity equation $\partial_t p_t + \nabla_x \cdot (v_t p_t) = 0$, where the tangent vector $v_t$ satisfies $-v_t \in \partial F(p_t)$ for all $t$.*

Thus, for our application, we are interested in computing the strong subdifferential of $F$. If $F$ is an integral of the type

$$F(p) = \int f(x, p(x), \nabla_x p(x)) \mathrm{d}x, \tag{6}$$

where $f : \mathcal{X} \times \mathbb{R} \times \mathcal{X} \to \mathbb{R}$ is sufficiently regular, then we will see that its strong subdifferential admits a analytic solution.

Functionals of this form of great interest in the calculus of variations (e.g., see Gelfand and Silverman (2000)). A vital quantity which is used to study these functionals is the first variation. Writing $\delta_p F[p] : \mathcal{X} \to \mathbb{R}$ for the first variation of $F$, the unique up to constants function such that

$$\left. \frac{\mathrm{d}}{\mathrm{d}\epsilon} F(q_\epsilon) \right|_{\epsilon=0} = - \int \delta_q F[q](x) \, \nabla_x \cdot (q(x) v(x)) \mathrm{d}x$$

for all $v$ such that $q_\epsilon := (i + \epsilon v)_\# q \in \mathcal{P}(\mathcal{X})$ for sufficiently small $\epsilon$, one can readily establish that for (6)-typed $F$, it is given by

$$\delta_p F[p](x) := \nabla_{(2)} f(x, p(x), \nabla_x p(x)) - \nabla_x \cdot (\nabla_{(3)} f(x, p(x), \nabla_x p(x))),$$

where $\nabla_{(i)}$ denotes the partial derivative w.r.t. the $i$-th argument (Ambrosio et al., 2005, (10.4.2)).

It can be shown that for any $\xi \in \partial F(p)$ which is a strong subdifferential, it holds that $\xi(x) \overset{p-a.e.}{=} \nabla_x \delta_p F[p](x)$ (Ambrosio et al., 2005, Lemma 10.4.1).

## C.2  Gradient flow on $\mathbb{R}^{d_\theta} \times \mathcal{P}(\mathbb{R}^{d_x})$

We provided here a generalization of the Fréchet differential (for a broad survey of Fréchet differentials, see Kruger (2003)) to the extended space $\mathbb{R}^{d_\theta} \times \mathcal{P}(\mathbb{R}^{d_x})$:

**Definition C.3** (Fréchet differential on $\mathbb{R}^{d_\theta} \times \mathcal{P}(\mathbb{R}^{d_x})$). *Let $\mathcal{F} : \mathbb{R}^{d_\theta} \times \mathcal{P}(\mathcal{X}) \to \mathbb{R}$ be a sufficiently regular function. We say that $(\xi_\theta, \xi_q) \in \mathbb{R}^{d_\theta} \times L_2(q)$ belongs to the Fréchet subdifferential $\partial \mathcal{F}(\theta, q)$ if for all $(\theta', q') \in \mathbb{R}^{d_\theta} \times \mathcal{P}(\mathbb{R}^{d_x})$, it holds that*

$$\mathcal{F}(\theta', q') - \mathcal{F}(\theta, q) \geq \langle \xi_\theta, \theta' - \theta \rangle + \left\langle \xi_q, t_q^{q'} - i \right\rangle_q + o(W_2(q', q) + \|\theta' - \theta\|).$$

*Furthermore, if $(\xi_\theta, \xi_q) \in \partial \mathcal{F}(\theta, q)$ also satisfies*

$$\mathcal{F}(\theta + \tau, t_\# q) - \mathcal{F}(\theta, q) \geq \langle \xi_\theta, \tau \rangle + \langle \xi_q, t - i \rangle_q + o(\|t - i\|_q + \|\tau\|),$$

*for all $(\tau, t) \in \mathbb{R}^{d_\theta} \times L_2(q)$, then we say that $(\xi_\theta, \xi_q)$ is a **strong** subdifferential.*

For the remainder of the section, we shall assume that for any perturbation $(\sigma, v)$, there exists some $\delta_q F[\theta, q] : \mathcal{X} \to \mathbb{R}$ (often called the first variation) such that the following expansion holds:

$$\frac{\mathrm{d}}{\mathrm{d}\epsilon} \mathcal{F}(\theta_\epsilon, q_\epsilon) \big|_{\epsilon=0} = \langle \nabla_\theta \mathcal{F}(\theta, q), \sigma \rangle - \int \delta_q \mathcal{F}[\theta, q](x) \nabla_x \cdot (v(x) q(x)) \, \mathrm{d}x, \tag{7}$$

where $q_\epsilon := (i + \epsilon v)_\# q$ and $\theta_\epsilon := \theta + \epsilon \sigma$. For all $\mathcal{F}$ of interest in this paper, this assumption holds; see for instance, see Proposition F.4.

We follow in the argument of Ambrosio et al. (2005, Lemma 10.4.1) to show that if $(\xi_\theta, \xi_q) \in \partial \mathcal{F}(\theta, q)$ is a strong subdifferential, then we have that $\xi_\theta = \nabla_\theta \mathcal{F}(\theta, q)$ and $\xi_q(x) \overset{q-a.e.}{=} \nabla_x \delta_q \mathcal{F}[\theta, q](x)$. Let $(\xi_\theta, \xi_p) \in \partial \mathcal{F}(\theta, p)$ be a strong subdifferential. By using the strong subdifferential property for some perturbation $(\epsilon \sigma, \epsilon v)$ and taking left and right limits of (C.3), we obtain that

$$\limsup_{\epsilon \uparrow 0} \frac{\mathcal{F}(\theta_\epsilon, q_\epsilon) - \mathcal{F}(\theta, q)}{\epsilon} \leq \langle \xi_\theta, \sigma \rangle + \langle \xi_q, v \rangle_q \leq \liminf_{\epsilon \downarrow 0} \frac{\mathcal{F}(\theta_\epsilon, q_\epsilon) - \mathcal{F}(\theta, q)}{\epsilon}.$$

On the other hand, from our assumption (7), we have that

$$\lim_{\epsilon \to 0} \frac{\mathcal{F}(\theta_\epsilon, q_\epsilon) - \mathcal{F}(\theta, q)}{\epsilon} = \langle \nabla_\theta \mathcal{F}(\theta, q), \sigma \rangle - \int \delta_q \mathcal{F}[\theta, q](x) \nabla_x \cdot (v(x) q(x)) \, \mathrm{d}x$$

$$= \langle \nabla_\theta \mathcal{F}(\theta, q), \sigma \rangle + \langle \nabla_x \delta_q \mathcal{F}[\theta, q], v \rangle_q,$$

where the last inequality follows from integration by parts and the divergence theorem. Hence, we have that $\xi_\theta = \nabla_\theta \mathcal{F}(\theta, q)$, and $\xi_q \overset{q-a.e.}{=} \nabla_x \delta_q \mathcal{F}[\theta, q]$.

## C.3  First Variation

In this section, we derive the first variation for integral expressions taking a certain form. In particular, we are interested in computing the strong subdifferential of $\mathcal{F}$ of the following types:

$$\mathcal{F}(\theta, q) := \int f(\theta, x, q(x), \nabla_x q(x)) \mathrm{d}x, \tag{VI-I}$$

$$\mathcal{F}(\theta, q) := \int \int f(\theta, x, y, q(x), q(y)) \mathrm{d}x \mathrm{d}y. \tag{VI-II}$$

An exampled of (VI-I)-typed $\mathcal{F}$ is when $\mathcal{F}$ is the free energy $\mathcal{E}$. Following standard techniques from the calculus of variations (Gelfand and Silverman, 2000), consider a perturbation $(\sigma, v) \in \mathbb{R}^{d_\theta} \times L_2(q)$ and define a mapping $\Phi$ by

$$\Phi(\epsilon) := \mathcal{F}(\theta_\epsilon, q_\epsilon)$$

where $q_\epsilon := (i + \epsilon v)_\# q$ and $\theta_\epsilon := \theta + \epsilon \sigma$.

(VI-I)-**typed** $\mathcal{F}$. By application of the change-of-variables formula (see for instance, see Ambrosio et al. (2005, Lemma 5.5.3)), we can compute the density $q_\epsilon$ and its derivative $\frac{\partial}{\partial \epsilon} q_\epsilon(y)$ as follows:

$$q_\epsilon(y) = \frac{q}{\det(I + \epsilon \nabla_x v)} \circ (i + \epsilon v)^{-1}(y), \quad \frac{\partial}{\partial \epsilon} q_\epsilon(y) \big|_{\epsilon=0} = -\nabla_x \cdot [qv](y).$$

Thus, we can compute the derivative of $\Phi$, provided that the interchange of derivative and integral can be justified, as

$$
\begin{aligned}
\frac{d}{d\epsilon}\Phi(\epsilon)\Big|_{\epsilon=0} &= \int \frac{d}{d\epsilon} f(\theta_\epsilon, x, q_\epsilon(x), \nabla_x q_\epsilon(x))\,\mathrm{d}x \\
&= \int \langle \nabla_\theta f(\theta, x, q(x), \nabla_x q(x)), \sigma \rangle\,\mathrm{d}x \\
&\quad - \int \nabla_{(3)} f(\theta, x, q(x), \nabla_x q(x))\, \nabla_x \cdot [vq](x)\,\mathrm{d}x \\
&\quad - \int \langle \nabla_{(4)} f(\theta, x, q(x), \nabla_x q(x)), \nabla_x[\nabla_x \cdot [vq]](x) \rangle\,\mathrm{d}x
\end{aligned}
$$

Applying integration by parts and the divergence theorem, we obtain that

$$
\begin{aligned}
\frac{d}{d\epsilon}\Phi(\epsilon)\Big|_{\epsilon=0} &= \langle \nabla_\theta \mathcal{F}(\theta, x, q(x), \nabla_x q(x)), \sigma \rangle \\
&\quad - \int \nabla_{(3)} f(\theta, x, q(x), \nabla_x q(x))\, [\nabla_x \cdot (vq)](x)\,\mathrm{d}x \\
&\quad + \int \nabla_x \cdot [\nabla_{(4)} f](\theta, x, q(x), \nabla_x q(x))[\nabla_x \cdot (vq)](x)\,\mathrm{d}x
\end{aligned}
$$

We can hence write the first variation of (VI-I)-typed $\mathcal{F}$ as

$$
\delta_q \mathcal{F}[\theta, q](x) = \nabla_{(3)} f(\theta, x, q(x), \nabla_x q(x)) - \nabla_x \cdot (\nabla_{(4)} f(\theta, x, q(x), \nabla_x q(x))). \tag{8}
$$

(VI-II)-**typed** $\mathcal{F}$. Similarly to above, we can define $\Phi(\epsilon) := \mathcal{F}(\theta_\epsilon, q_\epsilon)$, whose derivative is given by

$$
\begin{aligned}
\frac{d}{d\epsilon}\Phi(\epsilon)\Big|_{\epsilon=0} &= \langle \nabla_\theta \mathcal{F}(\theta, q), \sigma \rangle \\
&\quad - \int\int \nabla_{(4)} f(\theta, x, y, q(x), q(y))\, \nabla_x \cdot [qv](x)\,\mathrm{d}x\,\mathrm{d}y \\
&\quad - \int\int \nabla_{(5)} f(\theta, x, y, q(x), q(y))\, \nabla_y \cdot [qv](y)\,\mathrm{d}x\,\mathrm{d}y.
\end{aligned}
$$

Hence, the first variation is given by

$$
\delta_q \mathcal{F}[\theta, q](x) = \int \left[ \nabla_{(4)} f(\theta, x, y, q(x), q(y)) + \nabla_{(5)} f(\theta, y, x, q(y), q(x)) \right]\,\mathrm{d}y. \tag{9}
$$

**Example 1** (First Variation of $\mathcal{E}$). *It can be seen that the free energy is* (VI-I)-*typed with* $f(\theta, x, a, g) = a \cdot \log \frac{a}{p_\theta(y,x)}$. *Hence, by combining the formula* (8) *with the fact that*

$$
\nabla_3 f(\theta, x, a, b) = \log \frac{a}{p_\theta} + 1
$$
$$
\nabla_4(f(\theta, x, a, g) = 0,
$$

*we obtain the following expression for the first variation:*

$$
\delta_q \mathcal{F}[\theta, q](x) = \log \frac{q(x)}{p_\theta(y, x)} + 1.
$$

*This provides an alternative derivation of this expression to that given in Kuntz et al. (2023, Lemma 1).*

## D  A LOG SOBOLEV INEQUALITY FOR $\mathcal{E}$ CAN BE TRANSFERRED TO $\mathcal{F}$

In this section, we show that nice properties of the functional $\mathcal{E}$ transfer to the functional $\mathcal{F}$, i.e. if $\mathcal{E}$ satisfies a log Sobolev inequality (Assumption 1), then so does $\mathcal{F}$ (with a modified constant):

**Proposition D.1.** *If Assumption 1 holds, then upon defining $C := \min\{C_{\mathcal{E}}, \eta_\theta, \eta_x\}$, it holds that*

$$\mathcal{F}(\theta, m, q) + \log p_{\theta^*}(y) \leq \frac{1}{2C} \|\nabla \mathcal{F}(\theta, m, q)\|_q^2, \quad \forall \theta, m \in \mathbb{R}^{d_\theta}, \ q \in \mathcal{P}(\mathbb{R}^{2d_x}),$$

*where*

$$\|\nabla \mathcal{F}(\theta, m, q)\|_q^2 := \|\nabla_{(\theta, m)} \mathcal{F}(\theta, m, q)\|^2 + \|\nabla_{(x,u)} \log q - \nabla_{(x,u)} \log \rho_{\theta, \eta_x}\|_q^2. \tag{10}$$

*Proof.* As we show at the end of the proof, we have the fact that

$$\left\| \nabla_{(x,u)} \log \left( \frac{q}{\rho_\theta \otimes r_{\eta_x}} \right) \right\|_q^2 \geq \left\| \nabla_x \log \left( \frac{q_X}{\rho_\theta} \right) \right\|_{q(\mathrm{d}x)}^2 + \left\| \nabla_u \log \frac{q_{U|X}}{r_{\eta_x}} \right\|_q^2. \tag{11}$$

From the definition, we have

$$\|\nabla \mathcal{F}(\theta, m, q)\|_q^2 = \|\nabla_\theta \mathcal{F}(\theta, m, q)\|^2 + \|\eta_\theta m\|^2 + \left\| \nabla_{(x,u)} \log \frac{q}{\rho_{\theta, \eta_x}} \right\|_q^2.$$

Using (11) and the fact that

$$\nabla_\theta \mathcal{F}(\theta, m, q) = -\int \nabla_\theta \log p_\theta(y, x, u) q(\mathrm{d}x, \mathrm{d}u) = -\int \nabla_\theta \log p_\theta(y, x) q(\mathrm{d}x) = \nabla_\theta \mathcal{E}(\theta, q),$$

then it follows that

$$\|\nabla \mathcal{F}(\theta, m, q)\|_q^2 \geq \|\nabla_\theta \mathcal{E}(\theta, q_X)\|_q^2 + \eta_\theta^2 \|m\|^2 + \left\| \nabla_u \log \frac{q_{U|X}}{r_\eta} \right\|_q^2.$$

Because $C = \min\{C_{\mathcal{E}}, \eta_\theta, \eta_x\}$, the above implies that

$$\frac{1}{2C} \|\nabla \mathcal{F}(\theta, m, q)\|_q^2 \geq \frac{1}{2C_{\mathcal{E}}} \|\nabla_\theta \mathcal{E}(\theta, q)\|_q^2 + \frac{\eta_\theta}{2} \|m\|^2 + \frac{1}{2\eta_x} \left\| \nabla_u \log \frac{q_{U|X}}{r_{\eta_x}} \right\|_q^2. \tag{12}$$

Since $r_{\eta_x} = \mathcal{N}(0, \eta_x^{-1} I_{d_x})$ is $\eta_x$-strongly log-concave, by the Bakry-Émery criterion of Bakry and Émery (2006), it holds that

$$\frac{1}{2\eta_x} \left\| \nabla_u \log \left( \frac{q(\cdot|x)}{r_{\eta_x}} \right) \right\|_{q(\mathrm{d}u|x)}^2 \geq \mathsf{KL}(q_{U|x} | r_{\eta_x}) \quad \forall x \in \mathcal{X},$$

and so we obtain that

$$\frac{1}{2\eta_x} \left\| \nabla_u \log \frac{q_{U|X}}{r_{\eta_x}} \right\|_q^2 = \frac{1}{2\eta_x} \mathbb{E}_{q_X} \left[ \left\| \nabla_u \log \frac{q(\cdot|X)}{r_{\eta_x}} \right\|_{q(\mathrm{d}u|\cdot)}^2 \right]$$
$$\geq \mathbb{E}_{q(\mathrm{d}x)} \left[ \mathsf{KL}(q_{U|x} | r_{\eta_x}) \right].$$

Plugging the above into (12) and using the log Sobolev inequality for $\mathcal{E}$, we obtain that for $\mathcal{F}$:

$$\frac{1}{2C} \|\nabla \mathcal{F}(\theta, m, q)\|_q^2 \geq \mathcal{E}(\theta, q_X) + \log p_{\theta^*}(y) + \frac{\eta_\theta}{2} \|m\|_2^2 + \mathbb{E}_{q(\mathrm{d}x)} \left[ \mathsf{KL}(q_{U|X} | r_{\eta_x}) \right]$$
$$\geq \mathcal{F}(\theta, m, q) + \log p_{\theta^*}(y).$$

We have one loose end to tie up: proving (11). To do so, note that

$$\left\| \nabla_{(x,u)} \log \left( \frac{q}{\rho_\theta \otimes r_{\eta_x}} \right) \right\|_q^2 = \left\| \nabla_{(x,u)} \log \left( \frac{q_X}{\rho_\theta} \right) + \nabla_{(x,u)} \log \left( \frac{q_{U|X}}{r_{\eta_x}} \right) \right\|_q^2$$

$$= \left\| \nabla_x \log \left( \frac{q_X}{\rho_\theta} \right) + \nabla_x \log q_{U|X} \right\|_q^2 \tag{13}$$

$$+ \left\| \nabla_u \log \left( \frac{q_{U|X}}{r_{\eta_x}} \right) \right\|_q^2. \tag{14}$$

Taking the first term and expanding the square, we have

$$\left\|\nabla_x \log\left(\frac{q_X}{\rho_\theta}\right) + \nabla_x \log q_{U|X}\right\|_q^2 = \left\|\nabla_x \log\left(\frac{q_X}{\rho_\theta}\right)\right\|_q^2$$
$$+ 2\left\langle \nabla_x \log\left(\frac{q_X}{\rho_\theta}\right), \nabla_x \log q_{U|X}\right\rangle_q$$
$$+ \left\|\nabla_x \log q_{U|X}\right\|_q^2.$$

Applying Jensen's inequality to the final term, we obtain

$$\left\|\nabla_x \log\left(\frac{q_X}{\rho_\theta}\right) + \nabla_x \log q_{U|X}\right\|_q^2 \geq \left\|\nabla_x \log\left(\frac{q_X}{\rho_\theta}\right)\right\|_{q(\mathrm{d}x)}^2$$
$$+ 2\left\langle \nabla_x \log\left(\frac{q_X}{\rho_\theta}\right), \mathbb{E}_{q(\mathrm{d}u|\cdot)}\left[\nabla_x \log q(U|\cdot)\right]\right\rangle_{q(\mathrm{d}x)}$$
$$+ \left\|\mathbb{E}_{q(\mathrm{d}u|\cdot)}\left[\nabla_x \log q(U|\cdot)\right]\right\|_{q(\mathrm{d}x)}^2.$$

One can compute explicitly that for all $x$, it holds that $\mathbb{E}_{q(\mathrm{d}u|x)}\left[\nabla_x \log q(u|X)\right] = 0$; we thus obtain that

$$\left\|\nabla_x \log\frac{q_X}{\rho_\theta} + \nabla_x \log q_{U|X}\right\|_q^2 \geq \left\|\nabla_x \log\frac{q_X}{\rho_\theta}\right\|_{q(\mathrm{d}x)}^2,$$

and so (11) follows from (14). $\qquad\square$

## E $\quad \mathcal{F}(z_t)$ IS NON-INCREASING

In the following proposition, we show that $\mathcal{F}_t := \mathcal{F}(z_t)$ is non-increasing in time.

**Proposition E.1.** *For any $\gamma_x \geq 0$ and $\gamma_\theta \geq 0$, it holds that*

$$\dot{\mathcal{F}}_t = -\gamma_\theta \|\nabla_m \mathcal{F}_t\|^2 - \gamma_x \|\nabla_u \delta_q \mathcal{F}_t\|_{q_t}^2 \leq 0.$$

*Proof.* We begin by computing the time derivative

$$\dot{\mathcal{F}}_t = -\left\langle \nabla_{(\theta,m)}\mathcal{F}_t, \begin{pmatrix} 0 & -I_{d_\theta} \\ I_{d_\theta} & \gamma_\theta I_{d_\theta} \end{pmatrix} \nabla_{(\theta,m)}\mathcal{F}_t \right\rangle \tag{15}$$

$$- \left\langle \nabla_{(x,u)}\delta_q \mathcal{F}_t, \begin{pmatrix} 0 & -I_{d_x} \\ I_{d_x} & \gamma_x I_{d_x} \end{pmatrix} \nabla_{(x,u)}\delta_q \mathcal{F}_t \right\rangle_{q_t}. \tag{16}$$

Decomposing the matrix $\begin{pmatrix} 0 & -I_{d_\theta} \\ I_{d_\theta} & \gamma_\theta I_{d_\theta} \end{pmatrix}$ into symmetric and skew-symmetric components (write $D$, $Q$ respectively), i.e.

$$\begin{pmatrix} 0 & -I_{d_\theta} \\ I_{d_\theta} & \gamma_\theta I_{d_\theta} \end{pmatrix} = \underbrace{\begin{pmatrix} 0 & -I_{d_\theta} \\ I_{d_\theta} & 0 \end{pmatrix}}_{=:Q} + \underbrace{\begin{pmatrix} 0 & 0 \\ 0 & \gamma_\theta I_{d_\theta} \end{pmatrix}}_{=:D},$$

we can simplify the RHS of (15) to

$$\text{RHS (15)} = -\left\langle \nabla_{(\theta,m)}\mathcal{F}_t, D\nabla_{(\theta,m)}\mathcal{F}_t \right\rangle = -\gamma_\theta \|\nabla_m \mathcal{F}_t\|^2.$$

Similarly, we can show that

$$\text{(16)} = -\gamma_x \|\nabla_u \delta_q \mathcal{F}_t\|_{q_t}^2.$$

As such, for $\gamma_x \geq 0$, $\gamma_\theta \geq 0$, the claim follows. $\qquad\square$

## F   PROOF OF PROPOSITION 4.1

The outline of the proof of Proposition 4.1 is:

- **Step 1** (Appendix F.1): Explicitly computing an upper bound of the time derivative of $\mathcal{F}$ as a quadratic form.
- **Step 2** (Appendix F.2): Under the conditions specified in (34), we show that this time derivative is bounded above by another quadratic form that allows us to apply the log-Sobolev inequality of Proposition D.1.
- **Step 3**: Using log-Sobolev inequality, Proposition D.1, and Grönwall's inequality, we obtain the desired result.

The remainder of the sections are dedicated to supporting the proof of Steps 1 and 2. This is done by developing the technical tools and carrying out explicit computations. The particular roles of the following sections are:

1. Appendix F.4.1: Computing the time derivative of $\mathcal{L}$ for Step 1.
2. Appendices F.4.2 to F.4.4: Introducing the adjoint, commutator and another operator, as well as their explicit forms.
3. Appendix F.4.5: Using the operators and explicit forms in Appendices F.4.2 to F.4.4, we can upper bound terms introduced in Appendix F.4.1 for Step 1. We utilize this in Step 1.
4. Appendix F.4.6. Bounding the cross terms (or interaction terms) between $\theta$ and $x$ that arises in the time derivative.
5. Appendix F.4.7. Establishing sufficient conditions for a matrix with a particular form to be positive semi-definite given in Proposition F.18. This is used in Step 2.

Recall that the Lyapunov function is given by:

$$\mathcal{L} := \mathcal{F} + \log p_{\theta_*}(y) + \left\|\nabla_{(\theta,m)}\mathcal{F}\right\|^2_{T_{(\theta,m)}} + \left\|\nabla_{(x,u)}\delta_q\mathcal{F}\right\|^2_{T_{(x,u)}}$$

where

$$T_{(\theta,m)} := \frac{1}{K}\begin{pmatrix} \tau_\theta I_{d_\theta} & \frac{\tau_{\theta m}}{2}I_{d_\theta} \\ \frac{\tau_{\theta m}}{2}I_{d_\theta} & \tau_m I_{d_\theta} \end{pmatrix}, \quad T_{(x,u)} := \frac{1}{K}\begin{pmatrix} \tau_x I_{d_x} & \frac{\tau_{xu}}{2}I_{d_x} \\ \frac{\tau_{xu}}{2}I_{d_x} & \tau_u I_{d_x} \end{pmatrix} \tag{17}$$

*Proof of Proposition 4.1.* The proof is completed in the following three steps:

**Step 1.** In Section F.1, we show that the time derivative of $\mathcal{L}(z_t)$ satisfies the following upper bound:

$$\frac{d}{dt}\mathcal{L}_t \leq -\left\langle \nabla_{(\theta,m)}\mathcal{F}_t, S_{(\theta,m)}\nabla_{(\theta,m)}\mathcal{F}_t\right\rangle - \left\langle \nabla_{(x,u)}\delta_q\mathcal{F}_t, S_{(x,u)}\nabla_{(x,u)}\delta_q\mathcal{F}_t\right\rangle_{q_t}.$$

where $S_{(x,u)}$ and $S_{(\theta,m)}$ are suitable matrices defined in (30) and (31), respectively.

**Step 2.** Then, in Section F.2, when Equation (34) holds for some rate $\varphi > 0$, we have that

$$S_{(\theta,m)} \succeq \varphi C\left(T_{(\theta,m)} + \frac{1}{2C}I_{d_\theta}\right),$$

$$S_{(x,u)} \succeq \varphi C\left(T_{(x,u)} + \frac{1}{2C}I_{d_x}\right),$$

where $C := \min\{C_{\mathcal{E}}, \eta_\theta, \eta_x\}$.

**Step 3.** By Step 1 and Step 2, we obtain

$$\frac{d}{dt}\mathcal{L}_t \leq -\varphi C\left(\frac{1}{2C}\left\|\nabla_{(\theta,m)}\mathcal{F}_t\right\|^2 + \left\|\nabla_{(\theta,m)}\mathcal{F}_t\right\|^2_{T_{(\theta,m)}}\right)$$
$$-\varphi C\left(\frac{1}{2C}\left\|\nabla_{(x,u)}\delta_q\mathcal{F}_t\right\|^2_{q_t} + \left\|\nabla_{(x,u)}\delta_q\mathcal{F}_t\right\|^2_{T_{(x,u)}}\right).$$

Comparing the above to (10) and applying Proposition D.1,

$$\frac{d}{dt}\mathcal{L}_t \leq -\varphi C \mathcal{L}_t,$$

and application of Gronwall's inequality yields the final result

$$\mathcal{F}_t + \log p_{\theta_*} \leq \mathcal{L}_0 \exp\left(-\frac{C\mathcal{E}}{10}t\right).$$

$\square$

### F.1 PROOF OF STEP 1

As shown in Proposition F.2, the time derivative of the Lyapunov function $\mathcal{L}$ is given by

$$\frac{d}{dt}\mathcal{L}_t = -\left\langle \nabla_{(\theta,m)}\mathcal{L}_t, \begin{pmatrix} 0 & -I_{d_\theta} \\ I_{d_\theta} & \gamma_\theta I_{d_\theta} \end{pmatrix} \nabla_{(\theta,m)}\mathcal{F}_t \right\rangle \tag{18}$$

$$-\left\langle \nabla_{(x,u)}\delta_q\mathcal{L}_t, \begin{pmatrix} 0 & -I_{d_x} \\ I_{d_x} & \gamma_\theta I_{d_x} \end{pmatrix} \nabla_{(x,u)}\delta_q\mathcal{F}_t \right\rangle_{q_t}, \tag{19}$$

where we abbreviate $\mathcal{L}_t := \mathcal{L}(z_t)$ and $\mathcal{F}_t := \mathcal{F}(z_t)$.

We deal with the inner products in (18,19) one-by-one, starting with (18): we first compute the gradient of $\mathcal{L}$ w.r.t. to $(\theta, m)$, finding that

$$\nabla_{(\theta,m)}\mathcal{L} = \nabla_{(\theta,m)}\mathcal{F} + 2[\nabla^2_{(\theta,m)}\mathcal{F}]\,T_{(\theta,m)}\nabla_{(\theta,m)}\mathcal{F} + 2\mathbb{E}_q\left[[\nabla_{(\theta,m)}\nabla_{(x,u)}\delta_q\mathcal{F}]\,T_{(x,u)}\nabla_{(x,u)}\delta_q\mathcal{F}\right].$$

It then follows that

$$(18) = -\left\langle \nabla_{(\theta,m)}\mathcal{F}_t, \tilde{\Gamma}_\theta\nabla_{(\theta,m)}\mathcal{F}_t \right\rangle \tag{20}$$

$$-2\left\langle \nabla_{(\theta,m)}\nabla_{(x,u)}[\delta_q\mathcal{F}_t]\,T_{(x,u)}\nabla_{(x,u)}\delta_q\mathcal{F}_t, \begin{pmatrix} 0 & -I_{d_\theta} \\ I_{d_\theta} & \gamma_\theta I_{d_\theta} \end{pmatrix} \nabla_{(\theta,m)}\mathcal{F}_t \right\rangle_{q_t}. \tag{21}$$

where

$$\tilde{\Gamma}_\theta := \begin{pmatrix} 0 & -I_{d_\theta} \\ I_{d_\theta} & \gamma_\theta I_{d_\theta} \end{pmatrix} + 2T_{(\theta,m)}\nabla^2_{(\theta,m)}\mathcal{F}_t\begin{pmatrix} 0 & -I_{d_\theta} \\ I_{d_\theta} & \gamma_\theta I_{d_\theta} \end{pmatrix}.$$

By $\mathcal{F}$'s definition,

$$\nabla^2_{(\theta,m)}\mathcal{F} = \begin{pmatrix} \nabla^2_\theta\mathcal{F} & 0_{d_\theta} \\ 0_{d_\theta} & \eta_\theta I_{d_\theta} \end{pmatrix};$$

whence we see that

$$\nabla^2_{(\theta,m)}\mathcal{F}\begin{pmatrix} 0_{d_\theta} & -I_{d_\theta} \\ I_{d_\theta} & \gamma_\theta I_{d_\theta} \end{pmatrix} = \begin{pmatrix} 0_{d_\theta} & -\nabla^2_\theta\mathcal{F} \\ \eta_\theta I_{d_\theta} & \gamma_\theta\eta_\theta I_{d_\theta} \end{pmatrix}$$

$$\Rightarrow T_{(\theta,m)}\nabla^2_{(\theta,m)}\mathcal{F}\begin{pmatrix} 0_{d_\theta} & -I_{d_\theta} \\ I_{d_\theta} & \gamma_\theta I_{d_\theta} \end{pmatrix} = \begin{pmatrix} \frac{\tau_{\theta m}\eta_\theta}{2}I_{d_\theta} & \frac{\tau_{\theta m}\gamma_\theta\eta_\theta}{2}I_{d_\theta} - \tau_\theta\nabla^2_\theta\mathcal{F} \\ \tau_m\eta_\theta I_{d_\theta} & \tau_m\gamma_\theta\eta_\theta I_{d_\theta} - \frac{\tau_{\theta m}}{2}\nabla^2_\theta\mathcal{F} \end{pmatrix},$$

$$\Rightarrow \tilde{\Gamma}_\theta = \begin{pmatrix} \tau_{\theta m}\eta_\theta I_{d_\theta} & [\tau_{\theta m}\gamma_\theta\eta_\theta - 1]I_{d_\theta} - 2\tau_\theta\nabla^2_\theta\mathcal{F}_t \\ [2\tau_m\eta_\theta + 1]I_{d_\theta} & [2\tau_m\eta_\theta + 1]\gamma_\theta I_{d_\theta} - \tau_{\theta m}\nabla^2_\theta\mathcal{F}_t \end{pmatrix}.$$

Given that, for any matrix $A$, we have $\langle \cdot, A\cdot \rangle = \langle \cdot, A^{sym}\cdot \rangle$ where $A^{sym} = (A + A^\top)/2$, then with $\Gamma_\theta := \frac{\tilde{\Gamma}_\theta + \tilde{\Gamma}_\theta^\top}{2}$, we obtain that

$$(20) = -\left\langle \nabla_{(\theta,m)}\mathcal{F}_t, \Gamma_\theta\nabla_{(\theta,m)}\mathcal{F}_t \right\rangle, \tag{22}$$

and hence that

$$\Gamma_\theta = \begin{pmatrix} \tau_{\theta m}\eta_\theta I_{d_\theta} & \left[\tau_m + \frac{\tau_{\theta m}\gamma_\theta}{2}\right]\eta_\theta I_{d_\theta} - \tau_\theta \nabla_\theta^2 \mathcal{F}_t \\ \left[\tau_m + \frac{\tau_{\theta m}\gamma_\theta}{2}\right]\eta_\theta I_{d_\theta} - \tau_\theta \nabla_\theta^2 \mathcal{F}_t & [2\tau_m\eta_\theta + 1]\gamma_\theta I_{d_\theta} - \tau_{\theta m}\nabla_\theta^2 \mathcal{F}_t \end{pmatrix}. \tag{23}$$

We now turn our attention to (19). Defining $\mathcal{H}_t(x, u) := \sqrt{\frac{q_t(x,u)}{\rho_{\theta_t,\eta_x}(x,u)}}$ and Proposition F.2, we see that

$$(19) = -4\left\langle \nabla_{(x,u)}\log\mathcal{H}_t, \begin{pmatrix} 0_{d_x} & -I_{d_x} \\ I_{d_x} & \gamma_x I_{d_x} \end{pmatrix}\nabla_{(x,u)}\log\mathcal{H}_t\right\rangle_{q_t} \tag{24}$$

$$-8\left\langle \nabla_{(x,u)}\frac{\nabla_{(x,u)}^* T_{(x,u)}\nabla_{(x,u)}\mathcal{H}_t}{\mathcal{H}_t}, \begin{pmatrix} 0_{d_x} & -I_{d_x} \\ I_{d_x} & \gamma_x I_{d_x} \end{pmatrix}\nabla_{(x,u)}\log\mathcal{H}_t\right\rangle_{q_t} \tag{25}$$

$$+4\left\langle \nabla_{(x,u)}\nabla_{(\theta,m)}\left[\log\rho_{\theta_t}\right]T_{(\theta,m)}\nabla_{(\theta,m)}\mathcal{F}_t, \begin{pmatrix} 0_{d_x} & -I_{d_x} \\ I_{d_x} & \gamma_x I_{d_x} \end{pmatrix}\nabla_{(x,u)}\log\mathcal{H}_t\right\rangle_{q_t}, \tag{26}$$

Furthermore, we have

$$(24) = -4\gamma_x\left\|\nabla_u\log\mathcal{H}_t\right\|_{q_t}^2 = -\gamma_x\left\|\nabla_u\log\frac{q_t}{\rho_{\theta_t,\eta_x}}\right\|_{q_t}^2.$$

In Proposition F.11, we show that (25) can be further simplified as follows:

$$(25) = -2\gamma_x\left\langle \nabla_{(x,u)}\nabla_u\left[\log\frac{q_t}{\rho_{\theta_t,\eta_x}}\right], T_{(x,u)}\nabla_{(x,u)}\nabla_u\left[\log\frac{q_t}{\rho_{\theta_t,\eta_x}}\right]\right\rangle_{q_t}$$

$$-\left\langle \nabla_{(x,u)}\log\frac{q_t}{\rho_{\theta_t,\eta_x}}, \tilde{\Gamma}_x\nabla_{(x,u)}\log\frac{q_t}{\rho_{\theta_t,\eta_x}}\right\rangle_{q_t}$$

where

$$\tilde{\Gamma}_x = \begin{pmatrix} \frac{\tau_{xu}\eta_x}{2}I_{d_x} & \frac{\tau_u + \tau_{xu}\gamma_x}{2}\eta_x I_{d_x} + \frac{\tau_x}{2}\nabla_x^2\ell \\ \frac{\tau_u + \tau_{xu}\gamma_x}{2}\eta_x I + \frac{\tau_x}{2}\nabla_x^2\ell & 2\gamma_x\tau_u\eta_x I_{d_x} + \frac{\tau_{xu}}{2}\nabla_x^2\ell \end{pmatrix}.$$

We can then combine (24) and (25) to obtain

$$(24) + (25) = -2\gamma_x\left\langle \nabla_{(x,u)}\nabla_u\left[\log\frac{q_t}{\rho_{\theta_t,\eta_x}}\right], T_{(x,u)}\nabla_{(x,u)}\nabla_u\left[\log\frac{q_t}{\rho_{\theta_t,\eta_x}}\right]\right\rangle_{q_t}$$

$$-\left\langle \nabla_{(x,u)}\log\frac{q_t}{\rho_{\theta_t,\eta_x}}, \Gamma_x\nabla_{(x,u)}\log\frac{q_t}{\rho_{\theta_t,\eta_x}}\right\rangle_{q_t}.$$

where $\Gamma_x = \tilde{\Gamma}_x + \begin{pmatrix} 0_{d_x} & 0_{d_x} \\ 0_{d_x} & \gamma_x I_{d_x} \end{pmatrix}$ is again positive semi-definite.

Given any p.s.d. matrix $M \in \mathbb{R}^{d\times d}$ and a general matrix $A \in \mathbb{R}^{d\times d}$, it holds generically that

$$\langle A, MA\rangle = \langle A, LL^\top A\rangle = \langle L^\top A, L^\top A\rangle = \|L^\top A\|^2 \geq 0,$$

where $M = LL^\top$ is Cholesky decomposition of $M$. Thus, we have the following upper bound:

$$(24) + (25) \leq -\left\langle \nabla_{(x,u)}\log\frac{q_t}{\rho_{\theta_t,\eta_x}}, \Gamma_x\nabla_{(x,u)}\log\frac{q_t}{\rho_{\theta_t,\eta_x}}\right\rangle_{q_t} \tag{27}$$

$$= -\left\langle \nabla_{(x,u)}\delta_q\mathcal{F}_t, \Gamma_x\nabla_{(x,u)}\delta_q\mathcal{F}_t\right\rangle_{q_t} \tag{28}$$

**Cross Terms (21) and (26).** We will now deal with the cross terms of (26) and (21). In Proposition F.17, we show the following upper-bound: for all $\varepsilon > 0$, it holds that

$$(26) + (21) \leq \varepsilon\left\langle \nabla_{(\theta,m)}\mathcal{F}_t, \Gamma_\times\nabla_{(\theta,m)}\mathcal{F}_t\right\rangle + \frac{1}{\varepsilon}\left\langle \nabla_{(x,u)}\delta_q\mathcal{F}_t, \nabla_{(x,u)}\delta_q\mathcal{F}_t\right\rangle_{q_t}, \tag{29}$$

where $\Gamma_\times = K^2 \begin{pmatrix} \tau_\theta^2 I_{d_\theta} & \tau_\theta \left( \frac{\tau_{xu} + \tau_{\theta m}}{2} \right) I_{d_\theta} \\ \tau_\theta \left( \frac{\tau_{xu} + \tau_{\theta m}}{2} \right) I_{d_\theta} & \left( \left( \frac{\tau_{xu} + \tau_{\theta m}}{2} \right)^2 + \tau_x^2 \right) I_{d_\theta} \end{pmatrix}$.

**Conclusion of Step 1.** Combining the results in (22, 28, 29), we upper bound the time derivative of the Lyapunov function as

$$\frac{d}{dt} \mathcal{L}_t \leq - \left\langle \nabla_{(\theta,m)} \mathcal{F}_t, S_{(\theta,m)} \nabla_{(\theta,m)} \mathcal{F}_t \right\rangle - \left\langle \nabla_{(x,u)} \delta_q \mathcal{F}_t, S_{(x,u)} \nabla_{(x,u)} \delta_q \mathcal{F}_t \right\rangle_{q_t},$$

where

$$S_{(x,u)} = \Gamma_x - \frac{1}{\varepsilon} I_{d_x} = \begin{pmatrix} \left( \frac{\tau_{xu}\eta_x - 1}{2} - \frac{1}{\varepsilon} \right) I_{d_x} & \frac{\tau_u + \tau_{xu}\gamma_x}{2} \eta_x I_{d_x} + \frac{\tau_x}{2} \nabla_x^2 \ell \\ \frac{\tau_u + \tau_{xu}\gamma_x}{2} \eta_x I_{d_x} + \frac{\tau_x}{2} \nabla_x^2 \ell & \left( \gamma_x (2\tau_u \eta_x + 1) - \frac{1}{\varepsilon} \right) I_{d_x} + \frac{\tau_{xu}}{2} \nabla_x^2 \ell \end{pmatrix}, \tag{30}$$

$$S_{(\theta,m)} = \Gamma_\theta - \varepsilon \Gamma_\times = \begin{pmatrix} s_\theta I_{d_\theta} & s_{\theta m} I_{d_\theta} - \tau_\theta \nabla_\theta^2 \mathcal{F}_t \\ s_{\theta m} I_{d_\theta} - \tau_\theta \nabla_\theta^2 \mathcal{F}_t & s_m I_{d_\theta} - \tau_{\theta m} \nabla_\theta^2 \mathcal{F}_t \end{pmatrix}, \tag{31}$$

with constants defined as

$$s_\theta := \tau_{\theta m}\eta_\theta - \varepsilon K^2 \tau_\theta^2, \quad s_{\theta m} := \tau_m \eta_\theta + \frac{\tau_{\theta m}\gamma_\theta}{2}\eta_\theta - \varepsilon K^2 \tau_\theta \frac{\tau_{xu} + \tau_{\theta m}}{2},$$

$$s_m := (2\tau_m \eta_\theta + 1)\gamma_\theta - \varepsilon K^2 \left( \left( \frac{\tau_{xu} + \tau_{\theta m}}{2} \right)^2 + \tau_x^2 \right).$$

### F.2  Proof of Step 2

By the definitions of $T_{(\theta,m)}$ and $T_{(x,u)}$ in (17), and those of $S_{(\theta,m)}$ and $S_{(x,u)}$ in (30, 31),

$$S_{(\theta,m)} - \varphi C \left( T_{(\theta,m)} + \frac{1}{2C} I_{d_\theta} \right) = \begin{pmatrix} \alpha_\theta I_{d_\theta} & \beta_\theta I_{d_\theta} - \tau_\theta \nabla_\theta^2 \mathcal{F} \\ \beta_\theta I_{d_\theta} - \tau_\theta \nabla_\theta^2 \mathcal{F} & \kappa_\theta I_{d_\theta} - \tau_{\theta m} \nabla_\theta^2 \mathcal{F} \end{pmatrix}, \tag{32}$$

$$S_{(x,u)} - \varphi C \left( T_{(x,u)} + \frac{1}{2C} I_{d_x} \right) = \begin{pmatrix} \alpha_x I_{d_x} & \beta_x I_{d_x} + \frac{\tau_x}{2} \nabla_x^2 \ell \\ \beta_x I_{d_x} + \frac{\tau_x}{2} \nabla_x^2 \ell & \kappa_x I_{d_x} + \frac{\tau_{xu}}{2} \nabla_x^2 \ell \end{pmatrix}. \tag{33}$$

where

$$\alpha_\theta := \tau_{\theta m}\eta_\theta - \varepsilon K^2 \tau_\theta^2 - \varphi C \left( \tau_\theta + \frac{1}{2C} \right),$$

$$\beta_\theta := \left( \tau_m + \frac{\tau_{\theta m}\gamma_\theta}{2} \right) \eta_\theta - \varepsilon K^2 \tau_\theta \left( \frac{\tau_{xu} + \tau_{\theta m}}{2} \right) - \varphi C \frac{\tau_{\theta m}}{2},$$

$$\kappa_\theta := (2\tau_m \eta_\theta + 1)\gamma_\theta - \varepsilon K^2 \left( \left( \frac{\tau_{xu} + \tau_{\theta m}}{2} \right)^2 + \tau_x^2 \right) - \varphi C \left( \tau_m + \frac{1}{2C} \right),$$

$$\alpha_x = \frac{\tau_{xu}\eta_x}{2} - \frac{1}{\varepsilon} - \varphi C \left( \tau_x + \frac{1}{2C} \right),$$

$$\beta_x = \frac{(\tau_u + \tau_{xu}\gamma_x)\eta_x}{2} - \varphi C \frac{\tau_{xu}}{2},$$

$$\kappa_x = \gamma_x (2\tau_u \eta_x + 1) - \frac{1}{\varepsilon} - \varphi C \left( \tau_u + \frac{1}{2C} \right).$$

**Proposition F.1.** *Defining* $(\alpha_\theta, \beta_\theta, \kappa_\theta, \alpha_x, \beta_x, \kappa_x)$ *as above, assume that* $K \geq 2C_\mathcal{E}$ *if*

$$\tau_{\theta m} K - \alpha_\theta - \kappa_\theta \leq 0, \tag{34a}$$

$$\tau_\theta^2 K^2 + (\tau_{\theta m}\alpha_\theta - 2\beta_\theta\tau_\theta)K - \alpha_\theta\kappa_\theta + \beta_\theta^2 \leq 0, \tag{34b}$$

$$\tau_\theta^2 K^2 - (\tau_{\theta m}\alpha_\theta - 2\beta_\theta\tau_\theta)K - \alpha_\theta\kappa_\theta + \beta_\theta^2 \leq 0, \tag{34c}$$

$$\frac{\tau_{xu}}{2} K - \alpha_x - \kappa_x \leq 0, \tag{34d}$$

$$\frac{\tau_x^2}{4} K^2 - \left(\frac{\tau_{xu}}{2}\alpha_x - \beta_x\tau_x\right) K - \alpha_x\kappa_x + \beta_x^2 \leq 0, \tag{34e}$$

$$\frac{\tau_x^2}{4} K^2 + \left(\frac{\tau_{xu}}{2}\alpha_x - \beta_x\tau_x\right) K - \alpha_x\kappa_x + \beta_x^2 \leq 0, \tag{34f}$$

*hold for some rate* $\varphi > 0$. *The following lower bounds for* $S_{(x,u)}$ *and* $S_{(\theta,m)}$ *then hold:*

$$S_{(x,u)} \succeq \varphi C \left(T_{(x,u)} + \frac{1}{2C}I_{d_x}\right), \quad S_{(\theta,m)} \succeq \varphi C \left(T_{(\theta,m)} + \frac{1}{2C}I_{d_\theta}\right),$$

*where* $C := \min\{C_\mathcal{E}, \eta_\theta, \eta_x\}$.

*Proof.* Note the matrices in (32, 33) have the same form as that in (47). As we show below,

$$-KI_{d_x} \preceq \nabla_x^2 \ell \preceq KI_{d_x}, \quad -KI_{d_\theta} \preceq \nabla_\theta^2 \mathcal{F} \preceq KI_{d_\theta}. \tag{35}$$

Then, applying Proposition F.18 tells us that (32, 33) are positive semidefinite if the conditions (34) are satisfied.

Now, to prove the inequalities in (35). Assumption 2 and (Nesterov, 2003, Lemma 1.2.2) imply the first two inequalities and

$$-KI_{d_\theta} \preceq \nabla_\theta^2 \ell \preceq KI_{d_\theta}.$$

The other two inequalities in (35) follow from the above. To see this, for $\nabla_\theta^2 \mathcal{F} \preceq KI_{d_\theta}$, we have

$$\left\langle v, (KI_{d_\theta} - \nabla_\theta^2\mathcal{F})v \right\rangle = \int \left\langle v, (KI_{d_\theta} + \nabla_\theta^2\ell)v \right\rangle q(\mathrm{d}x \times \mathrm{d}u),$$

and since $\left\langle v, \nabla_\theta^2[\ell]v \right\rangle \geq -K\|v\|^2$, we have shown that $\left\langle v, (KI_{d_\theta} - \nabla_\theta^2\mathcal{F})v \right\rangle \geq 0$ implying the desired result of $\nabla_\theta^2\mathcal{F} \preceq KI_{d_\theta}$. A similar argument can be made for $-KI_{d_\theta} \preceq \nabla_\theta^2\mathcal{F}$. $\square$

## F.3 EXAMPLES FOR (34) HOLDING

We verified the above with a symbolic calculator written in Mathematica for the choices of $\eta_x, \eta_\theta, \gamma_x, \gamma_\theta$ hold for the following choices:

1. Rate $\varphi = \frac{1}{10}$, momentum parameters $\gamma_x = \frac{3}{2}$, $\gamma_\theta = 3$, $\eta_x = 2K$, $\eta_\theta = 2K$, and elements of the Lyapunov function $\tau_\theta = \frac{3}{8}$, $\tau_{\theta m} = \frac{5}{4}$, $\tau_m = 2$, $\tau_x = \frac{1}{4}$, $\tau_{xu} = \frac{5}{4}$, $\tau_u = \frac{7}{4}$, and $\varepsilon = 15$.

2. Rate $\varphi = \frac{1}{30}$, momentum parameters $\gamma_x = \frac{6}{5}$, $\gamma_\theta = \frac{5}{2}$, $\eta_x = 2K$, $\eta_\theta = 2K$, and elements of the Lyapunov function $\tau_\theta = \frac{3}{8}$, $\tau_{\theta m} = \frac{5}{4}$, $\tau_m = 2$, $\tau_x = \frac{1}{4}$, $\tau_{xu} = 1$, $\tau_u = \frac{3}{2}$, and $\varepsilon = 15$.

## F.4 SUPPORTING PROOFS AND DERIVATIONS

### F.4.1 THE DERIVATIVE OF $\mathcal{L}_t$ ALONG THE FLOW

Here, we prove (18,19).

**Proposition F.2.** *The derivative of the Lyapunov function* $\mathcal{L}$ *is given by*

$$\frac{d}{dt}\mathcal{L}_t = -\left\langle \nabla_{(\theta,m)}\mathcal{L}_t, \begin{pmatrix} 0 & -I_{d_\theta} \\ I_{d_\theta} & \gamma_\theta I_{d_\theta} \end{pmatrix} \nabla_{(\theta,m)}\mathcal{F}_t \right\rangle - \left\langle \nabla_{(x,u)}\delta_q\mathcal{L}_t, \begin{pmatrix} 0 & -I_{d_x} \\ I_{d_x} & \gamma_\theta I_{d_x} \end{pmatrix} \nabla_{(x,u)}\delta_q\mathcal{F}_t \right\rangle_{q_t},$$

*where the first variation is given by*

$$\delta_q \mathcal{L}_t = \delta_q \mathcal{F}_t + \delta_q \mathcal{L}_t^1 + \delta_q \mathcal{L}_t^2$$
$$= \log \frac{q(x,u)}{\rho_{\theta,\eta_x}(x,u)} + 1 - 2 \left\langle \nabla_{(\theta,m)} \log \rho_\theta(x,u), T_{(\theta,m)} \nabla_{(\theta,m)} \mathcal{F}[z] \right\rangle$$
$$+ \frac{4}{\mathcal{H}} \nabla^*_{(x,u)} T_{(x,u)} \nabla_{(x,u)} \mathcal{H},$$

*with $\nabla^*$ and $\mathcal{H}$ defined in Proposition F.5.*

*Proof.* From the linearity of $\mathcal{L}$, we have

$$\dot{\mathcal{L}}_t = \dot{\mathcal{F}}_t + \dot{\mathcal{L}}_t^1 + \dot{\mathcal{L}}_t^2$$

where

$$\mathcal{L}_t^1 := \left\langle \nabla_{(\theta,m)} \mathcal{F}_t, T_{(\theta,m)} \nabla_{(\theta,m)} \mathcal{F}_t \right\rangle,$$
$$\mathcal{L}_t^2 := \left\langle \nabla_{(x,u)} \delta_q \mathcal{F}_t, T_{(x,u)} \nabla_{(x,u)} \delta_q \mathcal{F}_t \right\rangle_{q_t}.$$

From Appendix C.2, it can be seen that the time derivatives of $\mathcal{F}_t$, $\mathcal{L}_t^1$, and $\mathcal{L}_t^2$, are given by

$$\dot{\mathcal{F}}_t = \left\langle \nabla_{(\theta,m)} \mathcal{F}_t, (\dot{\theta}, \dot{m})_t \right\rangle + \left\langle \nabla_{(x,u)} \delta_q \mathcal{F}_t, (\dot{x}, \dot{u})_t \right\rangle_{q_t}$$
$$\dot{\mathcal{L}}_t^1 = \left\langle \nabla_{(\theta,m)} \mathcal{L}_t^1, (\dot{\theta}, \dot{m})_t \right\rangle + \left\langle \nabla_{(x,u)} \delta_q \mathcal{L}_t^1, (\dot{x}, \dot{u})_t \right\rangle_{q_t}$$
$$\dot{\mathcal{L}}_t^2 = \left\langle \nabla_{(\theta,m)} \mathcal{L}_t^2, (\dot{\theta}, \dot{m})_t \right\rangle + \left\langle \nabla_{(x,u)} \delta_q \mathcal{L}_t^2, (\dot{x}, \dot{u})_t \right\rangle_{q_t}$$

where $(\dot{\theta}, \dot{m})_t = \begin{pmatrix} 0 & -I_{d_\theta} \\ I_{d_\theta} & \gamma_\theta I_{d_\theta} \end{pmatrix} \nabla_{(\theta,m)} \mathcal{F}_t$ and $(\dot{x}, \dot{u})_t = \begin{pmatrix} 0 & -I_{d_x} \\ I_{d_x} & \gamma_\theta I_{d_x} \end{pmatrix} \nabla_{(x,u)} \delta_q \mathcal{F}_t$. From Proposition F.3, Proposition F.4, and Proposition F.5, and linearity of the inner product, we obtain as desired. $\qquad\square$

**Proposition F.3.** *The first variation of $\mathcal{F}$ is given by*

$$\delta_q \mathcal{F}[z](x,u) = \log \frac{q(x,u)}{\rho_{\theta,\eta_x}(x,u)} + 1,$$

*where $z = (\theta, m, q)$.*

*Proof.* It can be seen that $\mathcal{F}$ is (VI-I)-typed equation with $f((\theta,m),(x,u),q,g) = q \log \frac{q}{\rho_{\theta,\eta_x}(x)}$. Thus, using the formula (8), we obtain as its first variation

$$\delta_q \mathcal{F}[z](x,u) = \log \frac{q(x,u)}{\rho_{\theta,\eta_x}(x,u)} + 1.$$

Hence, we have as desired. $\qquad\square$

For the first variation of $\mathcal{L}_1$, we have the following result:

**Proposition F.4.** *The first variation of $\mathcal{L}_1$ is given by*

$$\delta_q \mathcal{L}_1[z](x,u) = -2 \left\langle \nabla_{(\theta,m)} \log \rho_\theta(x,u), T_{(\theta,m)} \nabla_{(\theta,m)} \mathcal{F}[z] \right\rangle,$$

*where $z = (\theta, m, q)$.*

*Proof.* It can be seen that $\mathcal{L}_1$ is a variational integral of type (VI-II) with

$$f((\theta,m),(x,u),(x',u'),q,q') = qq' \left\langle \nabla_\theta \log \rho(x,u), T_{\theta\theta} \nabla_\theta \log \rho(x',u') \right\rangle$$
$$- q \left\langle \nabla_\theta \log \rho(x,u), T_{\theta m} m \right\rangle$$
$$- q' \left\langle m, T_{m\theta} \nabla_\theta \log \rho(x',u') \right\rangle$$
$$+ \left\langle m, T_{mm} m \right\rangle,$$

where we write $T_{\theta\theta}$ for the upper left block of $T_{(\theta,m)}$, and similarly for $T_{\theta m}$, and $T_{mm}$.

Using the formula (9) and the fact that

$$\nabla_{(4)}f((\theta,m),(x,u),(x',u'),q,q') = q'\left\langle \nabla_\theta \log \rho(x,u), T_{\theta\theta}\nabla_\theta \log \rho(x',u')\right\rangle$$
$$- \left\langle \nabla_\theta \log \rho(x,u), T_{\theta m}m\right\rangle$$

For the first term of (9), we have

$$\int \nabla_{(4)}f((\theta,m),(x,u),(x',u'),q(x,u),q(x',u'))\mathrm{d}x'\mathrm{d}u'$$
$$= -\left\langle \nabla_{(\theta,m)} \log \rho(x,u), T_{(\theta,m)}\nabla_{(\theta,m)}\mathcal{F}[z]\right\rangle,$$

and by symmetry, it follows similarly for the last term of (9). We have obtained as desired. $\qquad\square$

**Proposition F.5.** *The first variation of $\mathcal{L}_2$ is given by*

$$\delta_q \mathcal{L}_2[z] = \frac{4}{\mathcal{H}}\nabla^*_{(x,u)}T_{(x,u)}\nabla_{(x,u)}\mathcal{H},$$

*where $\mathcal{H}(x,u) := \sqrt{\frac{q(x,u)}{\rho_{\theta,\eta_x}(x,u)}}$ and the operator $\nabla^*_{(x,u)}$ is given as*

$$\nabla^*_{(x,u)}(f)(x,u) := -\left\langle \nabla_{(x,u)} \log \rho_{\theta,\eta_x}(x,u), f(x,u)\right\rangle - \nabla_{(x,u)} \cdot [f](x,u),$$

*where $\nabla_x\cdot$ is the divergence operator w.r.t. $x$.*

*Proof.* First note that $\mathcal{L}_2$ can be written equivalently as

$$\left\langle \frac{\nabla_{(x,u)}q}{q} - \nabla_{(x,u)} \log \rho_{\theta,\eta_x}, T_{(x,u)}\left[\frac{\nabla_{(x,u)}q}{q} - \nabla_{(x,u)} \log \rho_\theta\right]\right\rangle_q$$

It can be seen that $\mathcal{L}_2$ is a variational integral of type (VI-I) with

$$f((\theta,m),(x,u),q,g) = q\left\langle \frac{g}{q} - \nabla_{(x,u)} \log \rho_{\theta,\eta_x}(x,u), T_{(x,u)}\left[\frac{g}{q} - \nabla_{(x,u)} \log \rho_{\theta,\eta_x}(x,u)\right]\right\rangle.$$

This follows from since

$$\nabla_{(3)}f((\theta,m),(x,u),q,g) = \left\langle \frac{g}{q} - \nabla_{(x,u)} \log \rho_{\theta,\eta_x}(x,u), T_{(x,u)}\left[\frac{g}{q} - \nabla_{(x,u)} \log \rho_{\theta,\eta_x}(x,u)\right]\right\rangle$$
$$- 2q\left\langle \frac{g}{q^2}, T_{(x,u)}\left[\frac{g}{q} - \nabla_{(x,u)} \log \rho_{\theta,\eta_x}(x,u)\right]\right\rangle$$
$$= -\left\langle \frac{g}{q} + \nabla_{(x,u)} \log \rho_{\theta,\eta_x}(x,u), T_{(x,u)}\left[\frac{g}{q} - \nabla_{(x,u)} \log \rho_{\theta,\eta_x}(x,u)\right]\right\rangle$$

and

$$\nabla_{(4)}f((\nabla,m),(x,u),q,g) = 2T_{(x,u)}\left[\frac{g}{q} - \nabla_{(x,u)} \log \rho_{\theta,\eta_x}(x,u)\right]$$

Using formula (8), we obtain

$$\delta_q\mathcal{L}_2 = -\left\langle \nabla_{(x,u)} \log q(x,u) + \nabla_{(x,u)} \log \rho_{\theta,\eta_x}(x,u), T_{(x,u)}\nabla_{(x,u)} \log \frac{q(x,u)}{\rho_{\theta,\eta_x}(x,u)}\right\rangle \quad (36a)$$

$$- \nabla_{(x,u)} \cdot \left(2T_{(x,u)}\nabla_{(x,u)} \log \frac{q(x,u)}{\rho_{\theta,\eta_x}(x,u)}\right) \quad (36b)$$

To obtain the desired result, note that we can rewrite

$$(36a) = -\left\langle \nabla_{(x,u)} \log \sqrt{q(x,u)} + \nabla_{(x,u)} \log \sqrt{\rho_{\theta,\eta_x}(x,u)}, \frac{4}{\mathcal{H}}T_{(x,u)}\nabla_{(x,u)} \log \mathcal{H}\right\rangle$$

$$(36b) = -\nabla_{(x,u)} \cdot \left(\frac{4}{\mathcal{H}}T_{(x,u)}\nabla_{(x,u)}\mathcal{H}\right),$$

$$= -\left\langle \nabla_{(x,u)}\frac{4}{\mathcal{H}}, T_{(x,u)}\nabla_{(x,u)}\mathcal{H}\right\rangle - \frac{4}{\mathcal{H}}\nabla_{(x,u)} \cdot (T_{(x,u)}\nabla_{(x,u)}\mathcal{H})$$

$$= \left\langle \nabla_{(x,u)} \log \mathcal{H}, \frac{4}{\mathcal{H}}T_{(x,u)}\nabla_{(x,u)}\mathcal{H}\right\rangle - \frac{4}{\mathcal{H}}\nabla_{(x,u)} \cdot (T_{(x,u)}\nabla_{(x,u)}\mathcal{H}).$$

Hence, we have obtained the desired result. $\qquad\square$

Throughout the proofs, we use $\nabla^*$ to denote the linear map that is an adjoint of $\nabla$ (the Euclidean gradient operator). For this section, we write $\rho := \rho_{\theta, \eta_x}$.

In the view of $\nabla$ as a linear map from $(L_\rho^2(\mathcal{X}; \mathbb{R}), \langle \cdot, \cdot \rangle_\rho)$ to $(L_\rho^2(\mathcal{X}; \mathcal{X}), \langle \cdot, \cdot \rangle_\rho)$ where $\mathcal{X} \in \{\mathbb{R}^{d_x}, \mathbb{R}^{d_x} \times \mathbb{R}^{d_x}\}$ and

$$L_\rho^2(\mathcal{X}; \mathcal{X}) := \{\text{sufficiently regular } v : \mathcal{X} \to \mathcal{X} \text{ such that } ||v||_\rho < \infty\}.$$

Then its adjoint $\nabla^*$ is given by

$$\nabla^* v := -\frac{1}{\rho} \nabla \cdot [\rho v] = -\langle \nabla \log \rho, v \rangle - \nabla \cdot v.$$

where $\nabla \cdot$ denotes the divergence operator. Whatever the gradient $\nabla$ is taken with respect to, the adjoint $\nabla^*$ is taken with respect to the corresponding quantity. In order to see the adjoint property, we have

$$
\begin{aligned}
\langle \nabla f, v \rangle_\rho &= \int \langle \nabla f(x, u), v(x, u) \rangle \, \rho(\mathrm{d}x \times \mathrm{d}u) \\
&= \int \langle \nabla f(x, u), \rho(x, u) v(x, u) \rangle \, \mathrm{d}x \times du \\
&= -\int f(x, u) \nabla \cdot [\rho(x, u) v(x, u)] \, \mathrm{d}x \times du = \langle f, \nabla^* v \rangle_\rho,
\end{aligned}
$$

where we used integration by parts combined with the divergence theorem and the fact that $\rho$ must vanish at the boundary to obtain the second equation.

Another case of interest is in the view of $\nabla$ as a linear map from $(L_\rho^2(\mathcal{X}; \mathcal{X}'), \langle \cdot, \cdot \rangle_\rho)$ to $(L_\rho^2(\mathcal{X}; \mathcal{X} \times \mathcal{X}'), \langle \cdot, \cdot \rangle_\rho)$, where the underlying inner product on the latter space is induced by the Frobenius inner product. In this case, the adjoint $\nabla^*$ is defined a similar way: Let $M \in L^2(\mathcal{X}, \mathcal{X} \times \mathcal{X}')$ and $v \in L^2(\mathcal{X}, \mathcal{X}')$

$$
\begin{aligned}
\langle \nabla v, M \rangle_\rho &= \int \sum_{i=1}^{d_\mathcal{X}} \sum_{j=1}^{d_{\mathcal{X}'}} (\nabla v)_{ij} M_{ij} \, \rho(\mathrm{d}x \times \mathrm{d}u) \\
&= \int \sum_{i=1}^{d_\mathcal{X}} \sum_{j=1}^{d_{\mathcal{X}'}} \partial_i [v_j] M_{ij} \, \rho(\mathrm{d}x \times \mathrm{d}u) = \int \sum_{j=1}^{d_{\mathcal{X}'}} \langle \nabla v_j, M e_j \rangle \, \rho(\mathrm{d}x \times \mathrm{d}u) \\
&= -\int \sum_{j=1}^{d_{\mathcal{X}'}} (v_j) \nabla \cdot (\rho M e_j) \, \mathrm{d}x \times du \\
&= -\int \sum_{j=1}^{d_{\mathcal{X}'}} (v_j) [\langle \nabla \log \rho, M e_j \rangle + \nabla \cdot (M e_j)] \, \rho(\mathrm{d}x \times \mathrm{d}u) \\
&= \langle \nabla^* M, v \rangle_\rho.
\end{aligned}
$$

where, for the second line, we apply integration by parts combined with the divergence theorem and the fact that $\rho$ vanish at the boundary. Each element of $(\nabla^* M)_i \in \mathcal{X}'$ for $i \in [d_{\mathcal{X}'}]$, is defined to be

$$(\nabla^* M)_i := -\langle \nabla \log \rho, M^\top e_i \rangle - \nabla \cdot (M^\top e_i).$$

Or, more succinctly, we have

$$(\nabla^* M) := -M^\top \nabla \log \rho - \nabla \cdot M,$$

where $\nabla \cdot M$ denotes the divergence operator for a matrix field defined as $(\nabla \cdot M)_i = \partial_j M_{ji}$ in Einstein's notation.

### F.4.3 Commutator $[\cdot, \cdot]$

Another quantity widely used in the proof is the commutator denoted by $[\cdot, \cdot]$. It can be thought of as an indicator for if two operators commute. It is defined as

$$[A, B]f = (AB)f - (BA)f.$$

If $[A, B]f = 0$, then $A$ and $B$ is said to commute. We list the following propositions and their proofs that will be useful for the proof of 4.1. Again for this section, we write $\rho := \rho_{\theta, \eta_x}$.

**Proposition F.6.** *Let* $f : \mathbb{R}^{d_x} \to \mathbb{R}$, *and* $f \in C^3$, *then we have*

$$[\nabla_u, \nabla_u^*]\nabla_u f = \eta_x \nabla_u f.$$

*Proof.* Expanding out the commutator, we obtain

$$[\nabla_u, \nabla_u^*]\nabla_u f = -\nabla_u \langle \nabla_u \log \rho, \nabla_u f \rangle - \nabla_u [\nabla_u \cdot (\nabla_u f)] + \nabla_u^2 f \nabla_u \log \rho + \nabla_u \cdot (\nabla_u^2 f),$$

and since $\nabla_u [\nabla_u \cdot (\nabla_u f)] = \partial_{u_i} \partial_{u_j} \partial_{u_j} f = \partial_{u_j} \partial_{u_j} \partial_{u_i} f = \nabla_u \cdot (\nabla_u^2 f)$. We have that

$$[\nabla_u, \nabla_u^*]\nabla_u f = -\nabla_u^2 \log \rho \, \nabla_u f = \eta_x \nabla_u f.$$

$\square$

**Proposition F.7.** *Let* $f : \mathbb{R}^{d_x} \to \mathbb{R}$ *and* $f \in C^3$, *we have that* $\nabla_x^*$ *and* $\nabla_u$ *commutes for* $\nabla_u f$ . *In other words, we have*

$$[\nabla_x^*, \nabla_u]\nabla_u f = 0.$$

*Proof.* We begin from the definition

$$
\begin{aligned}
[\nabla_x^*, \nabla_u]\nabla_u f ={}& \nabla_x^* \nabla_u^2 f - \nabla_u \nabla_x^* \nabla_u f \\
={}& -\nabla_u^2 f \, \nabla_x \log \rho - \nabla_x \cdot (\nabla_u^2 f) \\
&+ \nabla_u \langle \nabla_x \log \rho, \nabla_u f \rangle + \nabla_u \nabla_x \cdot (\nabla_u f) \\
={}& 0.
\end{aligned}
$$

This follows as

$$\nabla_x \cdot (\nabla_u^2 f) = \partial_{x_j} \partial_{u_j} \partial_{u_i} f = \partial_{u_i} \partial_{x_j} \partial_{u_j} f = \nabla_u \nabla_x \cdot (\nabla_u f),$$
$$\nabla_u \langle \nabla_x \log \rho, \nabla_u f \rangle = \nabla_u^2 f \, \nabla_x \log \rho + \nabla_u \nabla_x [\log \rho] \nabla_u f = \nabla_u^2 f \, \nabla_x \log \rho.$$

$\square$

**Proposition F.8.** *For* $f : \mathbb{R}^{d_x} \to \mathbb{R}$ *and* $f \in C^3$, *we have that* $\nabla_x$ *and* $\nabla_u^* \nabla_u$ *commutes, i.e., we have*

$$[\nabla_x, \nabla_u^* \nabla_u]f = 0.$$

*Proof.* We can see that

$$[\nabla_x, \nabla_u^* \nabla_u]f = \nabla_x \nabla_u^* \nabla_u f - \nabla_u^* \nabla_u \nabla_x f = 0.$$

This follows since the first term can be written as

$$
\begin{aligned}
\nabla_x \nabla_u^* \nabla_u f &= \nabla_x \langle \nabla_u \log \rho, \nabla_u f \rangle + \nabla_x \nabla_u \cdot (\nabla_u f) \\
&= \nabla_x \nabla_u [f] \, \nabla_u \log \rho + \nabla_x \nabla_u \cdot (\nabla_u f),
\end{aligned}
$$

and, for the second term, we have

$$\nabla_u^* \nabla_u \nabla_x f = \nabla_u \nabla_x [f] \, \nabla_u \log \rho + \nabla_u \cdot (\nabla_u \nabla_x f).$$

Since we have

$$\nabla_x \nabla_u \cdot (\nabla_u f) = \partial_{x_i} \partial_{u_j} \partial_{u_j} f = \partial_{u_j} \partial_{u_j} \partial_{x_i} f = \nabla_u \cdot (\nabla_u \nabla_x f),$$

we have as desired. $\square$

For this section, we write $\rho := \rho_{\theta,\eta_x}$. Let $f : \mathbb{R}^{2d_x} \to \mathbb{R}$, another operator denoted by $B$ that we are interested in is defined as follow:

$$B[f] := \nabla^*_{(x,u)} \left[ fQ \begin{pmatrix} -\nabla_x \log \rho \\ -\nabla_u \log \rho \end{pmatrix} \right] = \nabla^*_{(x,u)} \left[ f \begin{pmatrix} \nabla_u \log \rho \\ -\nabla_x \log \rho \end{pmatrix} \right]$$

This can be written equivalently as

$$B[f] = -\left\langle \nabla_{(x,u)} \log \rho, f \begin{pmatrix} \nabla_u \log \rho \\ -\nabla_x \log \rho \end{pmatrix} \right\rangle - \nabla_{(x,u)} \cdot \left( f \begin{pmatrix} \nabla_u \log \rho \\ -\nabla_x \log \rho \end{pmatrix} \right)$$

$$= -\left\langle \nabla_{(x,u)} f, \begin{pmatrix} \nabla_u \log \rho \\ -\nabla_x \log \rho \end{pmatrix} \right\rangle = -\langle \nabla_x f, \nabla_u \log \rho \rangle + \langle \nabla_x \log \rho, \nabla_u f \rangle.$$

One can generalize the operator $B$ to vector-valued functions $f : \mathbb{R}^{2d_x} \to \mathbb{R}^{d_{x'}}$ as follows (its equivalence when $d_{x'} = 1$ is easy to see):

$$B[f] := \nabla^*_{(x,u)} \left[ Q \begin{pmatrix} -\nabla_x \log \rho \\ -\nabla_u \log \rho \end{pmatrix} \otimes f \right] = \nabla^*_{(x,u)} \left[ \begin{pmatrix} \nabla_u \log \rho \\ -\nabla_x \log \rho \end{pmatrix} \otimes f \right].$$

This can be equivalently written as

$$B[f] = -\left[ f \otimes \begin{pmatrix} \nabla_u \log \rho \\ -\nabla_x \log \rho_\theta \end{pmatrix} \right] \nabla_{(x,u)} \log \rho - \nabla_{(x,u)} \cdot \left( \begin{pmatrix} \nabla_u \log \rho \\ -\nabla_x \log \rho \end{pmatrix} \otimes f \right)$$

$$= \left\langle \begin{pmatrix} \nabla_u \log \rho \\ -\nabla_x \log \rho \end{pmatrix}, \nabla_{(x,u)} \log \rho \right\rangle f - \nabla_{(x,u)} f^\top \begin{pmatrix} \nabla_u \log \rho \\ -\nabla_x \log \rho \end{pmatrix}$$

$$= \nabla_{(x,u)} f^\top \begin{pmatrix} \nabla_u \log \rho \\ -\nabla_x \log \rho \end{pmatrix}$$

where we use the fact that

$$\nabla_{(x,u)} \cdot \left( \begin{pmatrix} \nabla_u \log \rho \\ -\nabla_x \log \rho \end{pmatrix} \otimes f \right) = \partial_{(x,u)_j} \left[ f_i \begin{pmatrix} \nabla_u \log \rho \\ -\nabla_x \log \rho \end{pmatrix}_j \right]$$

$$= \partial_{(x,u)_j} [f_i] \begin{pmatrix} \nabla_u \log \rho \\ -\nabla_x \log \rho \end{pmatrix}_j + f_i \partial_{(x,u)_j} \left[ \begin{pmatrix} \nabla_u \log \rho \\ -\nabla_x \log \rho \end{pmatrix}_j \right]$$

$$= \partial_{(x,u)_j} [f_i] \begin{pmatrix} \nabla_u \log \rho \\ -\nabla_x \log \rho \end{pmatrix}_j = \nabla_{(x,u)} f^\top \begin{pmatrix} \nabla_u \log \rho \\ -\nabla_x \log \rho \end{pmatrix}.$$

In the following Proposition, we show that the operator is anti-symmetric.

**Proposition F.9.** *For all $f, g \in \mathbb{R}^{2d_x} \to \mathbb{R}^{d_{x'}}$, we have $\langle B[f], g \rangle_\rho = -\langle f, B[g] \rangle_\rho$.*

*Proof.* From the definition, we have

$$\langle B[f], g \rangle_\rho = \left\langle \nabla^*_{(x,u)} \left[ \begin{pmatrix} \nabla_u \log \rho \\ -\nabla_x \log \rho \end{pmatrix} \otimes f \right], g \right\rangle_\rho = \left\langle \begin{pmatrix} \nabla_u \log \rho \\ -\nabla_x \log \rho \end{pmatrix} \otimes f, \nabla_{(x,u)} g \right\rangle_\rho$$

$$= \left\langle f, \nabla_{(x,u)} g^\top \begin{pmatrix} \nabla_u \log \rho \\ -\nabla_x \log \rho \end{pmatrix} \right\rangle_\rho$$

where we use the adjoint operator of $\nabla^*$. Hence, we have shown that

$$\langle B[f], g \rangle_\rho = -\langle f, B[g] \rangle_\rho.$$

$\square$

**Proposition F.10.** *For $f : \mathbb{R}^{2d_x} \to \mathbb{R}$, we have*

$$[\nabla_u, B]f = \eta_x \nabla_x f$$

*and*

$$[\nabla_x, B]f = \nabla_x^2 \log \rho \, \nabla_u f$$

*Proof.* For the first equality, we begin by expanding out the commutator

$$[\nabla_u, B]f = \nabla_u B[f] - B\nabla_u[f].$$

Since we have

$$\nabla_u B[f] = -\nabla_u \nabla_x f \, \nabla_u \log \rho - \nabla_u^2 \log \rho \, \nabla_x f + \nabla_u^2 f \, \nabla_x \log \rho,$$

and,

$$B\nabla_u[f] = \nabla_u \nabla_{(x,u)} f \begin{pmatrix} \nabla_u \log \rho \\ -\nabla_x \log \rho \end{pmatrix},$$

then, we must have

$$[\nabla_u, B]f = -\nabla_u^2 \log \rho \, \nabla_x f = \eta_x \nabla_x f.$$

For the second inequality, we begin similarly

$$[\nabla_x, B]f = \nabla_x B[f] - B\nabla_x[f].$$

For the first term, we have

$$\nabla_x B[f] = -\nabla_x^2 f \, \nabla_u \log \rho + \nabla_x^2 \log \rho \, \nabla_u f + \nabla_u \nabla_x f \, \nabla_x \log \rho$$

and, for the second term,

$$B\nabla_x[f] = \nabla_x \nabla_{(x,u)} f \begin{pmatrix} \nabla_u \log \rho \\ -\nabla_x \log \rho \end{pmatrix}.$$

Hence, we have

$$[\nabla_x, B]f = [\nabla_x, B]f = \nabla_x^2 \log \rho \, \nabla_u f.$$

$\square$

### F.4.5 SIMPLIFYING (25)

For this section, we drop the subscript $\rho_{\theta,\eta_x}$ and write $\rho$.

**Proposition F.11** (Simplifying (25)). *We have that*

$$(25) = -2\gamma_x \left\langle \nabla_{(x,u)} \nabla_u \left[\log \frac{q_t}{\rho_t}\right], T_{(x,u)} \nabla_{(x,u)} \nabla_u \left[\log \frac{q_t}{\rho_t}\right] \right\rangle_{q_t}$$

$$- \left\langle \nabla_{(x,u)} \log \frac{q_t}{\rho_t}, \tilde{\Gamma}_x \nabla_{(x,u)} \log \frac{q_t}{\rho_t} \right\rangle_{q_t},$$

*where $\rho := \rho_{\theta,\eta_x}$ (also similarly for $\rho_t$) and*

$$\tilde{\Gamma}_x = \frac{1}{2} \begin{pmatrix} \tau_{xu}\eta_x I_{d_x} & (\tau_u \eta_x + \tau_{xu} \gamma_x \eta_x) \, I_{d_x} + \tau_x \nabla_x^2[\ell] \\ (\tau_u \eta_x + \tau_{xu} \gamma_x \eta_x) \, I_{d_x} + \tau_x \nabla_x^2[\ell] & 4\gamma_x \tau_u \eta_x I_{d_x} + \tau_{xu} \nabla_x^2[\ell] \end{pmatrix}.$$

*Proof.* Recall that $\mathcal{H}_t := \sqrt{\frac{q_t}{p_{\theta_t}}}$. We begin by applying the quotient rule to obtain the fact that

$$\nabla_{(x,u)} \left( \frac{\nabla_{(x,u)}^* T_{(x,u)} \nabla_{(x,u)} \mathcal{H}_t}{\mathcal{H}_t} \right)$$

$$= \frac{\nabla_{(x,u)} \nabla_{(x,u)}^* \left[T_{(x,u)} \nabla_{(x,u)} \mathcal{H}_t\right]}{\mathcal{H}_t} - \frac{\nabla_{(x,u)} \mathcal{H}_t \nabla_{(x,u)}^* \left[T_{(x,u)} \nabla_{(x,u)} \mathcal{H}_t\right]}{\mathcal{H}_t^2}$$

Then, we can write (25) as

$$(25) = -8\left\langle \frac{\nabla_{(x,u)}\nabla_{(x,u)}^*\left[T_{(x,u)}\nabla_{(x,u)}\mathcal{H}_t\right]}{\mathcal{H}_t}, \begin{pmatrix} 0_{d_x} & -I_{d_x} \\ I_{d_x} & \gamma_x I_{d_x} \end{pmatrix}\nabla_{(x,u)}\log\mathcal{H}_t\right\rangle_{q_t} \tag{37}$$

$$+ 8\left\langle \frac{\nabla_{(x,u)}\mathcal{H}_t\nabla_{(x,u)}^*\left[T_{(x,u)}\nabla_{(x,u)}\mathcal{H}_t\right]}{\mathcal{H}_t^2}, \begin{pmatrix} 0_{d_x} & -I_{d_x} \\ I_{d_x} & \gamma_x I_{d_x} \end{pmatrix}\nabla_{(x,u)}\log\mathcal{H}_t\right\rangle_{q_t} \tag{38}$$

We can simplify the terms individually given in Proposition Proposition F.13 and Proposition F.12 for (37) and (38) Hence, we can write (25) equivalently the following:

$$(25) = 16\gamma_x\left\langle \nabla_{(x,u)}\nabla_u\left[\mathcal{H}_t\right], T_{(x,u)}\frac{\nabla_{(x,u)}\mathcal{H}_t}{\mathcal{H}_t}\otimes\nabla_u\mathcal{H}_t\right\rangle_{\rho_t} \tag{39a}$$

$$- 8\gamma_x\left\langle \frac{\nabla_{(x,u)}\mathcal{H}_t}{\mathcal{H}_t}\otimes\nabla_u\mathcal{H}_t, T_{(x,u)}\frac{\nabla_{(x,u)}\mathcal{H}_t}{\mathcal{H}_t}\otimes\nabla_u\mathcal{H}_t\right\rangle_{\rho_t} \tag{39b}$$

$$- 8\gamma_x\left\langle \nabla_{(x,u)}\nabla_u\left[\mathcal{H}_t\right], T_{(x,u)}\nabla_{(x,u)}\nabla_u\left[\mathcal{H}_t\right]\right\rangle_{\rho_t} \tag{39c}$$

$$- 2\tau_{xu}\eta_x\|\nabla_x\mathcal{H}_t\|_{\rho_t}^2 \tag{39d}$$

$$- 4\left\langle \nabla_x\mathcal{H}_t, \left[(\tau_u\eta_x + \tau_{xu}\gamma_x\eta_x)I_{d_x} + \tau_x\nabla_x^2\left[\ell\right]\right]\nabla_u\mathcal{H}_t\right\rangle_{\rho_t} \tag{39e}$$

$$- 8\left\langle \nabla_u\mathcal{H}_t, \left[\tau_u\eta_x\gamma_x I_{d_x} + \frac{\tau_{xu}}{4}\nabla_x^2\left[\ell\right]\right]\nabla_u\mathcal{H}_t\right\rangle_{\rho_t}. \tag{39f}$$

We can rewrite the sum of (39a), (39b), (39c) as

(39a) + (39b) + (39c)

$$= -8\gamma_x\left\langle \nabla_{(x,u)}\nabla_u\left[\mathcal{H}_t\right] - \frac{\nabla_{(x,u)}\mathcal{H}_t}{\mathcal{H}_t}\otimes\nabla_u\mathcal{H}_t, T_{(x,u)}\left(\nabla_{(x,u)}\nabla_u\left[\mathcal{H}_t\right] - \frac{\nabla_{(x,u)}\mathcal{H}_t}{\mathcal{H}_t}\otimes\nabla_u\mathcal{H}_t\right)\right\rangle_{\rho_t}.$$

Since we have that $\partial_x\partial_u\mathcal{H}_t = \partial_x(\mathcal{H}_t\partial_u\log\mathcal{H}_t) = (\partial_x\mathcal{H}_t)(\partial_u\log\mathcal{H}_t) + \mathcal{H}_t(\partial_x\partial_u\log\mathcal{H}_t)$, we can write the above as

$$(39a) + (39b) + (39c) = -8\gamma_x\left\langle \nabla_{(x,u)}\nabla_u\left[\log\mathcal{H}_t\right], T_{(x,u)}\nabla_{(x,u)}\nabla_u\left[\log\mathcal{H}_t\right]\right\rangle_{q_t}$$

$$= -2\gamma_x\left\langle \nabla_{(x,u)}\nabla_u\left[\log\frac{q_t}{\rho_t}\right], T_{(x,u)}\nabla_{(x,u)}\nabla_u\left[\log\frac{q_t}{\rho_t}\right]\right\rangle_{q_t}$$

As for the other terms, we have that

$$(39d) + (39e) + (39f) = -4\left\langle \nabla_{(x,u)}\log\mathcal{H}_t, \tilde{\Gamma}_x\nabla_{(x,u)}\log\mathcal{H}_t\right\rangle_{q_t}$$

$$= -\left\langle \nabla_{(x,u)}\log\frac{q_t}{\rho_t}, \tilde{\Gamma}_x\nabla_{(x,u)}\log\frac{q_t}{\rho_t}\right\rangle_{q_t},$$

where

$$\tilde{\Gamma}_x := \frac{1}{2}\begin{pmatrix} \tau_{xu}\eta_x I_{d_x} & (\tau_u\eta_x + \tau_{xu}\gamma_x\eta_x)I_{d_x} + \tau_x\nabla_x^2\ell \\ (\tau_u\eta_x + \tau_{xu}\gamma_x\eta_x)I_{d_x} + \tau_x\nabla_x^2\ell & 4\gamma_x\tau_u\eta_x I_{d_x} + \tau_{xu}\nabla_x^2\ell \end{pmatrix}.$$

We have as desired. □

**Proposition F.12** (Simplifying (38)). *We have*

$$(38) = 16\gamma_x\left\langle \nabla_{(x,u)}\nabla_u\left[\mathcal{H}_t\right], T_{(x,u)}\frac{\nabla_{(x,u)}\mathcal{H}}{\mathcal{H}_t}\otimes\nabla_u\mathcal{H}_t\right\rangle_{\rho_t}$$

$$- 8\gamma_x\left\langle \frac{\nabla_{(x,u)}\mathcal{H}_t}{\mathcal{H}_t}\otimes\nabla_u\mathcal{H}_t, T_{(x,u)}\frac{\nabla_{(x,u)}\mathcal{H}_t}{\mathcal{H}_t}\otimes\nabla_u\mathcal{H}_t\right\rangle_{\rho_t}$$

*Proof.* We have

$$(38) = 8\left\langle \frac{\nabla^*_{(x,u)}\left[T_{(x,u)}\nabla_{(x,u)}\mathcal{H}_t\right]}{\mathcal{H}_t}, \left\langle \nabla_{(x,u)}\mathcal{H}_t, \begin{pmatrix} 0 & -I \\ I & \gamma_x I \end{pmatrix}\nabla_{(x,u)}\mathcal{H}_t\right\rangle\right\rangle_{\rho_t}$$

$$= 8\gamma_x\left\langle \frac{\nabla^*_{(x,u)}\left[T_{(x,u)}\nabla_{(x,u)}\mathcal{H}_t\right]}{\mathcal{H}_t}, \|\nabla_u\mathcal{H}_t\|_2^2\right\rangle_{\rho_t}.$$

Then, using the adjoint of operator $\nabla^*_{(x,u)}$ as was described in Appendix F.4.2, we obtain

$$(38) = 8\gamma_x\left\langle \nabla_{(x,u)}\frac{\|\nabla_u\mathcal{H}_t\|_2^2}{\mathcal{H}_t}, T_{(x,u)}\nabla_{(x,u)}\mathcal{H}_t\right\rangle_{\rho_t}.$$

Then, using the quotient rule, we have that

$$(38) = 8\gamma_x\left\langle \frac{\nabla_{(x,u)}\|\nabla_u\mathcal{H}_t\|^2}{\mathcal{H}_t}, T_{(x,u)}\nabla_{(x,u)}\mathcal{H}_t\right\rangle_{\rho_t} - 8\gamma_x\left\langle \frac{\|\nabla_u\mathcal{H}_t\|_2^2\nabla_{(x,u)}\mathcal{H}_t}{\mathcal{H}_t}, T_{(x,u)}\nabla_{(x,u)}\mathcal{H}_t\right\rangle_{\rho_t}$$

$$= 16\gamma_x\left\langle \frac{\nabla_{(x,u)}\nabla_u\left[\mathcal{H}_t\right]\nabla_u\mathcal{H}_t}{\mathcal{H}_t}, T_{(x,u)}\nabla_{(x,u)}\mathcal{H}_t\right\rangle_{\rho_t} - 8\gamma_x\left\langle \frac{\|\nabla_u\mathcal{H}_t\|^2\nabla_{(x,u)}\mathcal{H}_t}{\mathcal{H}_t^2}, T_{(x,u)}\nabla_{(x,u)}\mathcal{H}_t\right\rangle_{\rho_t}.$$

Finally, we obtain

$$(38) = 16\gamma_x\left\langle \nabla_{(x,u)}\nabla_u\left[\mathcal{H}_t\right], T_{(x,u)}\frac{\nabla_{(x,u)}\mathcal{H}_t}{\mathcal{H}_t}\otimes\nabla_u\mathcal{H}_t\right\rangle_{\rho_t}$$

$$- 8\gamma_x\left\langle \frac{\nabla_{(x,u)}\mathcal{H}_t}{\mathcal{H}_t}\otimes\nabla_u\mathcal{H}_t, T_{(x,u)}\frac{\nabla_{(x,u)}\mathcal{H}_t}{\mathcal{H}_t}\otimes\nabla_u\mathcal{H}_t\right\rangle_{\rho_t}$$

where $\otimes$ denotes the outer product, and we use the fact that

$$\|\nabla_u\mathcal{H}_t\|\nabla_{(x,u)}\mathcal{H}_t = \left(\nabla_{(x,u)}\mathcal{H}_t\otimes\nabla_u\mathcal{H}_t\right)\nabla_u\mathcal{H}_t.$$

$\square$

**Proposition F.13** (Simplifying (37))**.** *We have*

$$(37) = -8\left\langle \nabla_{(x,u)}\nabla_u\left[\mathcal{H}_t\right], T_{(x,u)}\nabla_{x,u}\nabla_u\left[\mathcal{H}_t\right]\right\rangle_{\rho_t}$$

$$- 2\tau_{xu}\eta_x\|\nabla_x\mathcal{H}_t\|_{\rho_t}^2$$

$$- 8\left\langle \nabla_u\mathcal{H}_t, \left(\tau_x\eta_x\gamma_x I_{d_x} + \frac{\tau_{xu}}{4}\nabla_x^2\log\rho_t\right)\nabla_u\mathcal{H}_t\right\rangle_{\rho_t}$$

$$- 4\left\langle \nabla_x\mathcal{H}_t, \left(\left[\tau_x\eta_x + \tau_{xu}\gamma_x\eta_x\right]I_{d_x} + \tau_x\nabla_x^2\log\rho_t\right)\nabla_u\mathcal{H}_t\right\rangle_{\rho_t},$$

*where $\rho := \rho_{\theta,\eta_x}$ (also similarly for $\rho_t$).*

*Proof.* We have

$$(37) = -8\left\langle \nabla_{(x,u)}\nabla^*_{(x,u)}\left[T_{(x,u)}\nabla_{(x,u)}\mathcal{H}_t\right], \begin{pmatrix} 0_{d_x} & -I_{d_x} \\ I_{d_x} & \gamma_x I_{d_x} \end{pmatrix}\nabla_{(x,u)}\mathcal{H}_t\right\rangle_{\rho_t}.$$

Since we have

$$\nabla^*_{(x,u)}\left[T_{(x,u)}\nabla_{(x,u)}\mathcal{H}_t\right] = \nabla^*_x\left[\tau_x\nabla_x\mathcal{H}_t + \frac{\tau_{xu}}{2}\nabla_u\mathcal{H}_t\right] + \nabla^*_u\left[\frac{\tau_{xu}}{2}\nabla_x\mathcal{H}_t + \tau_u\nabla_u\mathcal{H}_t\right],$$

we obtain

$$(37) = -4\tau_{xu}\left\langle \nabla_{(x,u)}\left[\nabla^*_x\nabla_u\mathcal{H}_t + \nabla^*_u\nabla_x\mathcal{H}_t\right], \begin{pmatrix} 0_{d_x} & -I_{d_x} \\ I_{d_x} & \gamma_x I_{d_x} \end{pmatrix}\nabla_{(x,u)}\mathcal{H}_t\right\rangle_{\rho_t} \tag{40a}$$

$$- 8\tau_x\left\langle \nabla_{(x,u)}\left[\nabla^*_x\left(\nabla_x\mathcal{H}_t\right)\right], \begin{pmatrix} 0_{d_x} & -I_{d_x} \\ I_{d_x} & \gamma_x I_{d_x} \end{pmatrix}\nabla_{(x,u)}\mathcal{H}_t\right\rangle_{\rho_t} \tag{40b}$$

$$- 8\tau_u\left\langle \nabla_{(x,u)}\left[\nabla^*_u\left(\nabla_u\mathcal{H}_t\right)\right], \begin{pmatrix} 0_{d_x} & -I_{d_x} \\ I_{d_x} & \gamma_x I_{d_x} \end{pmatrix}\nabla_{(x,u)}\mathcal{H}_t\right\rangle_{\rho_t} \tag{40c}$$

In the following Proposition F.14, F.15 and F.16 we deal with the respective terms:

**Proposition F.14** (Simplifying (40a)). *We have*

$$
\begin{aligned}
(40a) = (43) + (44) = & -2\tau_{xu}\eta_x \|\nabla_x \mathcal{H}_t\|_{\rho_t}^2 \\
& -2\tau_{xu} \left\langle \nabla_u \mathcal{H}_t, \nabla_x^2 [\ell] \, \nabla_u \mathcal{H}_t \right\rangle_{\rho_t} \\
& -8\gamma_x \tau_{xu} \left\langle \nabla_u^2 [\mathcal{H}_t], \nabla_u \nabla_x [\mathcal{H}_t] \right\rangle_{\rho_t} \\
& -4\gamma_x \eta_x \tau_{xu} \left\langle \nabla_x \mathcal{H}_t, \nabla_u \mathcal{H}_t \right\rangle_{\rho_t}.
\end{aligned}
$$

**Proposition F.15** (Simplifying (40b)). *We have*

$$
(40b) = -8\tau_x \gamma_x \|\nabla_x \nabla_u \mathcal{H}_t\|_{\rho_t}^2 - 4\tau_x \left\langle \nabla_x \mathcal{H}_t, \nabla_x^2 [\ell] \, \nabla_u \mathcal{H}_t \right\rangle_{\rho_t},
$$

**Proposition F.16** (Simplifying (40c)).

$$
(40c) = -8\tau_u \gamma_x \left\| \nabla_u^2 \mathcal{H}_t \right\|_{\rho_t}^2 - 8\tau_u \eta_x \gamma_x \|\nabla_u \mathcal{H}_t\|_{\rho_t}^2 - 4\tau_u \eta_x \langle \nabla_x \mathcal{H}_t, \nabla_u \mathcal{H}_t \rangle_{\rho_t}.
$$

Their proofs can be found in Appendix F.4.5, Appendix F.4.5, Appendix F.4.5.

Thus, summing up the results of Proposition F.14, Proposition F.15 and Proposition F.16, we obtain

$$
\begin{aligned}
(37) = & (40a) + (40b) + (40c) \\
= & -8\tau_{xu}\gamma_x \left\langle \nabla_u^2 \mathcal{H}_t, \nabla_u \nabla_x [\mathcal{H}_t] \right\rangle_{\rho_t} & (41a) \\
& -8\tau_u \gamma_x \left\| \nabla_u^2 \mathcal{H}_t \right\|_{\rho_t}^2 & (41b) \\
& -8\tau_x \gamma_x \left\langle \nabla_x \nabla_u \mathcal{H}_t, \nabla_x \nabla_u \mathcal{H}_t \right\rangle_{\rho_t} & (41c) \\
& -2\tau_{xu}\eta_x \|\nabla_x \mathcal{H}_t\|_{\rho_t}^2 \\
& -8 \left\langle \nabla_u \mathcal{H}_t, \left( \tau_u \eta_x \gamma_x I_{d_x} + \frac{\tau_{xu}}{4} \nabla_x^2 [\ell] \right) \nabla_u \mathcal{H}_t \right\rangle_{\rho_t} \\
& -4 \left\langle \nabla_x \mathcal{H}_t, \left[ (\tau_u \eta_x + \tau_{xu}\gamma_x \eta_x) I_{d_x} + \tau_x \nabla_x^2 [\ell] \right] \nabla_u \mathcal{H}_t \right\rangle_{\rho_t}
\end{aligned}
$$

Since we can write the following sum equivalently as:

$$
(41a) + (41b) + (41c) = -8\gamma_x \left\langle \nabla_{(x,u)} \nabla_u [\mathcal{H}_t], T_{(x,u)} \nabla_{(x,u)} \nabla_u [\mathcal{H}_t] \right\rangle_{\rho_t},
$$

we obtain as desired. $\qquad\square$

*Proof of Proposition F.14.* We have

$$
\begin{aligned}
(40a) = & -4\tau_{xu} \left\langle \nabla_{(x,u)} \left[ \nabla_x^* \nabla_u \mathcal{H}_t + \nabla_u^* \nabla_x \mathcal{H}_t \right], \begin{pmatrix} 0_{d_x} & 0_{d_x} \\ 0_{d_x} & \gamma_x I_{d_x} \end{pmatrix} \nabla_{(x,u)} \mathcal{H}_t \right\rangle_{\rho_t} \\
& -4\tau_{xu} \left\langle \nabla_{(x,u)} \left[ \nabla_x^* \nabla_u \mathcal{H}_t + \nabla_u^* \nabla_x \mathcal{H}_t \right], \begin{pmatrix} 0_{d_x} & -I_{d_x} \\ I_{d_x} & 0_{d_x} \end{pmatrix} \nabla_{(x,u)} \mathcal{H}_t \right\rangle_{\rho_t} \\
= & -4\gamma_x \tau_{xu} \left\langle \nabla_u \left[ \nabla_x^* \nabla_u \mathcal{H}_t + \nabla_u^* \nabla_x \mathcal{H}_t \right], \nabla_u \mathcal{H}_t \right\rangle_{\rho_t} & (42a) \\
& -4\tau_{xu} \left\langle \nabla_{(x,u)} \left[ \nabla_x^* \nabla_u \mathcal{H}_t + \nabla_u^* \nabla_x \mathcal{H}_t \right], \begin{pmatrix} 0_{d_x} & -I_{d_x} \\ I_{d_x} & 0_{d_x} \end{pmatrix} \nabla_{(x,u)} \mathcal{H}_t \right\rangle_{\rho_t} & (42b)
\end{aligned}
$$

We will deal with (42a) and (42b) separately.

Using the adjoints of $\nabla_u^*, \nabla_u, \nabla_x$, we obtain

$$
(42a) = -4\gamma_x \tau_{xu} \left\langle \nabla_u \mathcal{H}_t, \nabla_x \nabla_u^* \nabla_u \mathcal{H}_t \right\rangle_{\rho_t} - 4\gamma_x \tau_{xu} \left\langle \nabla_x \mathcal{H}_t, \nabla_u \nabla_u^* \nabla_u \mathcal{H}_t \right\rangle_{\rho_t},
$$

From Proposition F.8, we have that $\nabla_x$ commutes with $\nabla_u \nabla_u^*$

$$(42a) = -4\gamma_x \tau_{xu} \left\langle \nabla_u \mathcal{H}_t, \nabla_u^* \nabla_u \nabla_x [\mathcal{H}_t] \right\rangle_{\rho_t} - 4\gamma_x \tau_{xu} \left\langle \nabla_x \mathcal{H}_t, \nabla_u \nabla_u^* \nabla_u \mathcal{H}_t \right\rangle_{\rho_t},$$

We can write this in terms of the commutator of $\nabla_u$ and $\nabla_u^*$ defined as $[\nabla_u, \nabla_u^*]v = \nabla_u \nabla_u^* v - \nabla_u^* \nabla_u v$. Thus, we have

$$
\begin{aligned}
(42a) = & -8\gamma_x \tau_{xu} \left\langle \nabla_u \mathcal{H}_t, \nabla_u^* \nabla_u \nabla_x [\mathcal{H}_t] \right\rangle_{\rho_t} - 4\gamma_x \tau_{xu} \left\langle \nabla_x \mathcal{H}_t, [\nabla_u, \nabla_u^*] \nabla_u \mathcal{H}_t \right\rangle_{\rho_t} \\
= & -8\gamma_x \tau_{xu} \left\langle \nabla_u^2 [\mathcal{H}_t], \nabla_u \nabla_x [\mathcal{H}_t] \right\rangle_{\rho_t} - 4\gamma_x \eta_x \tau_{xu} \left\langle \nabla_x \mathcal{H}_t, \nabla_u \mathcal{H}_t \right\rangle_{\rho_t}.
\end{aligned}
\tag{43}
$$

As for (42b), expanding the gradient term $\nabla \mathcal{H} = \frac{\mathcal{H}}{2}(\nabla \log q - \nabla \log \rho)$, we obtain

$$
\begin{aligned}
(42b) = & -4\tau_{xu} \left\langle \nabla_{(x,u)} \left[ \nabla_x^* \nabla_u \mathcal{H}_t + \nabla_u^* \nabla_x \mathcal{H}_t \right], \begin{pmatrix} 0_{d_x} & -I_{d_x} \\ I_{d_x} & 0_{d_x} \end{pmatrix} \nabla_{(x,u)} \mathcal{H}_t \right\rangle_{\rho_t} \\
= & -2\tau_{xu} \left\langle \nabla_x^* \nabla_u \mathcal{H}_t + \nabla_u^* \nabla_x \mathcal{H}_t, \nabla_{(x,u)}^* \left[ \mathcal{H}_t Q \begin{pmatrix} -\nabla_x \log \rho_\theta \\ -\nabla_u \log \rho_\theta \end{pmatrix} \right] \right\rangle_{\rho_t}.
\end{aligned}
$$

Writing this in terms of $B$ (see Appendix F.4.4) $B[\mathcal{H}](x,u) := \left\langle \nabla_x \mathcal{H}(x,u), \eta_x u \right\rangle + \left\langle \nabla_u \mathcal{H}(x,u), \nabla_x \log \rho(x,u) \right\rangle$, we can write

$$(42b) = -2\tau_{xu} \left\langle \nabla_{(x,u)}^* \nabla_{(u,x)} \mathcal{H}_t, B[\mathcal{H}_t] \right\rangle_{\rho_t}.$$

Using the fact that the adjoint of $B$ denoted by $B^*$ w.r.t. the inner product $\langle \cdot, \cdot \rangle_\rho$ is $B^* = -B$ (see F.9) we obtain

$$
\begin{aligned}
(42b) = & -2\tau_{xu} \left\langle \nabla_{(u,x)} \mathcal{H}_t, \nabla_{(x,u)} B[\mathcal{H}_t] \right\rangle_{\rho_t} \\
= & -2\tau_{xu} \left( \left\langle \nabla_x \mathcal{H}_t, \nabla_u B[\mathcal{H}_t] \right\rangle_{\rho_t} + \left\langle \nabla_u \mathcal{H}_t, \nabla_x B[\mathcal{H}_t] \right\rangle_{\rho_t} \right) \\
= & -2\tau_{xu} \left( \left\langle \nabla_x \mathcal{H}_t, \nabla_u B[\mathcal{H}_t] \right\rangle_{\rho_t} + \left\langle \nabla_u \mathcal{H}_t, B \nabla_x [\mathcal{H}_t] \right\rangle_{\rho_t} + \left\langle \nabla_u \mathcal{H}_t, [\nabla_x, B][\mathcal{H}_t] \right\rangle_{\rho_t} \right) \\
= & -2\tau_{xu} \left( \left\langle \nabla_x \mathcal{H}_t, \nabla_u B[\mathcal{H}_t] \right\rangle_{\rho_t} - \left\langle B \nabla_u \mathcal{H}_t, \nabla_x [\mathcal{H}_t] \right\rangle_{\rho_t} + \left\langle \nabla_u \mathcal{H}_t, [\nabla_x, B][\mathcal{H}_t] \right\rangle_{\rho_t} \right) \\
= & -2\tau_{xu} \left( \left\langle \nabla_x \mathcal{H}_t, [\nabla_u, B][\mathcal{H}_t] \right\rangle_{\rho_t} + \left\langle \nabla_u \mathcal{H}_t, [\nabla_x, B][\mathcal{H}_t] \right\rangle_{\rho_t} \right).
\end{aligned}
$$

From Proposition F.10, we have $[\nabla_u, B][\mathcal{H}_t] = \eta_x \nabla_x \mathcal{H}_t$ and $[\nabla_x, B][\mathcal{H}_t] = \nabla_x^2[\ell] \nabla_u \mathcal{H}_t$, and so

$$(42b) = -2\tau_{xu} \eta_x \|\nabla_x \mathcal{H}_t\|_{\rho_t}^2 - 2\tau_{xu} \left\langle \nabla_u \mathcal{H}_t, \nabla_x^2[\ell], \nabla_u \mathcal{H}_t \right\rangle_{\rho_t} \tag{44}$$

Thus, summing up (43) and (44), we have as desired

$$
\begin{aligned}
(40a) = (43) + (44) = & -2\tau_{xu} \eta_x \|\nabla_x \mathcal{H}_t\|_{\rho_t}^2 \\
& -2\tau_{xu} \left\langle \nabla_u \mathcal{H}_t, \nabla_x^2[\ell] \nabla_u \mathcal{H}_t \right\rangle_{\rho_t} \\
& -8\gamma_x \tau_{xu} \left\langle \nabla_u^2 [\mathcal{H}_t], \nabla_u \nabla_x [\mathcal{H}_t] \right\rangle_{\rho_t} \\
& -4\gamma_x \eta_x \tau_{xu} \left\langle \nabla_x \mathcal{H}_t, \nabla_u \mathcal{H}_t \right\rangle_{\rho_t}.
\end{aligned}
$$

$\square$

*Proof of Proposition F.15.* We have

$$(40b) = -8\tau_x \gamma_x \left\langle \nabla_u \left[ \nabla_x^* (\nabla_x \mathcal{H}_t) \right], \nabla_u \mathcal{H}_t \right\rangle_{\rho_t} - 8\tau_x \left\langle \nabla_x \mathcal{H}_t, \nabla_x B[\mathcal{H}_t] \right\rangle_{\rho_t}.$$

Writing this in terms of the commutator, we get

$$
\begin{aligned}
(40b) = & -8\tau_x \gamma_x \left\langle \nabla_u \left[ \nabla_x^* (\nabla_x \mathcal{H}_t) \right], \nabla_u \mathcal{H}_t \right\rangle_{\rho_t} - 4\tau_x \left\langle \nabla_x \mathcal{H}_t, [\nabla_x, B]\mathcal{H}_t + B \nabla_x \mathcal{H}_t \right\rangle_{\rho_t} \\
= & -8\tau_x \gamma_x \left\langle \nabla_x \nabla_u \mathcal{H}_t, \nabla_x \nabla_u \mathcal{H}_t \right\rangle_{\rho_t} - 4\tau_x \left\langle \nabla_x \mathcal{H}_t, \nabla_x^2[\ell] \nabla_u \mathcal{H}_t \right\rangle_{\rho_t},
\end{aligned}
$$

where, for the first line, we use the fact that $\nabla_x^*$ commutes with $\nabla_u$ (see Proposition F.7). $\square$

*Proof of Proposition F.16.* We begin with

$$(40c) = -8\tau_u\gamma_x \left\langle \nabla_u\nabla_u^*\left[\nabla_u\mathcal{H}_t\right], \nabla_u\mathcal{H}_t\right\rangle_{\rho_t}$$

$$-8\tau_u\left\langle \nabla_u\mathcal{H}_t, \nabla_u\nabla_{(x,u)}^*\left[\begin{pmatrix} 0_{d_x} & -I_{d_x} \\ I_{d_x} & 0_{d_x} \end{pmatrix}\nabla_{(x,u)}\mathcal{H}_t\right]\right\rangle_{\rho_t}$$

$$= -8\tau_u\gamma_x\left\langle \nabla_u\nabla_u^*\left[\nabla_u\mathcal{H}_t\right], \nabla_u\mathcal{H}_t\right\rangle_{\rho_t} - 4\tau_u\left\langle \nabla_u\mathcal{H}_t, \nabla_u B[\mathcal{H}_t]\right\rangle_{\rho_t}.$$

Writing in terms of the commutator $[\cdot,\cdot]$ and $B$, we have equivalently

$$(40c) = -8\tau_u\gamma\left\langle \nabla_u\nabla_u\left[\mathcal{H}_t\right], \nabla_u\nabla_u\left[\mathcal{H}_t\right]\right\rangle_{\rho_t} - 8\tau_u\gamma_x\left\langle [\nabla_u, \nabla_u^*]\nabla_u\mathcal{H}_t, \nabla_u\mathcal{H}_t\right\rangle_{\rho_t}$$

$$-4\tau_u\left\langle \nabla_u\mathcal{H}_t, B\nabla_u\mathcal{H}_t\right\rangle_{\rho_t} - 4\tau_u\eta_x\left\langle \nabla_x\mathcal{H}_t, \nabla_u\mathcal{H}_t\right\rangle_{\rho_t}.$$

Using the antisymmetric property of $B$ and (from Proposition F.6) that $[\nabla_u, \nabla_u^*]\nabla_u f = \eta_x\nabla_u f$, we obtain

$$(40c) = -8\tau_u\gamma\left\langle \nabla_u^2\left[\mathcal{H}_t\right], \nabla_u^2\left[\mathcal{H}_t\right]\right\rangle_{\rho_t}$$

$$-8\tau_u\eta_x\gamma_x\left\langle \nabla_u\mathcal{H}_t, \nabla_u\mathcal{H}_t\right\rangle_{\rho_t}$$

$$-4\tau_u\eta_x\left\langle \nabla_x\mathcal{H}_t, \nabla_u\mathcal{H}_t\right\rangle_{\rho_t}.$$

$\square$

### F.4.6 BOUNDING THE CROSS TERMS (21) AND (26).

**Proposition F.17.** *For all $\varepsilon > 0$, we have that*

$$(21) + (26) \le \varepsilon\left\langle \nabla_{(\theta,m)}\mathcal{F}_t, \Gamma_\times\nabla_{(\theta,m)}\mathcal{F}_t\right\rangle + \frac{1}{\varepsilon}\|\nabla_{(x,u)}\delta_q\mathcal{F}_t\|_{q_t}^2,$$

*where* $\Gamma_\times = K^2\begin{pmatrix} \tau_\theta^2 I_{d_\theta} & \tau_\theta\left(\frac{\tau_{xu}+\tau_{\theta m}}{2}\right)I_{d_\theta} \\ \tau_\theta\left(\frac{\tau_{xu}+\tau_{\theta m}}{2}\right)I_{d_\theta} & \left(\left(\frac{\tau_{xu}+\tau_{\theta m}}{2}\right)^2 + \tau_x^2\right)I_{d_\theta} \end{pmatrix}.$

*Proof.* Because $T_{(x,u)}$ is symmetric,

$$(21) = -2\left\langle A_t\nabla_{(\theta,m)}\mathcal{F}_t, \nabla_{(x,u)}\delta_q\mathcal{F}_t\right\rangle_{q_t}, \quad (26) = 2\left\langle A_t'\nabla_{(\theta,m)}\mathcal{F}_t, \nabla_{(x,u)}\delta_q\mathcal{F}_t\right\rangle_{q_t},$$

where

$$A_t := T_{(x,u)}\nabla_{(x,u)}\nabla_{(\theta,m)}\left[\delta_q\mathcal{F}_t\right]\begin{pmatrix} 0_{d_\theta} & -I_{d_\theta} \\ I_{d_\theta} & \gamma_\theta I_{d_\theta} \end{pmatrix},$$

$$A_t' := \begin{pmatrix} 0_{d_x} & I_{d_x} \\ -I_{d_x} & \gamma_x I_{d_x} \end{pmatrix}\nabla_{(x,u)}\nabla_{(\theta,m)}\left[\log\rho_{\theta_t,\eta_x}\right]T_{(\theta,m)}.$$

Given that

$$\nabla_{(x,u)}\nabla_{(\theta,m)}\left[\delta_q\mathcal{F}_t\right] = -\nabla_{(x,u)}\nabla_{(\theta,m)}\left[\log\rho_{\theta_t,\eta_x}\right] = -\begin{pmatrix} \nabla_x\nabla_\theta\left[\log\rho_{\theta_t}\right] & 0_{d_x\times d_\theta} \\ 0_{d_x\times d_\theta} & 0_{d_x\times d_\theta} \end{pmatrix},$$

we can re-write $A_t$ and $A_t'$ as

$$A_t = \begin{pmatrix} 0_{d_x\times d_\theta} & \tau_x\nabla_x\nabla_\theta\left[\log\rho_{\theta_t}\right] \\ 0_{d_x\times d_\theta} & \frac{\tau_{xu}}{2}\nabla_x\nabla_\theta\left[\log\rho_{\theta_t}\right] \end{pmatrix}, \quad A_t' = -\begin{pmatrix} 0_{d_x\times d_\theta} & 0_{d_x\times d_\theta} \\ \tau_\theta\nabla_x\nabla_\theta\left[\log\rho_{\theta_t}\right] & \frac{\tau_{\theta m}}{2}\nabla_x\nabla_\theta\left[\log\rho_{\theta_t}\right] \end{pmatrix}.$$

Hence,

$$(21) + (26) = 2\left\langle A_t\nabla_{(\theta,m)}\mathcal{F}_t, \nabla_{(x,u)}\delta_q\mathcal{F}_t\right\rangle_{q_t}$$

where

$$\tilde{A}_t := A_t' - A_t = -\begin{pmatrix} 0 & \tau_x \nabla_x \nabla_\theta \left[\log \rho_{\theta_t}\right] \\ \tau_\theta \nabla_x \nabla_\theta \left[\log \rho_{\theta_t}\right] & \frac{(\tau_{xu} + \tau_{\theta m})}{2} \nabla_x \nabla_\theta \left[\log \rho_{\theta_t}\right] \end{pmatrix}.$$

Fix any $\varepsilon > 0$. Applying the Cauchy-Schwarz inequality and Young's inequality, we obtain that

$$\begin{aligned}
(21) + (26) &\le 2 \left|\left| \tilde{A}_t \nabla_{(\theta,m)} \mathcal{F}_t \right|\right|_{q_t} \left|\left| \nabla_{(x,u)} \delta_q \mathcal{F}_t \right|\right|_{q_t} \\
&\le \varepsilon \left|\left| \tilde{A}_t \nabla_{(\theta,m)} \mathcal{F}_t \right|\right|_{q_t}^2 + \frac{1}{\varepsilon} \left|\left| \nabla_{(x,u)} \delta_q \mathcal{F}_t \right|\right|_{q_t}^2 .
\end{aligned} \tag{45}$$

But,

$$\begin{aligned}
\|\tilde{A}_t \nabla_{(\theta,m)} \mathcal{F}_t\|_{q_t}^2 &= \|\tau_x \nabla_x \nabla_\theta \left[\log \rho_{\theta_t}\right] \nabla_m \mathcal{F}_t\|_{q_t}^2 \\
&\quad + \left\|\nabla_x \nabla_\theta \left[\log \rho_{\theta_t}\right] \left[\tau_\theta \nabla_\theta \mathcal{F}_t + \frac{\tau_{xu} + \tau_{\theta m}}{2} \nabla_m \mathcal{F}_t\right]\right\|_{q_t}^2.
\end{aligned}$$

Given that $\nabla_\theta \mathcal{F}(z)$ and $\nabla_m \mathcal{F}(z)$ are constant in $x$, the above reads

$$\begin{aligned}
\|\tilde{A}_t \nabla_{(\theta,m)} \mathcal{F}_t\|_{q_t}^2 &= \|\nabla_x \langle \nabla_\theta \log \rho_{\theta_t}, \tau_x \nabla_m \mathcal{F}_t\rangle\|_{q_t}^2 \\
&\quad + \left\|\nabla_x \left\langle \nabla_\theta \log \rho_\theta, \left(\tau_\theta \nabla_\theta \mathcal{F}_t + \frac{\tau_{xu} + \tau_{\theta m}}{2} \nabla_m \mathcal{F}_t\right)\right\rangle\right\|_{q_t}^2.
\end{aligned} \tag{46}$$

Fix any $\theta, v$ in $\mathbb{R}^{d_\theta}$. Because $\nabla_{(\theta,x)} \ell$ is $K$-Lipschitz (Assumption 2), the function

$$f_{v,\theta}(x) := \langle \nabla_\theta \log \rho_\theta(x), v\rangle = \langle \nabla_\theta \ell(\theta, x), v\rangle \quad \forall x \in \mathbb{R}^{d_x},$$

is $(K\|v\|)$-Lipschitz: by the Cauchy-Schwarz inequality,

$$\begin{aligned}
f_{v,\theta}(x) - f_{v,\theta}(x') &= \langle \nabla_\theta \log \ell(\theta, x) - \nabla_\theta \ell(\theta, x'), v\rangle \le \|\nabla_\theta \ell(\theta, x) - \nabla_\theta \ell(\theta, x')\|\|v\|, \\
&\le \|\nabla_{(\theta,m)} \ell(\theta, x) - \nabla_{(\theta,m)} \ell(\theta, x')\|\|v\| \le K\|v\|\|x - x'\| \quad \forall x, x' \in \mathbb{R}^{d_x}.
\end{aligned}$$

Recalling that Lipschitz seminorms can be estimated by suprema of the norm of the gradient (e.g., see Van Handel (2014, Lemma 4.3)), we then see that

$$\|\nabla_x f_{v,\theta}\|_{q_t}^2 \le \sup_{x \in \mathbb{R}^{d_x}} \|\nabla_x f_{v,\theta}(x)\|^2 \le K^2 \|v\|^2.$$

Applying the above inequality to (46), we find that

$$\begin{aligned}
\|\tilde{A}_t \nabla_{(\theta,m)} \mathcal{F}_t\|_{q_t}^2 &\le K^2 \left[\|\tau_x \nabla_m \mathcal{F}_t\|^2 + \left\|\tau_\theta \nabla_\theta \mathcal{F}_t + \left(\frac{\tau_{xu} + \tau_{\theta m}}{2}\right) \nabla_m \mathcal{F}_t\right\|^2\right] \\
&= \left\langle \nabla_{(\theta,m)} \mathcal{F}_t, \Gamma_\times \nabla_{(\theta,m)} \mathcal{F}_t\right\rangle.
\end{aligned}$$

Plugging the above into (45) yields the desired inequality. $\qquad \square$

### F.4.7 Positive Semi-definiteness Conditions

In this section, we establish sufficient conditions to ensure that a matrix with the form described in (47).

**Proposition F.18.** *Given $\alpha, \beta, \kappa, a, c \in \mathbb{R}$, let $A$ be a symmetric matrix with the following form:*

$$A = \begin{pmatrix} \alpha I & \beta I + aH \\ \beta I + aH & \kappa I + cH \end{pmatrix}, \tag{47}$$

*where $H$ is a symmetric matrix that satisfies $-KI \preceq H \preceq KI$ for some $K > 0$. If $A$ satisfies the following conditions:*

$$\begin{cases} -cK - \alpha - \kappa \le 0 & c \le 0, \\ cK - \alpha - \kappa \le 0 & c > 0, \end{cases}$$
$$a^2 K^2 - (c\alpha - 2\beta a)K - \alpha\kappa + \beta^2 \le 0,$$
$$a^2 K^2 + (c\alpha - 2\beta a)K - \alpha\kappa + \beta^2 \le 0,$$

*then, we have that $A$ is positive semi-definite.*

*Proof.* We prove this by showing that if the conditions are satisfied, then the eigenvalues of $A$ are non-negative.

The eigenvalues $l$ of $A$ satisfy its characteristic equation

$$\det(A - lI) = 0.$$

We have that

$$\det(A - lI) = \det((\alpha - l)(\kappa - l)I + c(\alpha - l)H - (\beta I + aH)^2),$$

where we use the fact that $(\kappa I + cH)$ and $(\beta I + aH)$ are symmetric matrices and so their multiplication commutes (e.g., see Silvester (2000)).

Let $H = U\Lambda U^\top$ be the eigenvalue decomposition of $H$, then observe that

$$\det(A - lI) = \det(U(((\alpha - l)(\kappa - l) - \beta^2)I + (c(\alpha - l) - 2\beta a)\Lambda - a^2\Lambda^2)U^\top).$$

Hence, we obtain for each eigenvalue $\Lambda_i$ of $H$ the following constraints:

$$((\alpha - l)(\kappa - l) - \beta^2)I + (c(\alpha - l) - 2\beta a)\Lambda_i - a^2\Lambda_i^2 = 0.$$

These constraints can be written equivalently as

$$l^2 + (-c\Lambda_i - \alpha - \kappa)\, l + \left(\alpha\kappa - \beta^2 - a^2\Lambda_i^2 + (c\alpha - 2\beta a)\Lambda_i\right) = 0.$$

Since the equality constraint is a quadratic function in $l$, we can utilize the quadratic formula to solve for $l$. To ensure that $l$ is positive, we require that

$$-c\Lambda_i - \alpha - \kappa \leq 0, \tag{48}$$

$$a^2\Lambda_i^2 - (c\alpha - 2\beta a)\Lambda_i - \alpha\kappa + \beta^2 \leq 0, \tag{49}$$

for all $\Lambda_i$. Since we have that $-KI \preceq H \preceq KI$, it follows from the min-max theorem that we have for all $i$ the eigenvalues satisfy $\Lambda_i \in [-K, K]$

Since (48) is a linear function and (49) is a quadratic function in $\Lambda_i$, we only require that the inequalities are satisfied at the end points. Namely, we obtain the following conditions

$$\begin{cases} -cK - \alpha - \kappa \leq 0 & c \leq 0, \\ cK - \alpha - \kappa \leq 0 & c > 0, \end{cases}$$

$$a^2K^2 - (c\alpha - 2\beta a)K - \alpha\kappa + \beta^2 \leq 0,$$

$$a^2K^2 + (c\alpha - 2\beta a)K - \alpha\kappa + \beta^2 \leq 0.$$

This concludes the proof. $\qquad\square$

## G   EXISTENCE AND UNIQUENESS OF STRONG SOLUTIONS TO (2)

In this section, we show the existence and uniqueness of the McKean-Vlasov SDE (2) under Lipschitz assumption 2. The structure is as follows:

1. We begin by showing that $\mathcal{F}$ has Lipschitz $\theta$-gradients (Proposition G.1).
2. Then, we show that the drift is Lipschitz (Proposition G.2).
3. Finally, we prove the existence and uniqueness (Proposition 3.1).

In this section, we write the SDE (2) equivalently as

$$\mathrm{d}(\vartheta, \Upsilon)_t = b(\Upsilon_t, \vartheta_t, \mathrm{Law}(\Upsilon_t), \vartheta_t)\mathrm{d}t + \boldsymbol{\beta}\mathrm{d}W_t, \tag{50}$$

where $\vartheta_t = (\theta_t, m_t) \in \mathbb{R}^{2d_\theta}$, $\Upsilon_t = (X_t, U_t) \in \mathbb{R}^{2d_x}$, , $\boldsymbol{\beta} = \sqrt{2}[0_{2d_\theta}, 1_{2d_x}]^\top$ and $b : \mathbb{R}^{2d_x} \times \mathbb{R}^{2d_\theta} \times \mathcal{P}(\mathbb{R}^{2d_x}) \times \mathbb{R}^{2d_\theta} \to \mathbb{R}^{2d_\theta} \times \mathbb{R}^{2d_x}$ defined as

$$b((x, u), (\theta, m), q, (\theta', m')) = \begin{pmatrix} \eta_\theta m \\ -\gamma_\theta \eta_\theta m' - \nabla_\theta \mathcal{F}(\theta', q) \\ \eta_x u \\ -\gamma_x \eta_x u + \nabla_x \ell(\theta, x). \end{pmatrix}$$

We now prove that $\nabla_\theta \mathcal{F}$ is Lipschitz. Since the gradients do not depend on the momentum parameter $m$, we will drop the dependence on $m$ for brevity.

**Proposition G.1** ($\mathcal{F}$ has Lipschitz $\theta$-gradient)**.** *Under Assumption 2, we have that $\mathcal{F}$ is Lipschitz,i.e., there exist a constant $K_\mathcal{F} > 0$ such that the following inequality holds:*

$$\|\nabla_\theta \mathcal{F}(\theta, q) - \nabla_\theta \mathcal{F}(\theta', q')\| \leq K_\mathcal{F}(\|\theta - \theta'\| + W_1(q, q')),$$

*for all $\theta, \theta' \in \mathbb{R}^{d_\theta}$ and $q, q' \in \mathcal{P}(\mathbb{R}^{2d_x})$.*

*Proof.* From the definition, and adding and subtracting the same quantities and triangle inequality, we obtain

$$\|\nabla_\theta \mathcal{F}(\theta, q) - \nabla_\theta \mathcal{F}(\theta', q')\| \leq \|\nabla_\theta \mathcal{F}(\theta, q) - \nabla_\theta \mathcal{F}(\theta', q)\|$$
$$+ \|\nabla_\theta \mathcal{F}(\theta', q) - \nabla_\theta \mathcal{F}(\theta', q')\|$$

We treat the terms on the RHS separately. For the first, from Jensen's inequality, we obtain

$$\|\nabla_\theta \mathcal{F}(\theta, q) - \nabla_\theta \mathcal{F}(\theta', q)\| = \left\| \int \nabla_\theta \ell(\theta, x) - \nabla_\theta \ell(\theta', x) \, q_X(\mathrm{d}x) \right\|$$
$$\leq \int \|\nabla_\theta \ell(\theta, x) - \nabla_\theta \ell(\theta', x)\| \, q_X(\mathrm{d}x)$$
$$\leq K\|\theta - \theta'\|.$$

As for the other term, we have

$$\|\nabla_\theta \mathcal{F}(\theta', q) - \nabla_\theta \mathcal{F}(\theta', q')\| \leq \left\| \int \nabla_\theta \ell(\theta', x)(q - q')(x, u)\mathrm{d}x\mathrm{d}u \right\|$$
$$\leq K \int \frac{1}{K} \|\nabla_\theta \ell(\theta', x)\| \, |q - q'|(x, u)\mathrm{d}x\mathrm{d}u$$
$$\overset{(a)}{\leq} 2KW_1(q, q').$$

For $(a)$, we use the fact that $x \mapsto \|\nabla_\theta \ell(\theta', x)\|$ is $K$-Lipschitz (under assumption 2), (and so the map $x \mapsto \frac{1}{K}\|\nabla_\theta \ell(\theta', x)\|$ is 1-Lipschitz), from the dual representation of $W_1$ we have as desired.

To conclude, by combining the two bounds, we have shown that $\nabla_\theta \mathcal{F}$ is Lipschitz with constant $K_\mathcal{F} = 2K$. $\qquad\square$

We now prove that the drift of the SDE (50) is Lipschitz.

**Proposition G.2** (Lipschitz Drift)**.** *Under Assumption 2, the drift $b$ is Lipschitz, i.e., there is some constant $K_b > 0$ such that the following inequality holds:*

$$\|b(\Upsilon, \vartheta_1, q, \vartheta_2) - b(\Upsilon', \vartheta_1', q', \vartheta_2')\| \leq K_b(\|\Upsilon - \Upsilon'\| + \|\vartheta_1 - \vartheta_1'\| + \|\vartheta_2 - \vartheta_2'\| + W_1(q, q')),$$

*for all $\vartheta_1, \vartheta_1', \vartheta_2, \vartheta_2' \in \mathbb{R}^{d_\theta}$, $\Upsilon, \Upsilon' \in \mathbb{R}^{2d_\theta}$, and $q, q' \in \mathcal{P}(\mathbb{R}^{2d_x})$.*

*Proof.* We begin with the definition and applying the triangle inequality to obtain

$$\|b(\Upsilon, \vartheta_1, q, \vartheta_1) - b(\Upsilon', \vartheta_2', q', \vartheta_2')\| \leq \eta_\theta \|m_1 - m_1'\| + \gamma_\theta \eta_\theta \|m_2 - m_2'\|$$
$$+ \|\nabla_\theta \mathcal{F}(\theta_2, q) - \nabla_\theta \mathcal{F}(\theta_2', q')\|$$
$$+ \eta_x(1 + \gamma_x)\|u - u'\| + \|\nabla_x \ell(\theta_1, x) - \nabla_x \ell(\theta_1', x')\|$$
$$\leq \eta_\theta \|m_1 - m_1'\| + \gamma_\theta \eta_\theta \|m_2 - m_2'\|$$
$$+ K_\mathcal{F}(\|\theta_2 - \theta_2'\| + W_1(q, q'))$$
$$+ \eta_x(1 + \gamma_x)\|u - u'\| + K(\|\theta_1 - \theta_1'\| + \|x - x'\|),$$

where we use the Lipschitz assumption 2 of $\ell$ and Proposition G.1. Hence, we have as desired with Lipschitz constant $K_b = \max\{\eta_\theta, \gamma_\theta \eta_\theta, \eta_x(1 + \gamma_x), K_\mathcal{F}\}$. $\qquad\square$

*Proof of Proposition 3.1.* The proof follows similarly to Carmona (2016, Theorem 1.7) and with suitable generalizations to the product space.

Fix some $\nu \in C([0,T], \mathbb{R}^{2d_\theta} \times \mathcal{P}(\mathbb{R}^{2d_x}))$. We denoted by $\nu^\vartheta$ and $\nu^\Upsilon$ the projection to the $\mathbb{R}^{2d_\theta}$ and $\mathcal{P}(\mathbb{R}^{2d_x}))$ components respectively. Consider substituting $\nu_t$ into (50) in place of the $\mathrm{Law}(\Upsilon_t)$ and $\vartheta_t$, from Carmona (2016, Theorem 1.2), we have existence and uniqueness of the strong solution for some initial point $(\vartheta, \Upsilon)_0$. More explicitly, we have

$$(\vartheta, \Upsilon)_t^\nu = (\vartheta, \Upsilon)_0 + \int_0^t b(\Upsilon_t^\nu, \vartheta_t^\nu, \nu_s^\Upsilon, \nu_s^\vartheta)\mathrm{d}s + \int_0^t \beta \mathrm{d}W_t.$$

for $t \in [0,T]$. We define the operator $F_T : C([0,T], \mathbb{R}^{2d_\theta} \times \mathcal{P}(\mathbb{R}^{2d_x})) \to C([0,T], \mathbb{R}^{2d_\theta} \times \mathcal{P}(\mathbb{R}^{2d_x}))$ as

$$F_T : \nu \to (t \mapsto (\vartheta_t^\nu, \mathrm{Law}(\Upsilon_t^\nu))).$$

Clearly, if the process $(\vartheta, \Upsilon)_t$ is a solution to (50) then the function $t \mapsto (\vartheta_t, \mathrm{Law}(\Upsilon_t))$ is a fixed point to the operator $F_T$, and vice versa. Now we establish the existence and uniqueness of the fixpoint of the operator $F_T$.

We begin by endowing the space $\mathbb{R}^{2d_\theta} \times \mathcal{P}(\mathbb{R}^{2d_x})$ with the metric:

$$d((\vartheta, q), (\vartheta', q')) = \sqrt{\|\vartheta - \vartheta'\|^2 + W_2(q, q')^2}.$$

Note that the metric space $(\mathbb{R}^{2d_\theta} \times \mathcal{P}(\mathbb{R}^{2d_x}), d)$ is complete (Villani, 2009).

First note that using Jensen's inequality, $b$ is Lipschitz, and the fact that $(a + b + c + d)^2 \leq 4(a^2 + b^2 + c^2 + d^2)$ we obtain

$$\|\vartheta_t^\nu - \vartheta_t^{\nu'}\|^2 + \mathbb{E}[\|\Upsilon_t^\nu - \Upsilon_t^{\nu'}\|^2] = \mathbb{E}\left[\left\|\int_0^t b(\Upsilon_s^\nu, \vartheta_s^\nu, \nu_s^\Upsilon, \nu_s^\vartheta) - b(\Upsilon_s^{\nu'}, \vartheta_s^{\nu'}, \nu_s'^\vartheta, \nu_s'^\Upsilon)\mathrm{d}s\right\|^2\right]$$

$$\leq t \int_0^t \mathbb{E}\left[\left\|b(\Upsilon_s^\nu, \vartheta_s^\nu, \nu_s^\Upsilon, \nu_s^\vartheta) - b(\Upsilon_s^{\nu'}, \vartheta_s^{\nu'}, \nu_s'^\vartheta, \nu_s'^\Upsilon)\right\|^2\right]\mathrm{d}s$$

$$\leq tC \int_0^t \mathbb{E}\left[\left\|\Upsilon_s^\nu - \Upsilon_s^{\nu'}\right\|^2\right] + \|\vartheta_s^\nu - \vartheta_s^{\nu'}\|^2 + \|\nu_s^\vartheta - \nu_s'^\vartheta\|^2 + W_1^2(\nu_s^\Upsilon, \nu_s'^\Upsilon)\mathrm{d}s,$$

where $C = 4K_b^2$. Applying Grönwall's inequality, we obtain that

$$\|\vartheta_t^\nu - \vartheta_t^{\nu'}\|^2 + \mathbb{E}[\|\Upsilon_t^\nu - \Upsilon_t^{\nu'}\|^2] \leq C(t) \int_0^t \left[W_1^2(\nu_s^\Upsilon, \nu_s'^\Upsilon) + \|\nu_s^\vartheta - \nu_s'^\vartheta\|^2\right]\mathrm{d}s$$

where $C(t) = Ct \exp(Ct^2)$. Then, using the fact that the LHS is an upper bound for the squared distance $d$ and $W_1 \leq W_2$ (Villani, 2009, Remark 6.6), we have

$$d^2(F_T(\nu)_t, F_T(\nu')_t) \leq C(t) \int_0^t \left[W_1^2(\nu_s^\Upsilon, \nu_s'^\Upsilon) + \|\nu_s^\vartheta - \nu_s'^\vartheta\|^2\right]\mathrm{d}s$$

$$\leq C(t) \int_0^t \left[W_2^2(\nu_s^\Upsilon, \nu_s'^\Upsilon) + \|\nu_s^\vartheta - \nu_s'^\vartheta\|^2\right]\mathrm{d}s$$

$$\leq C(T) \int_0^t d^2(\nu_s, \nu_s')\mathrm{d}s \leq tC(T) \sup_{s \in [0,T]} d^2(\nu_s, \nu_s'), \tag{51}$$

where we use the fact that $C(t) \leq C(T)$ since $t \in [0,T]$. We show that for $k \geq 1$ successive compositions of the map $F_T$ denoted by $F_T^k$, we have the following inequality:

$$d^2(F_T^k(\nu)_t, F_T^k(\nu')_t) \leq \frac{(tC(T))^k}{k!} \sup_{u \in [0,T]} d^2(\nu_u, \nu_u'). \tag{52}$$

This can be proved inductively. The base case $k = 1$ follows immediately from (51), assume the inequality holds for $k - 1$, then for $k$ we have

$$d^2(F_T^k(\nu)_t, F_T^k(\nu')_t) \leq C(T) \int_0^t d^2(F_T^{k-1}(\nu)_s, F_T^{k-1}(\nu')_s)\mathrm{d}s$$

$$\leq \frac{C(T)^k}{(k-1)!} \sup_{u \in [0,T]} d^2(\nu_u, \nu_u') \int_0^t t^{k-1}\mathrm{d}s$$

$$\leq \frac{(tC(T))^k}{k!} \sup_{u \in [0,T]} d^2(\nu_u, \nu_u').$$

Hence, we have shown that the inequality (52) holds. Taking the supremum, we obtain

$$\sup_{s \in [0,T]} d^2(F_T^k(\nu)_s, F_T^k(\nu')_s) \leq \frac{(TC(T))^k}{k!} \sup_{s \in [0,T]} d^2(\nu_s, \nu'_s).$$

Since $k \mapsto (TC(T))^k \in o(k!)$, there exists a large enough $k$ such that there is a constant $0 < \alpha < 1$ for the following inequality holding:

$$\sup_{s \in [0,T]} d^2(F_T^k(\nu)_s, F_T^k(\nu')_s) \leq \alpha \sup_{s \in [0,T]} d^2(\nu_s, \nu'_s).$$

Hence, we have shown that the operator $F_T^k$ is a contraction and from the Banach Fixed Point theorem and completeness of the space $(C([0,T], \mathbb{R}^{d_\theta} \times \mathcal{P}(\mathbb{R}^{d_x})), \sup d)$, we have uniqueness and existence. $\qquad\square$

## H  SPACE DISCRETIZATION

In this section, we establish asymptotic pointwise propagation of chaos results. We are interested in justifying the use of a particle approximation in the flow (2). The flow (2) can be rewritten equivalently as follows. Let $(\Upsilon_0^i, W^i)_{i \in [M]}$ be i.i.d. copies of $(\Upsilon_0, W)$. We write $(\vartheta, \{\Upsilon^i\}_{i=1}^M)$ as solutions of (2) starting at $(\vartheta_0, \{\Upsilon_0^i\}_{i=1}^M)$ driven by the $\{W^i\}_{i=1}^M$. In other words, (2) satisfies

$$d\vartheta_t = b_\vartheta \left( \vartheta_t, \text{Law}(\Upsilon_t^1) \right) dt$$

$$\forall i \in [M] : d\Upsilon_t^i = b_\Upsilon(\Upsilon_t^i, \vartheta_t) dt + \sqrt{2} dW_t^i.$$

where

$$b_\vartheta(\vartheta_t, q) = \begin{bmatrix} \eta_\theta m \\ -\gamma_\theta \eta_\theta m - \nabla_\theta \mathcal{F}(\theta', q) \end{bmatrix}, \quad b_\Upsilon(\Upsilon, \vartheta) = \begin{bmatrix} \eta_x u \\ -\gamma_x \eta_x u + \nabla_x \ell(\theta, x) \end{bmatrix}.$$

Clearly, for all $i, j \in [M]$, we have $\text{Law}(\Upsilon_t^i) = \text{Law}(\Upsilon_t^j)$.

We will justify that we can replace the $\text{Law}(\Upsilon_t^1)$ with a particle approximation to obtain the approximate process (4), or equivalently as:

$$d\vartheta_t^M = b_\vartheta \left( \vartheta_t^M, \frac{1}{M} \sum_{i=1}^M \delta_{\Upsilon_t^{i,M}} \right) dt \tag{54a}$$

$$\forall i \in [M] : d\Upsilon_t^{i,M} = b_\Upsilon(\Upsilon_t^{i,M}, \vartheta_t^M) dt + \sqrt{2} dW_t^i. \tag{54b}$$

Similarly to Proposition G.2, we can show that $b_\vartheta$ and $b_\Upsilon$ are both Lipschitz. We are now prove ready to prove Proposition 5.1 justifying that (4) is a good approximation to (2).

*Proof of Proposition 5.1.* This is equivalent to showing that

$$\lim_{M \to \infty} \mathbb{E} \left[ \sup_{t \in [0,T]} \left\{ \|\vartheta_t - \vartheta_t^M\|^2 + \frac{1}{M} \sum_{i=1}^M \|\Upsilon_t^i - \Upsilon_t^{i,M}\|^2 \right\} \right] = 0.$$

More specifically, we will show that

$$\underbrace{\mathbb{E} \left[ \sup_{t \in [0,T]} \|\vartheta_t - \vartheta_t^M\|^2 \right]}_{(a)} + \underbrace{\mathbb{E} \left[ \sup_{t \in [0,T]} \frac{1}{M} \sum_{i=1}^M \|\Upsilon_t^i - \Upsilon_t^{i,M}\|^2 \right]}_{(b)} = o(1).$$

We begin with (a). From Jensen's inequality, we have

$$(a) \leq T\mathbb{E} \left[ \int_0^T \|b_\vartheta(\vartheta_s, \text{Law}(\Upsilon_s^1)) - b_\vartheta(\vartheta_s^M, \frac{1}{M} \sum_{i=1}^M \delta_{\Upsilon_s^{i,M}})\|^2 ds \right]$$

$$\leq C \int_0^T \mathbb{E}\|\vartheta_s - \vartheta_s^M\|^2 + \mathbb{E}W_2^2 \left( \text{Law}(\Upsilon_s^0), \frac{1}{M} \sum_{i=1}^M \delta_{\Upsilon_s^{i,M}} \right) ds,$$

where we use the fact that $b_\vartheta$ is $K_{b_\vartheta}$–Lipschitz, $C = 2TK_{b_\vartheta}$, and $\mathbb{E}(a+b)^2 \leq 2(\mathbb{E}a^2 + \mathbb{E}b^2)$ (known as the $C_r$–inequality (Loève, 1977, p157)).

Note that by the triangle inequality

$$\mathbb{E}W_2^2\left(\text{Law}(\Upsilon_s^0), \frac{1}{M}\sum_{i=1}^M \delta_{\Upsilon_s^{i,M}}\right) \leq 2\mathbb{E}W_2^2\left(\text{Law}(\Upsilon_s^0), \frac{1}{M}\sum_{i=1}^M \delta_{\Upsilon_s^i}\right)$$
$$+ 2\mathbb{E}W_2^2\left(\frac{1}{M}\sum_{i=1}^M \delta_{\Upsilon_s^i}, \frac{1}{M}\sum_{i=1}^M \delta_{\Upsilon_s^{i,M}}\right).$$

The two terms can be bounded using Carmona (2016, Eq. (1.24) and Lemma 1.9) to project the inequality

$$\mathbb{E}W_2^2\left(\text{Law}(\Upsilon_s^0), \frac{1}{M}\sum_{i=1}^M \delta_{\Upsilon_s^{i,M}}\right) \leq o(1) + \frac{2}{M}\sum_{i=1}^M \mathbb{E}\|\Upsilon_s^i - \Upsilon_s^{i,M}\|^2,$$

Hence, we have that

$$(a) \leq C'\int_0^T \left\{\mathbb{E}\|\vartheta_s - \vartheta_s^M\|^2 + o(1) + \frac{1}{M}\sum_{i=1}^M \mathbb{E}\|\Upsilon_s^i - \Upsilon_s^{i,M}\|^2\right\}\mathrm{d}s.$$

where $C' = \max(C, 2)$. We use the fact that

$$\|\vartheta_s - \vartheta_s^M\|^2 \leq \sup_{s'\in[0,s]}\|\vartheta_{s'} - \vartheta_{s'}^M\|^2, \quad \frac{1}{M}\sum_{i=1}^M \|\Upsilon_s^i - \Upsilon_s^{i,M}\|^2 \leq \sup_{s'\in[0,s]}\frac{1}{M}\sum_{i=1}^M \|\Upsilon_{s'}^i - \Upsilon_{s'}^{i,M}\|^2, \tag{55}$$

to obtain

$$(a) \leq C'\int_0^T \left\{\mathbb{E}\sup_{s'\in[0,s]}\|\vartheta_{s'} - \vartheta_{s'}^M\|^2 + o(1) + \mathbb{E}\sup_{s'\in[0,s]}\frac{1}{M}\sum_{i=1}^M \|\Upsilon_{s'}^i - \Upsilon_{s'}^{i,M}\|^2\right\}\mathrm{d}s \tag{56}$$

Similarly, for the other term, we have

$$(b) \leq \frac{T}{M}\sum_{i=1}^M \mathbb{E}\int_0^T \|b_\Upsilon(\Upsilon_s^i, \vartheta_s) - b_\Upsilon(\Upsilon_s^{i,M}, \vartheta_s^M)\|^2 \mathrm{d}s$$

$$\leq C\mathbb{E}\int_0^T \|\vartheta_s - \vartheta_s^M\|^2 + \frac{1}{M}\sum_{i=1}^M \|\Upsilon_s^i - \Upsilon_s^{i,M}\|^2 \mathrm{d}s$$

$$\leq C\int_0^T \mathbb{E}\sup_{s'\in[0,s]}\|\vartheta_{s'} - \vartheta_{s'}^M\|^2 + \sup_{s'\in[0,s]}\frac{1}{M}\sum_{i=1}^M \|\Upsilon_{s'}^i - \Upsilon_{s'}^{i,M}\|^2, \mathrm{d}s,$$

where for the last line we use the trick in (55).

Combining the bounds for $(a)$ and $(b)$, we obtain

$$(a) + (b) \leq C''\int_0^T \left\{\mathbb{E}\sup_{s'\in[0,s]}\|\vartheta_{s'} - \vartheta_{s'}^M\|^2 + o(1) + \mathbb{E}\sup_{s'\in[0,s]}\frac{1}{M}\sum_{i=1}^M \|\Upsilon_{s'}^i - \Upsilon_{s'}^{i,M}\|^2\right\}\mathrm{d}s,$$

where $C'' = C + C'$. Applying Gronwall's inequality, we obtain

$$\mathbb{E}\left[\sup_{t\in[0,T]}\|\vartheta_t - \vartheta_t^M\|\right] + \mathbb{E}\left[\sup_{t\in[0,T]}\frac{1}{M}\sum_{i=1}^M \|\Upsilon_t^i - \Upsilon_t^{i,M}\|\right] \leq o(1).$$

Taking the limit, we have as desired. $\qquad\square$

# I  Time Discretization

In this section, we are concerned with discretization schemes of various ODE/SDEs. The structure is as follows:

- Appendix I.1. We describe the Euler-Marayama discretization of PGD as described in Kuntz et al. (2023).
- Appendix I.2. We show we can obtain NAG as a discretization of damped Hamiltonian.
- Appendix I.3. We show a discretization of MPGD (described in Equation (2)) using a scheme replicating NAG as described in Appendix I.2 for the $(\theta, m)$-components, and Cheng et al. (2018)'s for $(x, u)$-components.
- Appendix I.4. We derive the transition using a scheme inspired by Cheng et al. (2018) while incorporating NAG-style gradient correction as described in Sutskever et al. (2013).

## I.1  PGD discretization

In order to obtain an implementable system, it is standard to then discretise the distribution $q_t$ by representing it with a finite particle system, i.e. $q_t(\mathrm{d}x) \approx \frac{1}{M} \sum_{i \in [M]} \delta\left(X_t^i, \mathrm{d}x\right)$. Upon making this approximation, one obtains the system

$$\dot{\theta}_t = \frac{1}{M} \sum_{i \in [M]} \nabla_\theta \ell\left(\theta_t, X_t^i\right)$$

$$\text{for } i \in [M], \quad \mathrm{d}X_t^i = \nabla_\theta \ell\left(\theta_t, X_t^i\right) \, \mathrm{d}t + \sqrt{2} \, \mathrm{d}W_t^i,$$

in which all terms are readily available. Discretising this process in time then yields the Particle Gradient Descent (PGD) algorithm of Kuntz et al. (2023), i.e. for $k \geq 1$, iterate

$$\theta_k = \theta_{k-1} + h\left(\frac{1}{M} \sum_{i=1}^M \nabla_\theta \ell\left(\theta_{k-1}, X_{k-1}^i\right)\right),$$

$$\text{for } i \in [M], \quad X_k^i = X_{k-1}^i + h\nabla_x \ell\left(\theta_{k-1}, X_{k-1}^i\right) + \sqrt{2h}\epsilon_k^i.$$

where $\epsilon_k^i \overset{\text{i.i.d.}}{\sim} \mathcal{N}(0_{d_x}, I_{d_x})$, for some initialization $(\theta_0, \{X_0^i\}_{i=1}^M)$.

## I.2  NAG as a discretization

Recall the accelerated ODE is given by

$$\ddot{\theta}_t + \gamma\eta\dot{\theta}_t + \eta\nabla_\theta f(\theta_t) = 0.$$

Let $m_t = \frac{1}{\eta}\dot{\theta}_t$, we can write the above equivalently as the following coupled first-order ODEs:

$$\dot{m}_t = -\gamma\eta m_t - \nabla f(\theta_t), \quad \dot{\theta}_t = \eta m_t.$$

When $\gamma = \frac{3}{t}$ and $\eta = 1$, we show that a particular discretization of the above is equivalent to Nesterov Accelerated Gradient (NAG) method (Nesterov, 1983). The argument is inspired by reversing the one of Su et al. (2014) who obtained the continuous limit of NAG.

Recall that NAG (Nesterov, 1983) for convex $f$ (but not strongly convex) is given by the following:

$$\theta_k = y_{k-1} - h\nabla f(y_{k-1}), \quad y_k = \theta_k + \left(\frac{k-1}{k+2}\right)(\theta_k - \theta_{k-1}),$$

and since $\frac{k-1}{k+2} \approx (1 - \frac{3}{k})$, we have

$$\theta_k \approx y_{k-1} - h\nabla_\theta f(y_{k-1}), \quad y_k \approx \theta_k + \left(1 - \frac{3}{k}\right)(\theta_k - \theta_{k-1}). \tag{57}$$

We will now show how a particular discretization scheme will produce (57). It uses a combination of implicit Euler for $\dot{\theta}_t$, and use a semi-implicit Euler scheme for $\dot{m}_t$. More specifically, the semi-implicit scheme for $\dot{m}_t = -\frac{3}{t}m_t - \nabla_\theta f(\theta_t)$ uses an explicit approximation for the momentum

$\frac{3}{t}m_t$, and implicit approximation for the gradient $\nabla_\theta f(\theta_t)$. In summary, we obtain the following discretization

$$m_t \approx m_{t-\sqrt{h}} - \sqrt{h}\left(\frac{3}{t-\sqrt{h}}m_{t-\sqrt{h}} + \nabla_\theta f(\theta_t)\right), \quad \theta_t \approx \theta_{t-\sqrt{h}} + \sqrt{h}m_t.$$

We can write the above equivalently through the map $t \mapsto \frac{t}{\sqrt{h}}$ and in terms of $k := \frac{t}{\sqrt{h}}$, to obtain

$$m_k \approx \left(1 - \frac{3}{k-1}\right)m_{k-1} - \sqrt{h}\nabla_\theta f(\theta_k),$$

$$\theta_k \approx \theta_{k-1} + \sqrt{h}m_k.$$

Hence,

$$\theta_k \approx \theta_{k-1} + \sqrt{h}\left(1 - \frac{3}{k-1}\right)m_{k-1} - h\nabla_\theta f(\theta_k).$$

From our discretization, we have that $\sqrt{h}m_k \approx \theta_k - \theta_{k-1}$ then we obtain

$$\theta_k \approx \theta_{k-1} + \left(1 - \frac{3}{k-1}\right)(\theta_{k-1} - \theta_{k-2}) - h\nabla_\theta f(\theta_k).$$

We can write the $\theta$ update in terms of $y_k$ (cf. (57)),

$$\theta_k \approx y_{k-1} - h\nabla_\theta f(\theta_k).$$

If $f$ is Lipschitz, then we have

$$\|h\nabla_\theta f(\theta_k) - h\nabla_\theta f(y_{k-1})\| \leq Lh\|\theta_k - y_{k-1}\| \leq h^2 L\|\nabla_\theta f(\theta_k)\| \leq o(h).$$

We replace the gradient $\nabla_\theta f(\theta_k)$ with $\nabla_\theta f(y_{k-1})$ to obtain as desired

$$\theta_k \approx y_{k-1} - h\nabla_\theta f(y_{k-1}), \quad y_k \approx \theta_k + \left(1 - \frac{3}{k}\right)(\theta_k - \theta_{k-1}).$$

Interestingly, for given step size $h$, the (approximate) NAG iterations approximates the flow for time $\sqrt{h}$ as opposed to $h$ (Su et al., 2014, Section 3.4)

### I.3   NESTEROV AND CHENG'S DISCRETIZATION OF MPGD

In this section, we show how to discretize the MPGD in using a combination of Nesterov's (see Appendix I.2) and Cheng et al. (2018)'s discretization. Recall, we have

$$\begin{aligned}
\mathrm{d}\theta_t &= \eta_\theta m_t\,\mathrm{d}t, \\
\mathrm{d}m_t &= -\nabla_\theta \mathcal{E}(\theta_t, q_{t,X}^M)\,\mathrm{d}t - \gamma_\theta\eta_\theta m_t\,\mathrm{d}t, \\
\mathrm{d}X_t^i &= \eta_x U_t^i\mathrm{d}t, \\
\mathrm{d}U_t^i &= \nabla_x\ell(\theta_t, X_0^i)\mathrm{d}t - \gamma_x\eta_x U_t\mathrm{d}t + \sqrt{2\gamma_x}\,\mathrm{d}W_t^i,
\end{aligned}$$

where $\nabla_\theta\hat{\mathcal{F}}(\theta, \{X^i\}_{i=1}^M) := -\frac{1}{N}\sum_{i=1}^M \nabla_\theta\ell(\theta, X^i)$.

In this section, we show how discretizing the $\theta$ component in the style of Nesterov (specifically, Sutskever et al. (2013)'s formulation) and $q$ component in the style of Cheng et al. (2018) obtains the MPGD-NC (Nesterov-Cheng) algorithm. The MPGD-NC algorithm is described as follows: given previous values $(\theta_k, v_k, \{X_k^i, U_k^i\}_{i=1}^M)$ and step-size $h > 0$, we iterate

$$\theta_{k+1} = \theta_k + v_k, \tag{58a}$$

$$v_{k+1} = \mu v_k - h^2\nabla_\theta\mathcal{E}(\theta_k + \mu v_k, q_{k,X}^M), \tag{58b}$$

$$\forall i \in [M] : X_{k+1}^i = X_k^i + \frac{1}{\gamma_x}\left[(1 - \omega_x(h))U_k + \nabla_x\ell(\theta_{k+1}, X_k^i)\left(h - \frac{1 - \omega_x(h)}{\gamma_x\eta_x}\right)\right] + L_\Sigma^{XX}\xi_k^i, \tag{58c}$$

$$\forall i \in [M] : U_{k+1}^i = \omega_x(h)U_k^i + \frac{1 - \omega_x(h)}{\gamma_x\eta_x}\nabla_x\ell(\theta_{k+1}, X_k^i) + L_\Sigma^{XU}\xi_k^i + L_\Sigma^{UU}\xi_k'^i, \tag{58d}$$

for all $i \in [N]$, where $\mu := 1 - h\gamma_\theta\eta_\theta$, $\omega_x(t) :=;$ $L_\Sigma^{XX}, L_\Sigma^{XU}, L_\Sigma^{UU}$ is described in (64), and $\xi_k, \xi_k' \sim \mathcal{N}(0, I_{d_x})$. Each iteration corresponds to (approximately) solving (2) for time $h$.

In Appendix I.3.1, we show how Nesterov's discretization can be used to produce the update

$$\theta_{k+1} = \theta_k + v_k,$$
$$v_{k+1} = \mu v_k - h^2 \nabla_\theta \mathcal{E}(\theta_k + \mu v_k, q_{k,X}^M),$$

and in Appendix I.3.2, we how given $(\theta_{k+1}, \{X_k^i, U_k^i\}_{i=1}^M)$, the transition is described by

$$X_{k+1}^i = X_k^i + \frac{1}{\gamma_x}\left[(1 - \omega_x(h))U_k + \nabla_x \ell(\theta_{k+1}, X_k^i)\left(h - \frac{1 - \omega_x(h)}{\gamma_x\eta_x}\right)\right] + L_\Sigma^{XX}\xi_k^i,$$
$$U_{k+1}^i = \omega_x(h)U_k^i + \frac{1 - \omega_x(h)}{\gamma_x\eta_x}\nabla_x \ell(\theta_{k+1}, X_k^i) + L_\Sigma^{XU}\xi_k^i + L_\Sigma^{UU}\xi_k'^i.$$

### I.3.1 DISCRETIZATION OF $(\dot\theta_t, \dot m_t)$

This discretization follows similarly to that in Appendix I.2. Similarly, let $k := \frac{t}{\sqrt{h}}$ with the time rescaling $t \mapsto \frac{t}{\sqrt{h}}$, then we consider the following discretization:

$$m_k = m_{k-1} - \sqrt{h}\left(\gamma_\theta\eta_\theta m_{k-1} + \nabla_\theta \mathcal{E}(\theta_k, q_{k-1,X}^M)\right)$$
$$= \left(1 - \sqrt{h}\gamma_\theta\eta_\theta\right)m_{k-1} - \sqrt{h}\nabla_\theta \mathcal{E}(\theta_k, q_{k-1,X}^M),$$
$$\theta_k = \theta_{k-1} + \sqrt{h}\eta_\theta m_k,$$

Expanding $m_k$, we obtain

$$\theta_k = \theta_{k-1} + \sqrt{h}\eta_\theta\left(1 - \sqrt{h}\gamma_\theta\eta_\theta\right)m_{k-1} - h\nabla_\theta \mathcal{E}(\theta_k, q_{k-1,X}^M)$$
$$= \theta_{k-1} + \left(1 - \sqrt{h}\gamma_\theta\eta_\theta\right)(\theta_{k-1} - \theta_{k-2}) - h\nabla_\theta \mathcal{E}(\theta_k, q_{k-1,X}^M),$$

Let $v_t = \theta_k - \theta_{k-1}$, then we have

$$\theta_k = \theta_{k-1} + v_k$$
$$v_k = \bar\mu v_{k-1} - h\nabla_\theta \mathcal{E}(\theta_k, q_{k-1,X}^M),$$

where $\bar\mu := 1 - \sqrt{h}\gamma_\theta\eta_\theta$. Then, as before, using the approximation $h\nabla_\theta \mathcal{E}(\theta_k, q_{k-1,X}^M) \approx h\nabla_\theta \mathcal{E}(\theta_{k-1} + \bar\mu v_{k-1}, q_{k-1,X}^M) + o(h)$, we obtain that

$$\theta_k = \theta_{k-1} + v_k,$$
$$v_k = \bar\mu v_{k-1} - h\nabla_\theta \mathcal{E}(\theta_{k-1} + \bar\mu v_{k-1}, q_{k-1,X}^M).$$

Similarly to Appendix I.2 this approximates the flow for time $\sqrt{h}$ instead of $h$. Hence, we have to apply the appropriate rescaling to obtain the following iteration with the desired behaviour:

$$\theta_k = \theta_{k-1} + v_k,$$
$$v_k = \mu v_{k-1} - h^2 \nabla_\theta \mathcal{E}(\theta_{k-1} + \mu v_{k-1}, q_{k-1,X}^M),$$

where $\mu := 1 - h\gamma_\theta\eta_\theta$. This is exactly that of Sutskever et al. (2013)'s characterization of NAG (see Equations 3 and 4 in their paper).

### I.3.2 DISCRETIZATION OF $(\dot X_t, \dot U_t)$

We describe the discretization scheme of Cheng et al. (2018). For simplicity, we derive the transition of a single particle $(X_t, U_t)$ since there are no interactions between the particles given $\theta$. Furthermore, for brevity and without loss in generality, we derive the transition for some step size $h > 0$ given initial values $(\theta, X_0, U_0)$, which can be easily generalized to future transitions.

Consider a time interval $t \in [0, h]$ and given $(X_0, U_0)$, we first approximate the gradient $\nabla_x \ell(\theta, X_t)$ with $\nabla_x \ell(\theta, X_0)$ to arrive at the following linear SDE:

$$d \begin{pmatrix} X_t \\ U_t \end{pmatrix} = \left[ \begin{pmatrix} 0 \\ \nabla_x \ell(\theta, X_0) \end{pmatrix} + \begin{pmatrix} 0 & \eta_x I \\ 0 & -\gamma_x \eta_x I \end{pmatrix} \begin{pmatrix} X_t \\ U_t \end{pmatrix} \right] dt + \sqrt{2\gamma_x} \begin{pmatrix} 0 \\ 1 \end{pmatrix} dW_t. \quad (59)$$

A $2d_x$-dimensional linear SDE is given by:

$$\mathrm{d}\boldsymbol{X}_t = (\boldsymbol{A}\boldsymbol{X}_t + \boldsymbol{\alpha}) \, \mathrm{d}t + \boldsymbol{\beta} \, \mathrm{d}W_t, \quad (60)$$

where $\boldsymbol{A} \in \mathbb{R}^{2d_x \times 2d_x}$ and $\boldsymbol{\alpha}, \boldsymbol{\beta} \in \mathbb{R}^{2d_x}$ are fixed matrices. It is clear that if we set

$$\boldsymbol{X}_t = \begin{pmatrix} X_t \\ U_t \end{pmatrix}, \quad \boldsymbol{A} = \begin{pmatrix} 0_{d_x} & \eta_x I_{d_x} \\ 0_{d_x} & -\gamma_x \eta_x I_{d_x} \end{pmatrix}, \quad \boldsymbol{\alpha} = \begin{pmatrix} 0_{d_x \times 1} \\ \nabla_x \ell(\theta, X_0) \end{pmatrix}, \quad \boldsymbol{\beta} = \sqrt{2\gamma_x} \begin{pmatrix} 0_{d_x \times 1} \\ 1_{d_x \times 1} \end{pmatrix},$$

then the discretized underdamped Langevin SDE of (59) falls within the class of linear SDE of the form specified in (60). These SDEs admits the following explicit solution (Platen and Bruti-Liberati, 2010, see page 48, 101),

$$\boldsymbol{X}_t = \boldsymbol{\Psi}_t \left( \boldsymbol{X_0} + \int_0^t \boldsymbol{\Psi}_s^{-1} \boldsymbol{\alpha} \, \mathrm{d}s + \int_0^t \boldsymbol{\Psi}_s^{-1} \boldsymbol{\beta} \, \mathrm{d}W_s \right), \quad (61)$$

where

$$\boldsymbol{\Psi}_t := \exp(\boldsymbol{A}t),$$

with $\exp$ to be understood as the matrix exponential. In our case, the matrix exponential and its inverse is given by

$$\boldsymbol{\Psi}_t = I_{2d_x} + \sum_{i=1}^\infty \frac{1}{k!} \begin{bmatrix} 0_{d_x} & \frac{(-\gamma_x \eta_x t)^k}{-\gamma_x} I_{d_x} \\ 0_{d_x} & (-\gamma_x \eta_x t)^k I_{d_x} \end{bmatrix} = \begin{pmatrix} I_{d_x} & \frac{1-\omega_x(t)}{\gamma_x} I_{d_x} \\ 0_{d_x} & \omega_x(t) I_{d_x} \end{pmatrix}, \quad (62a)$$

$$\boldsymbol{\Psi}_t^{-1} = \begin{pmatrix} I_{d_x} & \frac{1-\omega_x(-t)}{\gamma_x} I_{d_x} \\ 0_{d_x} & \omega_x(-t) I_{d_x} \end{pmatrix}, \quad (62b)$$

where $\omega_x(t) := \exp(-\gamma_x \eta_x t)$. It can be verified that they are indeed the inverse of each other, i.e., $\boldsymbol{\Psi}_t \boldsymbol{\Psi}_t^{-1} = I_{2d_x}$.

Plugging in (62) into (61), we obtain the following:

$$\begin{pmatrix} X_t \\ U_t \end{pmatrix} = \begin{pmatrix} I_{d_x} & \frac{1-\omega_x(t)}{\gamma_x} I_{d_x} \\ 0_{d_x} & \omega_x(t) I_{d_x} \end{pmatrix} \left( \begin{bmatrix} X_0 \\ U_0 \end{bmatrix} + \int_0^t \begin{pmatrix} \frac{1-\omega_x(-s)}{\gamma_x} \nabla_x \ell(\theta, X_0) \\ \omega_x(-s) \nabla_x \ell(\theta, X_0) \end{pmatrix} \mathrm{d}s + \sqrt{2} \int_0^t \begin{pmatrix} \frac{1-\omega_x(-s)}{\sqrt{\gamma_x}} 1_{d_x \times 1} \\ \sqrt{\gamma_x} \omega_x(-s) 1_{d_x \times 1} \end{pmatrix} \mathrm{d}W_s \right),$$

Or, equivalently, we have

$$U_t = \omega_x(t) U_0 + \nabla_x \ell(\theta, X_0) \int_0^t \omega_x(t-s) \, \mathrm{d}s + \sqrt{2\gamma_x} \int_0^t \omega_x(t-s) \, \mathrm{d}W_s$$

$$= \omega_x(t) U_0 + \frac{1-\omega_x(t)}{\gamma_x \eta_x} \nabla_x \ell(\theta, X_0) + \sqrt{2\gamma_x} \int_0^t \omega_x(t-s) \, \mathrm{d}W_s,$$

$$X_t = X_0 + \frac{1}{\gamma_x} \left[ (1-\omega_x(t)) U_0 + \nabla_x \ell(\theta, X_0) \left[ \int_0^t 1 - \omega_x(t-s) \, \mathrm{d}s \right] + \sqrt{\frac{2}{\gamma_x}} \int_0^t 1 - \omega_x(t-s) \, \mathrm{d}W_s \right]$$

$$= X_0 + \frac{1}{\gamma_x} \left[ (1-\omega_x(t)) U_0 + \nabla_x \ell(\theta, X_0) \left[ t - \frac{1-\omega_x(t)}{\gamma_x \eta_x} \right] + \sqrt{\frac{2}{\gamma_x}} \int_0^t 1 - \omega_x(t-s) \, \mathrm{d}W_s \right].$$

As noted by Cheng et al. (2018), this is a Gaussian transition. To characterize it, we need to calculate the first two moments:

For the first moments, we have

$$\mu_U(X_0, U_0, t) := \mathbb{E}[U_t]$$

$$= \omega_x(t)U_0 + \frac{1 - \omega_x(t)}{\gamma_x \eta_x} \nabla_x \ell(\theta, X_0)$$

$$\mu_X(X_0, U_0, t) := \mathbb{E}[X_t]$$

$$= X_0 + \frac{1}{\gamma_x} \left[ (1 - \omega_x(t))U_0 + \nabla_x \ell(\theta, X_0) \left[ t - \frac{1 - \omega_x(t)}{\gamma_x \eta_x} \right] \right]$$

For the second moments, we have

$$\Sigma_{UU}(t) := \mathbb{E}[(U_t - \mathbb{E}[U_t])(U_t - \mathbb{E}[U_t])^\top]$$

$$= 2\gamma_x \mathbb{E}\left[ \left( \int_0^t \omega_x(t-s) \, dW_s \right) \left( \int_0^t \omega_x(t-s) \, dW_s \right)^\top \right]$$

$$= 2\gamma_x \left( \int_0^t \omega_x(2(t-s)) \, ds \right) I_{d_x}$$

$$= \left( \frac{1 - \omega_x(2t)}{\eta_x} \right) I_{d_x}$$

$$\Sigma_{XX}(t) := \mathbb{E}[(X_t - \mathbb{E}[X_t])(X_t - \mathbb{E}[X_t])^\top]$$

$$= \frac{2}{\gamma_x} \mathbb{E}\left[ \left( \int_0^t 1 - \omega_x(t-s) \, dW_s \right) \left( \int_0^t 1 - \omega_x(t-s) \, dW_s \right)^\top \right]$$

$$= \frac{2}{\gamma_x} \left( \int_0^t [1 - \omega_x(t-s)]^2 \, ds \right) I_{d_x}$$

$$= \frac{1}{\gamma_x} \left[ 2t - \frac{\omega_x(2t)}{\gamma_x \eta_x} + \frac{4\omega_x(t)}{\gamma_x \eta_x} - \frac{3}{\eta_x \gamma_x} \right] I_{d_x},$$

$$\Sigma_{UX}(t) := \mathbb{E}[(U_t - \mathbb{E}[U_t])(X_t - \mathbb{E}[X_t])^\top]$$

$$= 2\mathbb{E}\left[ \left( \int_0^t \omega_x(t-s) \, dW_s \right) \left( \int_0^t 1 - \omega_x(t-s) \, dW_s \right)^\top \right]$$

$$= 2 \left( \int_0^t \omega_x(t-s)(1 - \omega_x(t-s)) \, ds \right) I_{d_x}$$

$$= \frac{1}{\gamma_x \eta_x} (1 - 2\omega_x(t) + \omega_x(2t)) I_{d_x}.$$

Hence, the transition can be described as $\begin{pmatrix} X_t \\ U_t \end{pmatrix} \sim \mathcal{N}(\mu(t), \Sigma(t))$ where

$$\mu(t) = \begin{pmatrix} \mu_X(X_k, U_k, t) \\ \mu_U(X_k, U_k, t) \end{pmatrix}, \quad \Sigma(t) = \begin{pmatrix} \Sigma_{XX}(t) & \Sigma_{UX}(t) \\ \Sigma_{UX}(t) & \Sigma_{UU}(t) \end{pmatrix}.$$

Therefore, given some point $(X_k, U_k)$ and some step-size $h > 0$, we have that

$$\begin{pmatrix} X_{k+1} \\ U_{k+1} \end{pmatrix} \sim \mathcal{N}(\mu(h), \Sigma(h)). \tag{63}$$

**Sampling from the transition**. Using samples from a standard Gaussian $z \sim \mathcal{N}(0_d, I_d)$, one may produce samples from of a multivariate distribution $\mathcal{N}(\mu, \Sigma)$ by using the fact that

$$X = \mu + Lz \sim \mathcal{N}(\mu, \Sigma),$$

where $L$ is a lower triangular matrix with positive diagonal entries obtained via the Cholesky decomposition of $\Sigma$, i.e, $\Sigma = LL^\top$.

The Cholesky decomposition of the covariance matrix of (63) will be described here. Consider the LDL decomposition of a symmetric matrix

$$\Sigma = \begin{bmatrix} A & B \\ B^\top & C \end{bmatrix} = \begin{bmatrix} I & 0 \\ B^\top A^{-1} & I \end{bmatrix} \begin{bmatrix} A & 0 \\ 0 & S \end{bmatrix} \begin{bmatrix} I & A^{-1}B \\ 0 & I \end{bmatrix},$$

where $S = D - B^\top A^{-1}B$ is the Schur complement. Given Cholesky factorization of $A$ and $B$, written as $A = L_A L_A^\top$ and $S = L_S L_S^\top$ respectively, we can write the Cholesky decomposition of $\Sigma$ as

$$\Sigma = \begin{bmatrix} L_A & 0 \\ B^\top L_A^{-\top} & L_S \end{bmatrix} \begin{bmatrix} L_A^\top & L_A^{-1}B \\ 0 & L_S^\top \end{bmatrix} = L_\Sigma L_\Sigma^\top.$$

In our case, we compute the Cholesky decomposition of $\Sigma$ as follows:

$$L_\Sigma = \begin{pmatrix} L_\Sigma^{XX} I_{d_x} & 0 \\ L_\Sigma^{XU} I_{d_x} & L_\Sigma^{UU} I_{d_x} \end{pmatrix},$$

where the constants are defined as

$$L_\Sigma^{XX} = \sqrt{\frac{1}{\gamma_x}\left[2h - \frac{\omega_x(2h)}{\gamma_x\eta_x} + \frac{4\omega_x(h)}{\gamma_x\eta_x} - \frac{3}{\eta_x\gamma_x}\right]}, \tag{64a}$$

$$L_\Sigma^{XU} = \frac{\frac{1}{\gamma_x\eta_x}(1 - 2\omega_x(h) + \omega_x(2h))}{\sqrt{\frac{1}{\gamma_x}\left[2h - \frac{\omega_x(2h)}{\gamma_x\eta_x} + \frac{4\omega_x(h)}{\gamma_x\eta_x} - \frac{3}{\eta_x\gamma_x}\right]}}, \tag{64b}$$

$$L_\Sigma^{UU} = \sqrt{\frac{1 - \omega_x(2h)}{\eta_x} - \frac{\left(\frac{1}{\gamma_x\eta_x}(1 - 2\omega_x(h) + \omega_x(2h))\right)^2}{\frac{1}{\gamma_x}\left[2h - \frac{\omega_x(2h)}{\gamma_x\eta_x} + \frac{4\omega_x(h)}{\gamma_x\eta_x} - \frac{3}{\eta_x\gamma_x}\right]}}. \tag{64c}$$

Therefore, we the transition can be written as follows:

$$\forall i \in [M] : X_{k+1}^i = X_k^i + \frac{1}{\gamma_x}\left[(1 - \omega_x(h))U_k + \nabla_x\ell(\theta, X_k^i)\left[h - \frac{1 - \omega_x(h)}{\gamma_x\eta_x}\right]\right] + L_\Sigma^{XX}\xi_k^i,$$

$$\forall i \in [M] : U_{k+1}^i = \omega_x(h)U_k^i + \frac{1 - \omega_x(h)}{\gamma_x\eta_x}\nabla_x\ell(\theta, X_k^i) + L_\Sigma^{XU}\xi_k^i + L_\Sigma^{UU}\xi_k^{\prime,i}.$$

where $\{\xi_k^i, \xi_k^{\prime,i}\}_{i\in[M],k} \overset{\text{i.i.d.}}{\sim} \mathcal{N}(0_{d_x}, I_{d_x})$.

## I.4 PROPOSED DISCRETIZATION OF (5)

Recall, our approximating SDE in (5) is given by:

$$\mathrm{d}\tilde{\theta}_t = \eta_\theta \tilde{m}_t\,\mathrm{d}t$$

$$\mathrm{d}\tilde{m}_t = -\gamma_\theta\eta_\theta\tilde{m}_t\,\mathrm{d}t - \nabla_\theta\mathcal{E}\left(\bar{\theta}_0, \tilde{q}_{0,X}^M\right)\,\mathrm{d}t$$

$$\text{for } i \in [M], \quad \mathrm{d}\tilde{X}_t^i = \eta_x\tilde{U}_t^i\,\mathrm{d}t$$

$$\text{for } i \in [M], \quad \mathrm{d}\tilde{U}_t^i = -\gamma_x\eta_x\tilde{U}_t^i\,\mathrm{d}t + \nabla_x\ell\left(\tilde{\theta}_h, X_0^i\right)\,\mathrm{d}t + \sqrt{2\gamma_x}\,\mathrm{d}W_t^i.$$

For simplicity and without loss in generality, we assume there is a single particle, i.e., $M = 1$, and derive the transition for a single step. Clearly, (5) can be written as an linear $2d_x + 2d_\theta$ SDE:

$$\mathrm{d}\boldsymbol{X}_t = (\boldsymbol{A}\boldsymbol{X}_t + \boldsymbol{\alpha})\,\mathrm{d}t + \boldsymbol{\beta}\,\mathrm{d}W_t,$$

where

$$\boldsymbol{X}_t = \begin{pmatrix} \tilde{\theta}_t \\ \tilde{m}_t \\ \tilde{X}_t \\ \tilde{U}_t \end{pmatrix}, \quad \boldsymbol{A} = \begin{pmatrix} 0_{d_\theta} & \eta_\theta I_{d_\theta} & & \\ 0_{d_\theta} & -\eta_\theta\gamma_\theta I_{d_\theta} & & 0_{2d_\theta \times 2d_x} \\ & & 0_{d_x} & \eta_x I_{d_x} \\ 0_{2d_x \times 2d_\theta} & & 0_{d_x} & -\eta_x\gamma_x I_{d_x} \end{pmatrix},$$

$$\boldsymbol{\alpha} = \begin{pmatrix} 0_{d_\theta \times 1} \\ -\nabla_\theta \mathcal{E}(\bar{\theta}_0, q_{0,X}^M) \\ 0_{d_x \times 1} \\ \nabla_x \ell(\tilde{\theta}_h, X_0) \end{pmatrix}, \quad \boldsymbol{\beta} = \begin{pmatrix} 0_{d_\theta \times 1} \\ 0_{d_\theta \times 1} \\ 0_{d_x \times 1} \\ 1_{d_x \times 1} \end{pmatrix}$$

Hence it admits the following explicit solution (Platen and Bruti-Liberati, 2010, see page 48, 101):

$$\boldsymbol{X}_t = \boldsymbol{\Psi}_t \left( \boldsymbol{X_0} + \int_0^t \boldsymbol{\Psi}_s^{-1} \boldsymbol{\alpha} \, \mathrm{d}s + \int_0^t \boldsymbol{\Psi}_s^{-1} \boldsymbol{\beta} \, \mathrm{d}W_s \right), \tag{65}$$

where

$$\boldsymbol{\Psi}_t := \exp\left( \boldsymbol{A} t \right),$$

with exp to be understood as the matrix exponential. In our case, similarly to (62), we have

$$\boldsymbol{\Psi}_t = \begin{pmatrix} I_{d_\theta} & \frac{1-\omega_\theta(t)}{\gamma_\theta} I_{d_\theta} & & \\ 0_{d_\theta} & \omega_\theta(t) I_{d_\theta} & & 0_{2d_\theta \times 2d_x} \\ & & I_{d_x} & \frac{1-\omega_x(t)}{\gamma_x} I_{d_x} \\ 0_{2d_x \times 2d_\theta} & & 0_{d_x} & \omega_x(t) I_{d_x} \end{pmatrix},$$

$$\boldsymbol{\Psi}_t^{-1} = \begin{pmatrix} I_{d_\theta} & \frac{1-\omega_\theta(-t)}{\gamma_\theta} I_{d_\theta} & & \\ 0_{d_\theta} & \omega_\theta(-t) I_{d_\theta} & & 0_{2d_\theta \times 2d_x} \\ & & I_{d_x} & \frac{1-\omega_x(-t)}{\gamma_x} I_{d_x} \\ 0_{2d_\theta \times 2d_x} & & 0_{d_x} & \omega_x(-t) I_{d_x} \end{pmatrix},$$

where $\omega_x(t) := \exp(-\gamma_x\eta_x t)$, and similarly $\omega_\theta(t) := \exp(-\gamma_\theta\eta_\theta t)$. Hence, we have

$$\tilde{\theta}_t = \theta_0 + \frac{1}{\gamma_\theta} \left[ (1 - \omega_\theta(t))m_0 - \left( t - \frac{1-\omega_\theta(t)}{\gamma_\theta\eta_\theta} \right) \nabla_\theta\mathcal{E}(\bar{\theta}_0, q_{0,X}^M) \right]$$

$$\tilde{m}_t = \omega_\theta(t)m_0 - \frac{1-\omega_\theta(t)}{\gamma_\theta\eta_\theta} \nabla_\theta\mathcal{E}(\bar{\theta}_0, q_{0,X}^M)$$

$$\tilde{X}_t = X_0 + \frac{1}{\gamma_x} \left[ (1 - \omega_x(t))U_0 + \left( t - \frac{1-\omega_x(t)}{\gamma_x\eta_x} \right) \nabla_x\ell(\tilde{\theta}_h, X_0) \right] + L_\Sigma^{XX}\xi,$$

$$\tilde{U}_t = \omega_x(t)U_0 + \frac{1-\omega_x(t)}{\gamma_x\eta_x} \nabla_x\ell(\tilde{\theta}_h, X_0) + L_\Sigma^{XU}\xi + L_\Sigma^{UU}\xi',$$

where the constants are exactly the same as those from Equation (64), and $\xi, \xi' \sim \mathcal{N}(0, I_{d_x})$.

It can be seen that taking setting $t = h$, the general iterations are given by

$$\tilde{\theta}_{k+1} = \tilde{\theta}_k + \frac{1}{\gamma_\theta} \left[ (1 - \omega_\theta(k))\tilde{m}_k - \left( t - \frac{1-\omega_\theta(k)}{\gamma_\theta\eta_\theta} \right) \nabla_\theta\mathcal{E}(\bar{\theta}_k, \tilde{q}_{k,X}^M) \right]$$

$$\tilde{m}_{k+1} = \omega_\theta(k)\tilde{m}_k - \frac{1-\omega_\theta(k)}{\gamma_\theta\eta_\theta} \nabla_\theta\mathcal{E}(\bar{\theta}_k, \tilde{q}_{k,X}^M)$$

$$\forall i \in [M] : \tilde{X}_{k+1}^i = \tilde{X}_k^i + \frac{1}{\gamma_x} \left[ (1 - \omega_x(k))\tilde{U}_k^i + \left( t - \frac{1-\omega_x(k)}{\gamma_x\eta_x} \right) \nabla_x\ell(\tilde{\theta}_{k+1}, \tilde{X}_k^i) \right] + L_\Sigma^{XX}\xi_k^i$$

$$\forall i \in [M] : \tilde{U}_{k+1}^i = \omega_x(k)\tilde{U}_k^i + \frac{1-\omega_x(k)}{\gamma_x\eta_x} \nabla_x\ell(\tilde{\theta}_{k+1}, \tilde{X}_k^i) + L_\Sigma^{XU}\xi_k + L_\Sigma^{UU}\xi_k'$$

where $\{\xi, \xi'\}_{k,i\in[M]} \sim \mathcal{N}(0, I_{d_x})$.

# J  PRACTICAL CONCERNS AND EXPERIMENTAL DETAILS

Here, we describe practical considerations and experiment details. First, in Appendix J.1, we describe the RMSProp precondition (Tieleman and Hinton, 2012; Staib et al., 2019) used in our experiments. Then, in Appendix J.2, we describe the subsampling procedure used for our image generation task. After, in Appendix J.3, we discuss the heuristic we introduced for choosing momentum parameters $(\gamma_\theta, \gamma_x, \eta_\theta, \eta_x)$. Finally, in Appendix J.4, we detail all the parameters and models used in our experiment for reproducibility.

## J.1  RMSPROP PRECONDITIONER

Given $0 < \beta < 1$, the RMSProp preconditioner $G_k$ (Tieleman and Hinton, 2012; Staib et al., 2019) at iteration $k$ is defined by the following iteration:

$$G_{k+1} = \beta G_k + (1 - \beta)\nabla_\theta \mathcal{E}(\bar{\theta}_k, q_{k,X}^M)^2,$$

where $G_0 = 0_{d_\theta}$ and $x^2$ denotes element-wise square.

## J.2  SUBSAMPLING

In the presence of a large dataset, it is common to develop computationally cheaper implementations by appropriate using of data sub-sampling. Such a scheme was devised in Kuntz et al. (2023, Appendix E.4). We utilize the same subsampling scheme for $(\theta, m)$-components. However, we found it necessary to devise a new scheme for $(x, u)$-components. We found that this alternative subsampling scheme substantially improved the performance of MPGD but did not noticeably affect the PGD algorithm.

It is desirable to update the whole particle cloud. However, in cases where each sample has its own posterior cloud approximation, as in the generative modelling task, the dataset can become prohibitively large for updating the whole cloud. In Kuntz et al. (2023), the subsampling scheme only updated the portion of the particle cloud associated with the mini-batch, which neglects the remainder of the posterior approximation. As such, we propose to update the subsampled cloud at the beginning of each iteration at a step proportional to the "time"/steps it has missed. This is described more succinctly in Algorithm 1.

---

**Algorithm 1** A single subsampled step. In pink, we indicate the existing subsampling scheme of Kuntz et al. (2023). We indicate our proposed additions in teal.

---

**Require:** Subsampled indices $\mathcal{I}$, subsampled data $\{y^i\}_{i \in \mathcal{I}}$, step-size $h$, previous iterates $(\theta_k, m_k, \{X_k^i, U_k^i\}_{i \in \mathcal{I}})$
  // Updated cloud for missed time
  $\{X_{k+\epsilon}^i, U_{k+\epsilon}^i\}_{i \in \mathcal{I}} \leftarrow$ Solve Equations (5c) and (5d) with step-size $(h \times missed[\mathcal{I}])$ and $(\theta_k, m_k, \{X_k^i, U_k^i\}_{i \in \mathcal{I}})$.
  // Reset time missed
  $missed[\mathcal{I}] = 0$
  // Take a step with step-size $h$
  $\theta_{k+1}, m_{k+1} \leftarrow$ Solve Equations (5a) and (5b) with step-size $h$ with $(\theta_k, m_k, \{X_{k+\epsilon}^i, U_{k+\epsilon}^i\}_{i \in \mathcal{I}})$.
  $X_{k+1}^\mathcal{I}, u_{k+1}^\mathcal{I} \leftarrow$ Solve Equations (5c) and (5d) with step-size $h$ with $(\theta_k, m_k, \{X_{k+\epsilon}^i, U_{k+\epsilon}^i\}_{i \in \mathcal{I}})$..
  // Increment missed time for particles that are not updated
  $missed[\text{not in } \mathcal{I}] = missed[\text{not in } \mathcal{I}] + 1$.

---

## J.3  HEURISTIC

We (partially) circumvent the choice of momentum parameters by leveraging the relationship between Equation (1) and NAG to define a heuristic. For completeness, we briefly describe the heuristic here. For a given step-size $h$, one can define the "momentum coefficient" $\mu_\theta = 1 - h\gamma_\theta \eta_\theta$ (and similarly, for $\mu_x$). Since, in NAG, $\mu$ has typical values, we can use say $\mu_\theta = 0.9$ with a fixed value of $\gamma_\theta$ to find a suitable value of $\eta_\theta$ (and similarly, for $\eta_x, \gamma_x$). In our experiments, we found that $\gamma_\theta \in [\frac{1}{2}, 5]$

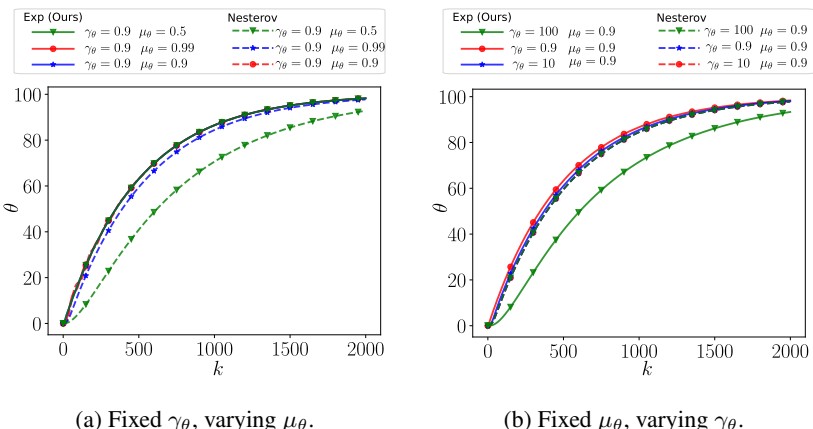

(a) Fixed $\gamma_\theta$, varying $\mu_\theta$.

(b) Fixed $\mu_\theta$, varying $\gamma_\theta$.

Figure 4: The effect of the damping parameter $\gamma_\theta$ and momentum coefficient $\mu_\theta$ on MPGD.

performed well. Another possible approach to handling hyperparameters is to borrow inspiration from adaptive restart methods (O'Donoghue and Candes, 2015). While some practical heuristics exist, it seems that to a large extent, the problem of tuning these hyperparameters remains open; we leave this topic for future work.

In Appendix I.3.1, we show how we can discretize Equation (2) to obtain the following update in the $(\theta, m)$-components:

$$\theta_k = \theta_{k-1} + v_k,$$
$$v_k = \mu v_{k-1} - h^2 \nabla_\theta \mathcal{E}(\theta_{k-1} + \mu v_{k-1}, q_{k-1,X}^M),$$

where $\mu_\theta = 1 - h\gamma_\theta\eta_\theta$, and $h > 0$ is the step size. We refer to the resulting iterations as MPGD-NAG-Cheng. This is exactly the update used in Sutskever et al. (2013)'s characterization of NAG with step-size $h^2$. Since there are some well-accepted choices for choosing $\mu_\theta$ in NAG (e.g., Nesterov (1983) advocating for $\mu_t = 1 - 3/(t+5)$), we can leverage the formula $\mu_\theta = 1 - h\gamma_\theta\eta_\theta$ to define a suitable choice of $\eta_\theta$ for a fixed $\gamma_\theta$. One can, of course, fix $\eta_\theta$ instead, but we found that suitable values for $\gamma_\theta$ are easier to choose from than $\eta_\theta$. We note that for MPGD-NAG-Cheng, it depends only on the value of $\mu_\theta$ which is not the case for our discretization. In Figure 4, we show the effect of varying the momentum coeffect $\mu_\theta$, while keeping the damping parameter $\gamma_\theta$ fixed (and vice versa). In Figure 4a, it can be seen that for a fixed damping parameter $\gamma_\theta$ and varying the momentum coefficient MPGD-NAG-Cheng changes significantly while ours does not. As for the vice versa case, in Figure 4b, it can be observed that MPGD-NAG-Cheng does not change while ours varies significantly instead. This suggests that the momentum coefficient in our discretization no longer has the same interpretation as that in MPGD-NAG-Cheng. Nonetheless, it can be observed that for a suitably chosen $\gamma_\theta$ and momentum parameter $\mu_\theta$, our discretization performs better than MPGD-NAG-Cheng. Hence, we use this as a heuristic when choosing momentum parameters $(\gamma_\theta, \gamma_x, \eta_\theta, \eta_x)$.

## J.4 EXPERIMENTAL DETAILS

Here, we outline for reproducibility the settings: for the Toy Hierarchical Model (Appendix J.4.1); image generation experiment (Appendix J.4.2); and, additional figures are in Appendix J.6. Much akin to Kuntz et al. (2023, Section 2), we found it beneficial to consider separate time-scales for the $(\theta_t, m_t)$ and $(X_t, U_t)$ which we denote $h_\theta$ and $h_x$ respectively.

### J.4.1 TOY HIERARCHICAL MODEL

**Model** The model is described in Kuntz et al. (2023, Example 1). For completeness, we describe it here. The model is defined by

$$p_\theta(x, y) = \prod_{i=1}^{d_x} \mathcal{N}(y_i; x_i, 1)\mathcal{N}(x_i; \theta, \sigma^2). \tag{66}$$

Thus, for the log-likelihood with $\sigma^2 = 1$, we have

$$\log p_\theta(x, y) = C - \frac{1}{2} \sum_{i=1}^{d_x} \left( (x_i - \theta)^2 + (y_i - x_i)^2 \right),$$

where $C$ is a constant independent of $\theta$ and $x$.

**Figure 1a**. For the different regimes, we use the following momentum parameters:

- **Underdamped**: $\gamma_\theta = 0.1, \eta_\theta = 2K_{hm}, \gamma_x = 0.1, \eta_x = 2K_{hm}$
- **Overdamped**: $\gamma_\theta = 1, \eta_\theta = 2K_{hm}, \gamma_x = 1, \eta_x = 2K_{hm}$
- **Critically Damped** $\gamma_\theta = 0.7, \eta_\theta = 2K_{hm}, \gamma_x = 0.7, \eta_x = 2K_{hm}$

where $K_{hm}$ is the Lipschitz constant of Assumption 2 (can be found at the end of Appendix J.4.1), and stepsizes are $h_\theta = 10^{-3}/N = 10^{-5}$, $h_x = 10^{-3}$, with number of samples $N = 100$, and number of particles $M = 100$.

**Figure 1b**. We compare the discretization outlined in Appendix I.3, and Section 5. We keep all parameters equal except for changing the momentum coefficient $\mu$. For step-sizes, we have $h_\theta = 10^{-5/2} \approx 5.8 \times 10^{-3}, h_x = 10^{-3}, \gamma_x = \gamma_\theta = 0.5$, with the momentum coefficient set to $\mu_\theta = \mu_x = \mu$ where $\mu \in \{0.9, 0.8, 0.5\}$.

**Figure 1c**. The momentum parameters are given by $\gamma_x = \gamma_\theta = 0.293$ and $\eta_\theta = \eta_x = 2K$ where $K_{hm}$ is the Lipschitz constant of Assumption 2 (can be found at the end of Appendix J.4.1), and step sizes are $h_\theta = 10^{-5/2} \approx 5.8 \times 10^{-3}, h_x = 10^{-3}$

**Lipschitz Constant and Strong log concavity.** For the toy Hierarchical model with $\sigma^2 = 1$, one can show that it satisfies our Lipschitz assumption, as well as being strongly log-concave. We have

$$\nabla^2_{(\theta,x)} \log p_\theta = \begin{pmatrix} -d_x & 1_{1 \times d_x} \\ 1_{d_x \times 1} & -2I_{d_x} \end{pmatrix}$$

The characteristic equation can be written as

$$\det(\nabla^2_{(\theta,x)} \log p_\theta - lI_{d_x+1}) = 0.$$

The determinant can be evaluated by expanding the minors. Thus, we obtain

$$\begin{aligned} \det(\nabla^2_{(\theta,x)} \log p_\theta - lI_{d_x+1}) &= (-d_x - l)(-2 - l)^{d_x} - d_x(-2 - l)^{d_x-1} \\ &= (-2 - l)^{d_x-1}((2 + l)(d_x + l) - d_x) \\ &= (-2 - l)^{d_x-1}(l^2 + (2 + d_x)l + d_x) \end{aligned}$$

Hence, the characteristic equation is satisfied when $l \in \{-2, \frac{-(2+d_x)\pm\sqrt{d_x^2+4}}{2}\}$. Note that $\frac{-(2+d_x)-\sqrt{d_x^2+4}}{2} \leq -2$.

From the above calculations, it can be seen that $\nabla^2_{(\theta,x)} \log p_\theta$ is strongly-log concave, i.e., we have $\nabla^2_{(\theta,x)} \log p_\theta \preceq -2I$.

In order to calculate its Lipschitz constant of Assumption 2, by Nesterov (2003, Lemma 1.2.2), this is equivalent to finding a constant $K_{hm} > 0$ such that

$$\|\nabla^2_{(\theta,x)} \log p_\theta\| \leq K_{hm}.$$

Since $\nabla^2_{(\theta,x)} \log p_\theta$ is symmetric, then the matrix norm is given by the largest absolute value of the eigenvalue of $\nabla^2_{(\theta,x)} \log p_\theta$ (i.e., it's spectral radius).

Hence, we have $K_{hm} = \frac{(2+d_x)+\sqrt{d_x^2+4}}{2}$.

| Layers | Size | Stride | Pad |
|---|---|---|---|
| Input | 1x1x128 | - | - |
| 8x8 ConvT(ngf x 8), LReLU | 8x8x(ngf x 8) | 1 | 0 |
| 4x4 ConvT(ngf x 4), LReLU | 16x16x(ngf x 4) | 2 | 1 |
| 4x4 ConvT(ngf x 2), LReLU | 32x32x(ngf x 2) | 2 | 1 |
| 3x3 ConvT(3), Tanh | 32x32x3 | 1 | 1 |

Table 2: Generator network for the VAE used for CIFAR-10 (ngf = 256).

| Layers | Size |
|---|---|
| Input | $d_x$ |
| Linear($d_x$, 512), LReLU | 512 |
| Linear(512,512), LReLU | 512 |
| Linear(512,512), LReLU | 512 |
| Linear(512,$d_y$), Tanh | $d_y$ |

Table 3: Generator network for the VAE used for MNIST.

### J.4.2 IMAGE GENERATION

The dataset of both MNIST and CIFAR-10 is processed similarly. We use $N = 5000$ images for training and 3000 for testing. The model is updated 6280 times using subsampling with a batch size of 32. Hence, it completes 40 epochs. The model is defined as:

$$p_\theta(y,x) = \prod_{i=1}^{N} p_\theta(y^i, x^i),$$

where:

- The datum $(y^i, x^i) \in \mathbb{R}^{d_y} \times \mathbb{R}^{d_x}$ denotes a single image and its corresponding latent variable. For MNIST, we have $d_y = 28 \times 28 = 784$ and for CIFAR, we have $d_y = 32 \times 32 \times 3 = 3072$. For both datasets, we set $d_x = 64$. Thus, we have that $(y,x) \in \mathbb{R}^{d_y \times N} \times \mathbb{R}^{d_x \times N}$.

- For the VAE model, we have $p(y^i, x^i) = \mathcal{N}(y^i|g_\psi(x^i), \sigma^2 I)p_\phi(x^i)$, where $\psi$ and $\phi$ are parameters of the generator and prior respectively. Thus, the parameter of the model is given by $\theta = \{\psi, \phi\}$. We set $\sigma^2 = 0.1$. The following specifies the details regarding the generator $g_\psi$ and prior $p_\phi$:

    - **Generator** $g_\psi$. For CIFAR, we use a Convolution Transpose network (as in Pang et al. (2020)) shown in Table 2. For MNIST, we use a fully connected network specified in Table 3 with $d_{in} = d_x$ and $d_{out} = d_y$.
    - **Prior** $p_\phi(x)$. The prior is inspired by the VampPrior (Tomczak and Welling, 2018) and is defined as:

    $$p_\phi(x) = \frac{1}{K} \sum_{i=1}^{K} \mathcal{N}(x|\mu_\nu(z_i), \sigma_\nu^2(z_i)I_{d_z}),$$

    where $\mu_\nu, \sigma_\nu^2 : \mathbb{R}^{d_z} \to \mathbb{R}^{d_x}$ are neural networks (with parameters $\nu$) that parameterized the mean and variance; pseudo-points $\{z_i\}_{i=1}^{K}$ are optimized; and so the prior parameters are $\phi := \{\nu\} \cup \{z_i\}_{i=1}^{K}$. The architecture can be found in Table 4 where $d_{in} = d_z$ and $d_{out} = d_z$

| Layers | Size |
|---|---|
| Input | $d_{in}$ |
| Linear($d_{in}$, 512), LReLU | 512 |
| Linear(512,512), LReLU | 512 |
| Linear(512,512), LReLU | 512 |
| Linear(512,$d_{out} \times 2$) | $d_{out} \times 2$ |
| Output: Id([:$d_{out}$]), Softplus([$d_{out}$:]) | $d_{out}, d_{out}$ |

Table 4: Neural network parametrization of the mean and variance of a Gaussian Distribution used in VI and VAMPprior. "Id" is the identity function, and "[:$d_{out}$]" is a standard pseudo-code notation that refers to the operation that extracts the first $d_{out}$ dimensions and similarly for [$d_{out}$:].

## J.5 HYPERPARAMETER SETTINGS

Unless specified otherwise, we use the same parameters for both MNIST and CIFAR datasets. Our approach to selecting the step size involved first setting it to $10^{-3}$, then we would decrease it appropriately if instabilities arise.

- **MPGD**. For step sizes, we have $h_\theta = h_x = 10^{-4}$. The number of particles is set to $M = 5$. For the momentum parameters, we use $\gamma_\theta = \gamma_x = 0.9$ with the momentum coefficient $\mu_\theta = 0.95, \mu_x = 0$ (or equivalently, $\eta_\theta \approx 556, \eta_x \approx 11,111$ )for MNIST and $\mu_\theta = 0.95, \mu_x = 0.5$ (or equivalently, $\eta_\theta \approx 556, , \eta_x \approx 55,555$ ) for CIFAR. We use the RMSProp preconditioner (see Appendix J.1) with $\beta = 0.9$.

- **PGD**. We have $h_\theta = 10^{-4}, h_x = 10^{-3}$. The number of particles is set to $M = 5$. We use the RMSProp preconditioner (see Appendix J.1) with $\beta = 0.9$.

- **ABP**. We use SGD optimizer with step size $h_\theta = 10^{-4}$, and run a length 5 ULA chain 5 with step-size $h_x = 10^{-3}$. We found that RMSProp worked well in the transient period but failed to achieve better performance than SGD.

- **SR**. We used RMSProp optimizer with step size $h_\theta = 10^{-4}$, and run a length 5 ULA chain 5 with step-size $h_x = 10^{-3}$

- **VI**. We use RMSProp optimizer with a step size of $h = 10^{-5}$. We found that this resulted in the best performance. For the encoder, we use a Mean Field approximation as in Kingma and Welling (2014). The neural network is a 3-layer fully connected network with Leaky Relu activation functions. In the last layer, we use a softmax to parameterize the variance. See Table 4.

## J.6 ADDITIONAL FIGURES

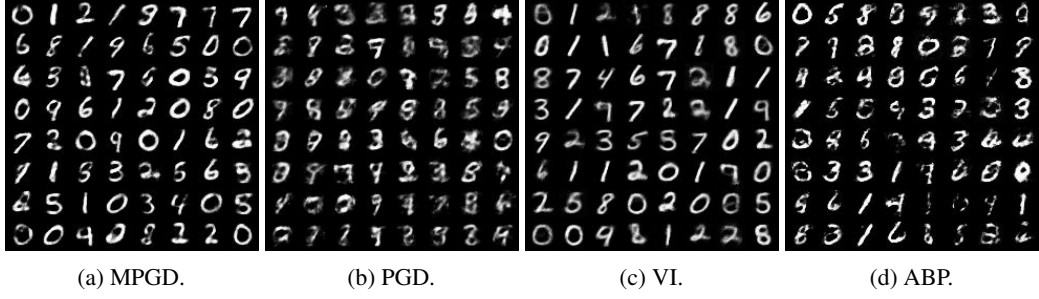

  (a) MPGD.      (b) PGD.      (c) VI.      (d) ABP.

Figure 5: **MNIST**. Samples generated from various algorithms.

## K    ADDITIONAL EXPERIMENTS

In this section, we show additional experiments particularly concerned with comparing MPGD with PGD and other variants of MPGD that only accelerated one component of the space instead of two. We say $X$-enriched to indicate algorithms with momentum incorporated in $X$ with either momentum included/excluded in $\theta$. To separate the two cases, we call $X$-only-enriched to refer to the algorithm with gradient descent in $\theta$. We use similar terminology for $\theta$-enriched and $\theta$-only-enriched.

We consider two settings: in Appendix K.1, further results with the toy Hierarchical model (see Appendix J.4.1); and, in Appendix K.2 a density estimation task using VAEs on a Mixture of Gaussian dataset.

### K.1    TOY HIERARCHICAL MODEL

**Figure 6.** As the model, we consider a Toy Hierarchical model for different $\sigma$ values in (66) where $\sigma$ is chosen to be the same as the data generating process. The data is generated from a toy Hierarchical model with parameters $(\theta = 10, \sigma)$ where $\sigma = \{5, 10, 12\}$. As noted by (Kuntz et al., 2023, Eq 51), the marginal maximum likelihood of $\theta$ is the empirical average of the observed samples, i.e., $\frac{1}{d_x} \sum_{i=1}^{d_x} y_i$. For each trial, we process the data $(y_i)_{i=1}^{d_x}$ to have an empirical average of 10 for simplicity.

**Figure 7.** We use a toy Hierarchical model with $\sigma = 12$, and, similarly, the data-generating processing is a toy Hierarchical model with $(\theta = 10, \sigma = 12)$. We are interested in how the initialization of the particle cloud affects MPGD, PGD and other algorithms that only momentum-enriched one component. Each subplot shows the evolution of the parameter $\theta$ for the initialization from $\mathcal{N}(\mu, I)$ where $\mu \in \{-5, -20, -100\}$. This example serves as an illustration of the typical case where the cloud is initialized badly. It can be seen that all methods are affected by poor initialization. For $X$-enriched, the algorithms can recover faster to achieve almost identical performances (at the later iterations) between different initializations. In other words, for $X$-enriched algorithms transient phase is short. Methods without such $X$-enrichment take longer to recover and are unable to achieve similar performances compared to the cases where they are better-initialized. Overall, it can be seen that MPGD perform better than PGD.

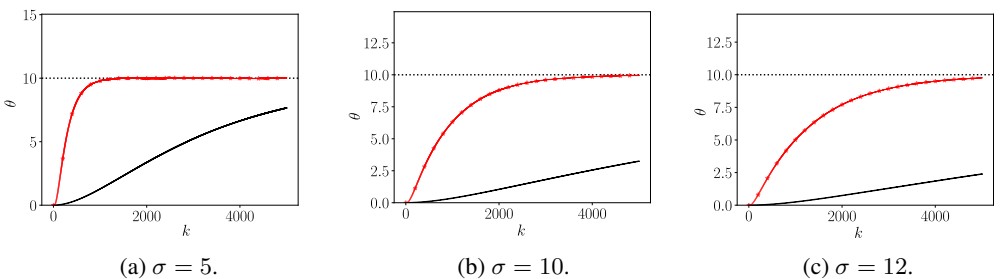

(a) $\sigma = 5$.      (b) $\sigma = 10$.      (c) $\sigma = 12$.

Figure 6: Comparison between MPGD and PGD on the Toy Hierarchical Model for different choices of $\sigma$. The plot shows the average and standard deviation of $\theta$ across iterations computed over 10 independent trials. MPGD is shown in red and PGD in black. The dashed line shows the true value.

### K.2    DENSITY ESTIMATION

In this problem, we show the result of training a VAE with VAMP Prior on a 1-d Mixture of Gaussians (MoG) dataset with PDF shown in Figure 8a. The dataset is composed of 100 samples. In Figure 8b, we show the performance of various acceleration algorithms. PGD is shown in black, and MPGD is shown in green. In this case, all accelerated algorithms perform better than PGD, and MPGD performs best of all.

**Hyperparameters**: No subsampling used. We have $d_y = 1, d_x = 10$. We set $\gamma_x = \gamma_\theta = 0.4$, and use the momentum heuristic with $\mu_\theta = \mu_x = 0.1$ (see Appendix J.3). We use the same VAE

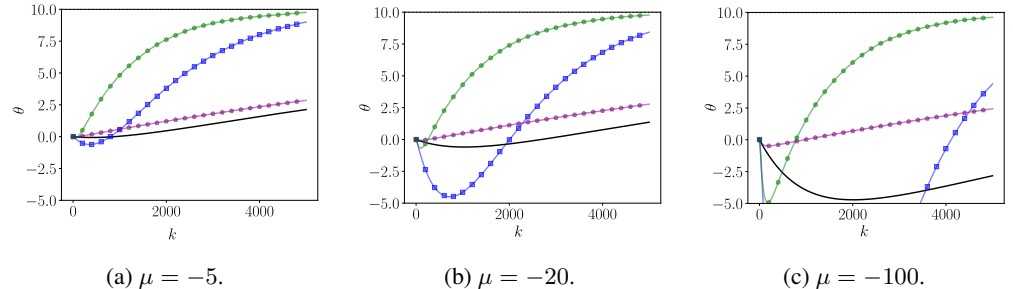

(a) $\mu = -5$.        (b) $\mu = -20$.        (c) $\mu = -100$.

Figure 7: The performance of various algorithms for the Toy HM with fixed $\sigma = 12$. For each subplot, we initialize the particle cloud with different means, i.e., $\{X_t^i\}_{i=1}^M \sim \mathcal{N}(\mu, I)$. MPGD is shown in green, $\theta$-only-enriched shown in blue, $X$-only-enriched in purple, and PGD in black. The true value is $\theta = 10$.

architecture in Appendix J.4.2 with likelihood noise $\sigma = 0.01$, an MLP decoder (see Table 4), and VAMP prior with $d_z = 2$ and $K = 20$.

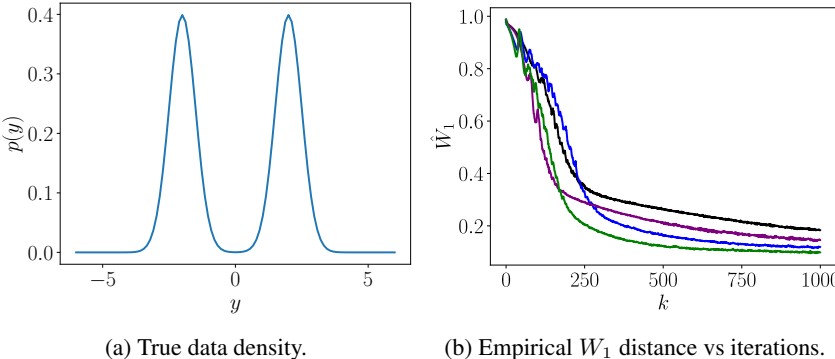

(a) True data density.     (b) Empirical $W_1$ distance vs iterations.

Figure 8: Performance of various methods on a density estimation task. In Figure 8a, we show the true data-generating process. In Figure 8b, we show the empirical Wasserstein-1 distance (denoted by $\hat{W}_1$) estimated from 1000 samples across each iteration. The results show the average (with standard error bars which are barely noticeable) computed over 100 trials. MPGD is shown in green, $\theta$-only-enriched shown in blue, $X$-only-enriched in purple, and PGD in black.

