# OpenReview forum: "Momentum Particle Maximum Likelihood"
_ICLR.cc/2024/Conference — Submitted to ICLR 2024_

### Official Review · Reviewer_1aSJ · 2023-10-18

**Soundness:** 2 fair
**Presentation:** 3 good
**Contribution:** 2 fair
**Rating:** 5
**Confidence:** 3

**Summary:**

The Expectation Maximization (EM) algorithm applied to latent variable models has previously been seen as a gradient flow in the joint space of parameters and probability distributions. Previous research has also found that representing a probability distribution as particles simplifies the computation of gradients and the resulting gradient flow algorithm. These have been presented in the literature as Stein Variational Gradient Descent (SVGD) or Particle Gradient Descent (PGD).

This work seeks to accelerate the gradient step in PGD by infusing it with Nesterov momentum which is a well known technique that can be applied to any gradient descent algorithm. The authors provide the following technical contribution in this work
1) mathematical justification for their momentum infused gradient flow in this joint space (of parameters and probability distributions).
2) a proof of convergence of their gradient flow algorithm.
3) a partial update discretization for faster convergence.

In addition the authors show results on two datasets. One trivial dataset which demonstrates the effectiveness of their approach by including an ablation study of the partial update. The second dataset is a more realistic problem of building a Variational Auto Encoder where the momentum method described here seems to work very effectively.

**Strengths:**

Gradient flow algorithms in the Wasserstein-2 distance infused metric spaces over probability measures have suddenly become very trendy and as such this paper would certainly be of significant interest to the audience of this conference. The authors have taken a current algorithm, PGD, in this area and demonstrated through their experiments that the provided enhancement is an improvement. They also demonstrate that the addition of momentum is done in a theoretically sound manner, i.e. it "reproduces the dynamics of a damped harmonic oscillator".

The authors provide a slightly improved momentum method that makes partial updates and they provide a theoretical proof of convergence. The partial update definitely appears to be original and the results on the toy dataset show that this partial update does have an impact on faster convergence.

The results on the Variational Auto Encoder suggest that their method is at least as good if not better than the nearest competing method.

**Weaknesses:**

The idea of adding momentum to gradient updates is very well known and it doesn't require deep theoretical justification to come up
with Equations 2a - 2d which follow quite obviously from PGD and the definition of Nesterov accelerated momentum.

The authors do provide a partial update discretization of the gradient flow which is not very obvious and the toy example does justify the partial updates. However, a toy example is not sufficient to prove the value of this novelty in the paper. I would refer to the paper https://arxiv.org/abs/2103.00065, "Gradient Descent on Neural Networks Typically Occurs at the Edge of Stability" which shows that conclusions that are made on toy examples are quite the opposite of those in real world problems.

Showing convergence is valuable, but most gradient flow algorithms have a natural recipe to provide convergence under strong convexity and/or smoothness assumptions. So, this is not really a novelty here. It does go to overall quality of the paper though.

If we were to simply add a momentum step to every paper with a gradient update algorithm and show some results where it performs better then we would have a deluge of papers! For example in the Coin EM paper there is a SVGD-like gradient step for updating the density represented by particles. Now, one could certainly add momentum to that, does that make it a novel contribution?

I would readily admit that if adding momentum to an existing algorithm gives an improvement then it is worth disseminating. In my opinion though such contributions don't reach the level of originality demanded by a conference publication.

Minor:

typo: Confernece on page 10 in the citation for Kuntz 2023

typo: on page 9 AGPD -> MPGD

I would suggest some rearrangement to ensure that the inequalities in 35 are included in the main text.

**Questions:**

How does the partial update discretization compare to a naive momentum step for the Variational Auto Encoder problem?

How does this algorithm compare to the performance of other current particle-based approaches such as SVGD-like approaches. For example, the authors have cited the Coin EM paper which was published very recently and which provides two such algorithms (on a host of interesting datasets)?

---

> ### Author Response · Authors · 2023-11-20
>
> Thank you for your suggestions and comments. Here is our response:
>
> > The idea of adding momentum to gradient updates is very well known and it doesn't require deep theoretical justification to come up with Equations 2a - 2d which follow quite obviously from PGD and the definition of Nesterov accelerated momentum.
>
> While we agree that the idea of adding momentum is well-known, we disagree that it follows obviously from PGD and Nesterov's Accelerated Gradient (NAG) algorithm. What may be immediate from PGD and NAG is a version of MPGD that only accelerates one component, which does not perform as well as MPGD (see replies to yQ9T). Furthermore, there are multiple methods to incorporate momentum into PGD, for example, through the principle of least action [1]. Taking this perspective, the resulting algorithm is difficult to simulate and so was discarded [2, see Appendix D]. In our work, we advocate for the vector field perspective (as seen in [3]) as a suitable perspective for incorporating momentum into gradient flow algorithms in the product space $\mathbb{R}^{d_\theta}\times \cal{P(\mathbb{R}^{d_x})}$.
>
> > Showing convergence is valuable, but most gradient flow algorithms have a natural recipe to provide convergence under strong convexity and/or smoothness assumptions. So, this is not really a novelty here. It does go to overall quality of the paper though.
>
> We agree that arguing convergence for gradient flows under strong convexity assumptions is straightforward and not a substantial contribution. However, our MPGD flow in Equation 2 is not a gradient flow. This is due to the fact that it cannot be written as $\dot{z} = -\nabla F(z)$ for some $F$ and suitable notation of gradient $\nabla$. As a result, proving convergence does not follow from standard gradient flow theory and can be seen in Appendix F to be far from trivial.
>
> We stress that when working in a momentum-enriched system, establishing convergence does not follow the same recipe as for 'pure' gradient flows. One of the key differences between gradient flows and momentum-enriched flows is that we can only show that the objective is non-increasing (which does not establish convergence). A different strategy is needed for momentum momentum-enriched system. One usually has to construct a Lyapunov function that couples the momentum variable with the variable of interest (for example, under convexity, see [4, Theorem 3]; and under the log-Sobolev assumption, see [3, Proposition 1]). We note again that the proof under log-Sobolev assumptions, see [3, Appendix C] is far from trivial. Similarly, the bulk of our appendix is required to prove this fact, see Appendix F. Specifically, this is done by a generalization of the argument in [3] to product space. An example of a unique extension is that we have to control an interaction term that occurs from the interaction in the product space (see Appendix F.4.6).

---

> ### Author Response · Authors · 2023-11-20
>
> > If we were to simply add a momentum step to every paper with a gradient update algorithm and show some results where it performs better then we would have a deluge of papers!
>
> We agree that the general idea of accelerating PGD using momentum is natural and, on its own, need not count as exceedingly novel. The novelty (and the challenge) lies in how exactly this should be done. Should one add momentum to the parameters? To the particles? To both? Should one do so at the level of $(\theta,q)$ or at that of $(\theta,x)$? What discretization should one use? Should one use full or partial updates? And so on.
>
> As such one of our implicit contributions is that we have provided a roadmap for others to incorporate momentum in this space. We have shown how one might proceed to derive and study momentum methods in this space which particularly requires care at all stages. Specifically, the stages where the process can go awry are:
>
> * Derivation of the flow: As mentioned previously, there are alternative recipes that one can employ. These will lead to different continuous-time dynamical systems - some are more practical than others. We show that taking the vector field perspective when deriving a recipe for a momentum-enriched system is a reasonable place to start.
> * Proving convergence: As mentioned previously, momentum-enriched algorithms are not gradient flows, and so establishing convergence is not easy, due to the presence of interaction terms between components in different spaces (which we note is not typically a concern for typical momentum-enriched methods), there is a question whether the Lyapunov approach is sufficient to prove convergence of the flow, and if so what will the Lyapunov function look like? We answer in the affirmative and provide hints for future work on what to choose as a suitable Lyapunov function.
> * Discretization of the flow: As mentioned by eLbr, discretization of momentum-enriched systems requires care, and, as you have mentioned, the partial update is crucial for stable discretization. This idea may also be applied to other momentum-enriched systems in this product space.
>
> Given our work, we believe that questions concerning i) how to incorporate momentum into algorithms in this product space, ii) how to study the convergence of such algorithms, as well as iii) how to discretize such algorithms are now made clearer. As such, we believe that these contributions are worthy of dissemination.
>
> > How does the partial update discretization compare to a naive momentum step for the Variational Auto Encoder problem.
>
> We performed this experiment, keeping all the hyperparameters the same as before, and as suggested by the toy example, we ran into numerical instabilities. This occurred in the first couple of iterations; there is not much to add beyond this.
>
> > How does this algorithm compare to the performance of other current particle-based approaches such as SVGD-like approaches.
>
> In our experiment, we compare against other particle methods such as Alternating Backpropagation and Short Run MCMC. We chose not to compare SVGD since it incurs an $\mathcal{O}(M^2)$ cost instead of $\mathcal{O}(M)$ like ours and our other baselines.
>
> [1] Wibisono, Andre, Ashia C. Wilson, and Michael I. Jordan. "A variational perspective on accelerated methods in optimization." proceedings of the National Academy of Sciences 113.47 (2016): E7351-E7358.
> [2] Taghvaei, Amirhossein, and Prashant Mehta. "Accelerated flow for probability distributions." International Conference on Machine Learning. PMLR, 2019.
> [3] Yi-An Ma. Niladri S. Chatterji. Xiang Cheng. Nicolas Flammarion. Peter L. Bartlett. Michael I. Jordan. "Is there an analog of Nesterov acceleration for gradient-based MCMC?." Bernoulli 27 (3) 1942 - 1992, August 2021. https://doi.org/10.3150/20-BEJ1297
> [4] Su, Weijie, Stephen Boyd, and Emmanuel Candes. "A differential equation for modeling Nesterov’s accelerated gradient method: theory and insights." Advances in neural information processing systems 27 (2014).

---

### Official Review · Reviewer_eLbr · 2023-10-29

**Soundness:** 3 good
**Presentation:** 2 fair
**Contribution:** 2 fair
**Rating:** 5
**Confidence:** 3

**Summary:**

This paper aims to improve the convergence speed of the algorithm by extending the method that was the Underdamped Langevin dynamics (Mckean-Vlasov process) in the particle-based approach recently proposed for the EM algorithm in latent variable models to the Overdamped form. The author develops the theory and formulation from the perspective of gradient flow in extending the Underdamped Langevin dynamics (Mckean-Vlasov process).

**Strengths:**

In this paper, the authors successfully extended the conventional Overdamped to an Underdamped approach of the Mckean-Vlasov process for latent variable models. They base their discussion and formulation on the principles of Gradient flow. Importantly, they demonstrate performance enhancements when their method is applied to real data.

**Weaknesses:**

I have concerns about whether the discussion on discretization in this paper is sufficient. There are two main types of discretization involved: one for the expected values of the particles, and the other along the time direction. The paper only briefly mentions using Kuntz's method in Section 5. However, it’s important to note that while the proposed method is based on Underdamped dynamics, Kuntz’s approach is for Overdamped scenarios. In usual Langevin dynamics, not specifically Mckean-Vlasov processes, it’s well-known that not paying enough attention to time discretization in Underdamped cases can lead to unstable dynamics, which is a crucial issue for sampling. But this paper does not seem to discuss these potential problems.

Additionally, the paper does not provide enough information on how to choose parameters properly. In the field of Langevin dynamics, it’s recognized that using Underdamped dynamics can lead to faster results if the parameters, like the momentum parameter, are chosen correctly. However, there is no clear guidance in the paper on how to achieve this acceleration in the context of the Mckean-Vlasov process.

**Questions:**

How should we choose the momentum parameter? Additionally, due to variations in this choice, how much of a difference in performance can we expect with real data, as opposed to a toy model?

---

> ### Author Response · Authors · 2023-11-20
>
> Thank you for your comments.
>
> >  I have concerns about whether the discussion on discretization in this paper is sufficient ... But this paper does not seem to discuss these potential problems.
>
> We agree that naive discretisation of Underdamped Langevin dynamics can fail to yield accelerated rates of convergence. In the underdamped Langevin case, one example of an integrator that yields accelerated rates is the Exponential Integrator (see [2,3]). Our proposed algorithm is a result of utilizing the Exponential integrator with a partial update. As noted by Reviewer 1aSJ, the stable partial update is crucial for a stable discretization. We have further expanded the discussion on this matter in the manuscript.
>
> > Additionally, the paper does not provide enough information on how to choose parameters properly. ... How should we choose the momentum parameter?
>
> In Appendix J.3, we discuss a heuristic that we used throughout our experiments. It is based on the relationship between our discretization and NAG. While we found that this heuristic was helpful in our problems, we recognize that this is not a complete solution. This is an issue we have inherited from Underdamped Langevin dynamics where it is generally understood to be somewhat more challenging than its overdamped counterpart, as witnessed by the dearth of papers proposing practical tuning strategies for this class of algorithms. [1] describes some strategies for practical tuning of the friction parameter based on heuristics adapted to Gaussian targets, as well as tuning the step size by monitoring energy errors (in their context, acceptance rates in their Metropolis-Hastings step). In our work, we have adopted some ad-hoc strategies based on empirical monitoring of the algorithm; future work could seek to carefully extend the proposed strategies of [1] to the present context. We will expand on this in the manuscript.
>
> [1] Riou-Durand, Lionel, et al. "Adaptive Tuning for Metropolis Adjusted Langevin Trajectories." International Conference on Artificial Intelligence and Statistics. PMLR, 2023.
> [2] Cheng, Xiang, et al. "Underdamped Langevin MCMC: A non-asymptotic analysis." Conference on learning theory. PMLR, 2018.
> [3] Sanz-Serna, Jesus Maria, and Konstantinos C. Zygalakis. "Wasserstein distance estimates for the distributions of numerical approximations to ergodic stochastic differential equations." The Journal of Machine Learning Research 22.1 (2021): 11006-11042.

---

### Official Review · Reviewer_cMKu · 2023-10-30

**Soundness:** 2 fair
**Presentation:** 3 good
**Contribution:** 2 fair
**Rating:** 5
**Confidence:** 4

**Summary:**

The authors propose a particle-based optimization method for Maximum Likelihood Estimation (MLE) of latent variable models. This method is a momentum variant of the particle gradient descent (PGD) method, which represents the gradient flow over the product space of Euclidean space and the space of probability distributions equipped with Wasserstein geometry. Specifically, the authors present the particle dynamics, known as the McKean-Vlasov process, and the corresponding continuity equation of the proposed method. Furthermore, they conduct a convergence analysis under the conditions of Lipschitz smoothness and extended log-Sobolev inequality. Experimental results validate the advantages of this proposed method.

**Strengths:**

This is the first study of the particle-based momentum method for solving MLE of latent variable models. The authors extend the analysis of the PGD method by Kuntz et al. (2023) into the momentum setup.

**Weaknesses:**

1. The momentum method is usually used for acceleration. However, the theoretical advantage, such as an improved convergence rate over PGD, has not been discussed. Indeed, we cannot see the benefit of the convergence analysis (Proposition 4.1) compared to the PGD method.

2. In light of the theoretical work on sampling and particle-based optimization methods, the provided analysis seems somewhat weak. For instance, the existence and smoothness of the solution of SDE (2a)-(2d), and any guarantees of the discretization (in time and space), are not provided.

3. The important assumptions should be exposed in the main text, and their reasonability should be discussed. However, Assumptions 1 and 2 (Lipschitz smoothness and LSI) are hidden in the Appendix. It is better that the authors provide an example that satisfies all assumptions to convince the readers that the theory is not vacuous.

4. The authors missed the series of mean-field optimization works. For instance, see the following papers and references therein:
- [Mei, Montanari, and Nguyen (2018)] A mean-field view of the landscape of two-layer neural networks.
- [Chizat and Bach (2018)] On the global convergence of gradient descent for over-parameterized Convex Analysis of the Mean Field Langevin Dynamics models using optimal transport.
- [Nitanda and Suzuki (2017)] Stochastic particle gradient descent for infinite ensembles.
- [Hu, Ren, Siska, and Szpruch (2019)] Mean-field Langevin dynamics and energy landscape of neural networks.
- [Nitanda, Wu, and Suzuki (2022)] Convex Analysis of the Mean Field Langevin Dynamics.
- [Chizat (2022)] Mean-field langevin dynamics: Exponential convergence and annealing.
- [Chen, Ren, and Wang (2023)] Uniform-in-time propagation of chaos for mean field Langevin dynamics.
- [Suzuki, Wu, and Nitanda (2023)] Convergence of mean-field Langevin dynamics: Time and space discretization, stochastic gradient, and variance reduction

Minor comments:
- Page 4: notation $\ell$ is undefined.
- Equation (1) is missing.

**Questions:**

Is it possible to discuss the time and particle complexities to achieve a given optimization precision? Such an analysis is quite standard in the optimization/sampling literature.

---

> ### Author Response · Authors · 2023-11-20
>
> We thank the reviewer for their comments.
>
> > The momentum method is usually used for acceleration. However, the theoretical advantage, such as an improved convergence rate over PGD, has not been discussed. Indeed, we cannot see the benefit of the convergence analysis (Proposition 4.1) compared to the PGD method.
>
> Since the convergence analysis of PGD does not extend to MPGD, the role of Proposition 4.1 is largely to establish convergence, rather than to provide evidence that MPGD is more performant than PGD. As noted by 1aSJ, gradient flow algorithms have a straightforward recipe for convergence, but since MPGD is not a gradient flow algorithm, the convergence proof is considerably more involved. If interested, please see our reply to 1aSJ where we are more explicit about the differences between establishing convergence for momentum-enriched methods and gradient flows.
>
> > For instance, the existence and smoothness of the solution of SDE (2a)-(2d), and any guarantees of the discretization (in time and space), are not provided.
>
> In the revised manuscript, we provided additional theoretical results proving the existence and uniqueness of the solutions with the appropriate extension, as well as asymptotic guarantees of the space discretization. The proof of existence and uniqueness of the solutions (as well as the space discretization) require an extension of McKean-Vlasov SDE arguments to the product space that is not immediate from the current theory. The statement of the results can be found in Proposition 3.1 and Proposition 5.1 with the proofs in Appendix G and H.
>
> > The important assumptions should be exposed in the main text, and their reasonability should be discussed. However, Assumptions 1 and 2 (Lipschitz smoothness and LSI) are hidden in the Appendix.
>
> We agree with this and have made the necessary changes (see Section 4).
>
> > The authors missed the series of mean-field optimization works
>
> Thank you for your references. We have added a section in the Appendix dedicated to related work which included your reference (see Appendix B).

---

> ### Author Response · Authors · 2023-11-20
>
> > It is better that the authors provide an example that satisfies all assumptions to convince the readers that the theory is not vacuous.
>
> There is already an example in the paper that satisfies the assumptions: the toy hierarchical model in Section 6.1. Its log-likelihood is a quadratic polynomial of $(\theta,x)$, so the log-likelihood's gradient is linear and trivially Lipschitz. The log-likelihood is also strongly concave (cf. Appendix J.4.1), and the extended log Sobolev-Inequality holds for all models whose log-likelihoods are strongly concave (i.e., whose Hessian is upper bounded by $-\lambda$ times the identity matrix for some constant $\lambda>0$). The argument for this is as follows: if $(\theta_t,q_t)$ denote the solution to $\cal{E}$'s gradient flow (see Section 2.3 in our paper), then Eqs. (28,30) in [1] respectively read
>
> $$
> I(\theta_t, q_t) := -\frac{\mathrm{d}}{\mathrm{d}t} \mathcal{E}(\theta_t, q_t) =\|\nabla \mathcal{E}(\theta_t, q_t)\|^2_{q_t},\qquad
> \frac{\mathrm{d}}{\mathrm{d}t}I(\theta_t, q_t) \le -2\lambda I(\theta_t, q_t) .$$
>
> (In [1] the minus sign is missing from (30)'s RHS, but this is a typo as can be easily verified from the equation's proof.) The inequality implies that $I(\theta_t, q_t)\leq e^{-2\lambda t}I(\theta_0, q_0)$. Plugging this into (28) and integrating it over time we find that
>
> $$
>  \mathcal{E}(\theta_0, q_0) - \mathcal{E}(\theta_t, q_t) \le \|\nabla \mathcal{E}(\theta_0, q_0)\|^2_{ q_0}\left (\frac{1- \exp(-2\lambda t)}{2\lambda} \right ).\qquad(*)$$
>
> In [1, Theorem 3], it was argued that $(\theta_t,q_t)$ converges to $(\theta^*, p_{\theta^*}(\cdot|y))$ in a certain sense as $t$ tends to infinity. It is possible to strengthen this statement to $\lim_{t\to\infty} \mathcal{E}(\theta_t,q_t)=\mathcal{E}(\theta^*, p_{\theta^*}(\cdot |y))$. By $\cal{E}$'s definition, $\mathcal{E}(\theta^*, p_{\theta^*}(\cdot |y)) = - \log p_{\theta^*}(y)$ and, because the flow's initial condition $(\theta_0,q_0)$ was arbitrary, the LSI follows by taking the limit $t\to\infty$ in $(*)$.
>
> However, formally establishing the limit $\lim_{t\to\infty} \mathcal{E}(\theta_t,q_t)=\mathcal{E}(\theta^*, p_{\theta^*}(\cdot |y))$ requires a somewhat involved and lengthy study of the PGD gradient flow which lies beyond the scope of this paper (we are aware of an upcoming publication that covers this). Nevertheless, we hope the above sketch will persuade you that there are indeed many models that satisfy our assumptions and that our theoretical analysis is not ultimately vacuous.
>
> Aside: We conjecture that the extended log-Sobolev inequality is more general than strong log-concavity (as is the case for the classical LSI). This is motivated by the hints in [6, Proposition 1], which established log-Sobolev inequality for potentials that are strongly concave outside a ball. However, to establish this fact would require a non-trivial generalization of the widely-used Holley-Stroock comparison Theorem [7].
>
> > Is it possible to discuss the time and particle complexities to achieve a given optimization precision? Such an analysis is quite standard in the optimization/sampling literature.
>
> Sharp analyses of PGD and MPGD are not yet available, so we cannot theoretically guarantee this at present. While such an analysis may be standard in the optimization/sampling literature, it is not standard for algorithms that operate in the product space (see [1,2,3,4,5] to name a few).
>
> [1] Kuntz, J., Lim, J. N., & Johansen, A. M. (2023, April). Particle algorithms for maximum likelihood training of latent variable models. In International Conference on Artificial Intelligence and Statistics (pp. 5134-5180). PMLR.
> [2] Han, Tian, et al. "Alternating back-propagation for generator network." Proceedings of the AAAI Conference on Artificial Intelligence. Vol. 31. No. 1. 2017.
> [3] Sharrock, Louis, Daniel Dodd, and Christopher Nemeth. "CoinEM: Tuning-Free Particle-Based Variational Inference for Latent Variable Models." arXiv preprint arXiv:2305.14916 (2023).
> [4] De Bortoli, Valentin, et al. "Efficient stochastic optimisation by unadjusted Langevin Monte Carlo: Application to maximum marginal likelihood and empirical Bayesian estimation." Statistics and Computing 31 (2021): 1-18.
> [5] Nijkamp, Erik, et al. "Learning multi-layer latent variable model via variational optimization of short run mcmc for approximate inference." Computer Vision–ECCV 2020: 16th European Conference, Glasgow, UK, August 23–28, 2020, Proceedings, Part VI 16. Springer International Publishing, 2020.
> [6] Ma, Yi-An, et al. "Sampling can be faster than optimization." Proceedings of the National Academy of Sciences 116.42 (2019): 20881-20885.
> [7] Holley, Richard, and Daniel W. Stroock. "Logarithmic Sobolev inequalities and stochastic Ising models." (1986).

---

### Official Review · Reviewer_yQ9T · 2023-10-31

**Soundness:** 3 good
**Presentation:** 3 good
**Contribution:** 3 good
**Rating:** 8
**Confidence:** 3

**Summary:**

The paper considers the classical problem of MLE in latent variable models. It proposes an extension of a recent work of Kuntz et al (2023, AISTATS) that itself exploits Neal & Hinton's perspective that classical EM (and variational EM) perform coordinate descent on the energy functional $\mathcal{E}(q, \theta)$ to have flexible nonparametric $q$ with convergence guarantees to the MLE. The extension here proposes to incorporate momentum methods typically used in stochastic optimization and popular in deep learning withing Kuntz's et al framework. It also derives the corresponding dynamical system in continuous time corresponding to the algorithm to establish convergence guarantees as Kuntz et al.

**Strengths:**

- The framework of Kuntz et al 2023 is interesting and impactful, and extending it is a worthwile direction which this paper addresses
- The paper is clear and fairly honest in its contribution, with a clear objective

**Weaknesses:**

- Motivation for this particular extension. While it is natural to consider "momentum enriched" optimization algorithms in this context, I am left wondering whether there is a **strong** motivation, precisely: are there problems where PGD actually **struggles**,  *and* the new MPGD (even if well-tuned) really resolves the situation ? This is not very clear from the current version of the paper. Yes, we "can" do this extension and it's natural, and it seems to perform better (focussing for now on the toy example, where metrics are more interpretable and relate directly to the MLE problem) than PGD, but the toy example only shows a (relatively minor) *improvement* over PGD which is already working well. Could you think of an example where the benefits of MPGD vs PGD really show, i.e., PGD is converging very slowly and MPGD "fixes" the problem ?  Again in other words, the authors claim "we derive a discretization of the flow that can achieve better performance than PGD ...", and this is supported by the experiments. It "can be better" indeed, but how relevant is that? It would be stronger to show cases where PGD doesn't do well and MPGD really does (as opposed to cases where they both do well and MPGD does a bit better).
- This point is related to the above; in the toy experiment, there is not a significant (in the sense of practically, not statistically) difference between MPGD and PGD (Figs 1(a) and (b) ) except for the underdamped and overdamped regimes (which presumably are less desirable). Can you try with different parameters (e.g. variances of the hierarchical model) to see if there is a setting where MPGD really shows its strength ?

**Questions:**

- In the second paragraph of Sec 2.2, "Under suitable assumptions, one can then compute that ..." What does the $< >$ denote here (it was not defined before) ( I am guessing it is an inner product, but it should be clearly defined) - can you provide a derivation for this in the Appendix, even if standard ? What are the assumptions required for the vector field $v$
- In the last paragraph before 2.3 starts, what does the symbol $\bigotimes$ denote ?
- You enrich both components of the space with momentum variables but why ? Is one of the two not enough ?
- Regarding convergence of the flow, can you compare (explicitly) your results to those of Kuntz et al ? It is not cited at all in Sec 4. It is also quite unclear how informative these rates are, as there is no explicit dependence on $M$.

Note that I did not carefully check the correctness of Section 4 and the related proofs in the Appendix.

---

> ### Author Response · Authors · 2023-11-20
>
> Thank you for your comments and questions.
>
> > While it is natural to consider "momentum enriched" optimization algorithms in this context, I am left wondering whether there is a strong motivation, precisely: are there problems where PGD actually struggles, and the new MPGD (even if well-tuned) really resolves the situation?
>
> We find that in many cases, MPGD does indeed perform substantially better than the vanilla PGD. Originally, we aimed to demonstrate this through the VAE example (for instance, Table 1), which other reviewers agree demonstrates its performance gains
>
> > [1aSJ] The second dataset is a more realistic problem of building a Variational Auto Encoder where the momentum method described here seems to work very effectively.
>
> The toy hierarchical example was intended to illustrate various aspects of MPGD (e.g., its regimes) and validate its design (in particular, the specific discretization we use and the gradient correction).
>
> Still, you are entirely correct: by simply increasing the model's variance, the performance gains for MPGD relative to PGD again become large for this example; see Appendix K. As the variance of the model increases, PGD struggles. Additionally, we have demonstrated empirically that PGD is more sensitive to initialization of the cloud and thus degrades with poor initialization. Thank you very much for this suggestion. As further evidence, we have added a new example (Appendix K.2). It is density estimation in $1$-d where the true density is a bi-model Gaussian. In 1-d, we can estimate the Wasserstein-$1$ ($W_1$) distance easily. In this problem, MPGD shows a significant improvement over PGD - the final $W_1$ of PGD is around $0.3$ and MPGD is around $0.1$ - a three-fold decrease.
>
> > Can you try with different parameters (e.g. variances of the hierarchical model)
>
> Thank you for your suggestion, this is a good idea. The results can be found in Appendix J.1. As the variance increases, it can be seen that PGD struggles to converge while MPGD still converges, thus performing significantly better. This can be seen in Figure 6, where we provide the requested experiment comparing PGD and MPGD on the Toy HM with different variances.
>
> > You enrich both components of the space with momentum variables but why ? Is one of the two not enough?
>
> We found that MPGD performed better than algorithms that only enrich one component (or none at all). The ordering can be seen to be MPGD > Momentum in $\theta$ $\approx$ Momentum in $X$ > PGD. The $\approx$ indicates that there are settings where one is better than the other and vice versa. To illustrate this observation, we provide two additional experiments (see Figure 7 and Figure 8 in Appendix K).
>
> In Figure 7, we use the Toy HM example with $\sigma=12$ and compare MPGD (and PGD) with algorithms that enrich one of the spaces with momentum. In Figure 7, we show the effect of initialization of the particle cloud on each method. Overall, it can be seen that the presence of momentum results in a better performance. Furthermore, we observe that the algorithms which only enrich one component can exhibit problems which are not present for MPGD. For instance, in Figure 7, poor initialization of the cloud drastically lengthens the transient phase of the $\theta$-only enriched algorithm, whereas MPGD can recover more rapidly. This is a typical setting in high-dimensional settings (see Figure 3). Conversely, $X$-only-enriched algorithms are observed to suffer from slow convergence in the $\theta$ component (intuitively, one pays a larger price for poor conditioning of the ideal MLE objective). Overall, it can be observed that MPGD noticeably outperforms PGD and all other intermediate methods on all settings.
>
> In Figure 8, as a more realistic example, we provide an additional experiment on a density estimation task using VAEs on a $1$-dimensional problem. Figure 8b shows the empirical Wasserstein-$1$ distance between the model and the data throughout the training process. It can be seen that by doubly enriching the space (i.e. using MPGD), the transient period is noticeably shortened, attaining a lower $W_1$ than both PGD and the other algorithms which enrich only one of the spaces.

---

> > ### Author Response · Authors · 2023-11-20
> >
> > >  What does the <> denote here?
> > >   can you provide a derivation for this in the Appendix, even if standard
> >
> > Yes, it denotes an inner product. Thank you for pointing this out, we have now defined it. The derivation be found in [1, Lemma 10.4.1] where $v\in L^2$. We will expand on the exposition in the Appendix C.
> >
> > > What does the symbol $\otimes$ denote
> >
> > $p \otimes q$ refers to the product measure $p(\mathrm{d}x) \cdot q(\mathrm{d} y)$. Thank you for pointing this out, we have now defined it when it first appears.
> >
> > > Regarding convergence of the flow, can you compare (explicitly) your results to those of Kuntz et al ? It is not cited at all in Sec 4. It is also quite unclear how informative these rates are, as there is no explicit dependence on $M$.
> >
> > We believe that Prop. 4.1. should be viewed as evidence that the algorithm is doing something sensible, in constrast to viewing it as evidence that MPGD is better than PGD. Ultimately, we agree that comparison of these continuous-time rates provide somewhat limited information about the practical algorithm, as we do not account for discretization error. Equally, a discretization analysis is also not yet available for PGD, and so a rigorous comparison would require significant work studying the PGD flow, and thus lies outside the scope of the present work. We instead focus on empirical comparisons, studying a number of practical problems, for which we have consistently found that MPGD performs better than PGD.
> >
> > [1] Ambrosio, Luigi, Nicola Gigli, and Giuseppe Savaré. Gradient flows: in metric spaces and in the space of probability measures. Springer Science & Business Media, 2005.

---

> > ### Comment · Reviewer_yQ9T · 2023-11-22
> > **Response to authors**
> >
> > Thanks for the very thorough work.
> >
> > I appreciate the new experiments and pointing out to the relevant section in the Appendix.
> > I am not a fan of the VAE examples (very personal opinion) since they don't bring me any understanding how why the method is working better (too many moving parts), but I am gaining more understanding in Appendix K and in the density estimation example. My 2 cents is that these things are more valuable for understanding and are a bit hidden in the Appendix in favour of the VAE example in the main.
> > I will leave the judgement to the authors  to how to rearrange the content and increase my score to accept.

---

> > > ### Author Response · Authors · 2023-11-22
> > >
> > > Thank you again for your helpful comments and for the time and effort you spent going through all of this.
> > >
> > > We are glad that the additional experiments addressed your concerns. We will include some of these experiments in the main text and will add pointers to the main text to make them easier to locate for other readers.

---

### Author Response · Authors · 2023-11-20

We thank the reviewers for their helpful comments, and for the time and effort they spent going through our paper.

The reviewers generally seem to agree that the our momentum-based extension of PGD is sound, novel, and relevant to this community:

> [yQ9T] The framework of Kuntz et al 2023 is interesting and impactful, and extending it is a worthwile direction which this paper addresses. [cMKu] first study of the particle-based momentum method for solving MLE of latent variable models. [eLBr] In this paper, the authors successfully extended the conventional Overdamped to an Underdamped approach of the Mckean-Vlasov process for latent variable models. [1aSJ] this paper would certainly be of significant interest to the audience of this conference... They also demonstrate that the addition of momentum is done in a theoretically sound manner.

They also appear positive about its practical performance:

> [cMKu] Experimental results validate the advantages of this proposed method. [eLbr] Importantly, they demonstrate performance enhancements when their method is applied to real data. [1aSJ] The authors have taken a current algorithm, PGD, in this area and demonstrated through their experiments that the provided enhancement is an improvement... The second dataset is a more realistic problem of building a Variational Auto Encoder where the momentum method described here seems to work very effectively.

However, they diverge regarding how we could improve the manuscript and each had distinct concerns. We go through these one-by-one below. Spurred by the reviewers' suggestions, we have made two substantial modifications to the draft:

*  We established existence and uniqueness of solutions for the MPGD SDE (2), as well as asymptotic guarantees for the space discretization (see Prop. 3.1 and 5.1 respectively). The proof of which can be found in Appendix G and H.

* In Appendix K, we include additional experiments comparing the performance of MPGD, PGD, and algorithms that incorporate momentum in only the parameters or the latent variables (but not both).

---

### Meta-Review · Area_Chair_uV9u · 2023-12-10

**Metareview:**

The paper discusses a maximum likelihood optimization approach based on particle gradient descent with momentum. The aim is to improve convergence properties of EM. Three reviewers scored this paper as below acceptance threshold, while the fourth reviewer increased the score from weak reject to accept during the discussion phase. The overall sentiment was toward rejecting the paper, which the reviewers generally thought to only represent a minor technical contribution since momentum is a common method applied to any problem where SGD optimization arises.

**Justification For Why Not Higher Score:**

Originally was all 5's, the first reviewer went from 5 to 8, but it's not clear why the large jump. Doesn't seem to make a new case for acceptance.

**Justification For Why Not Lower Score:**

NA

---

### Decision · Program_Chairs · 2024-01-16

Reject